# Compilation of relative pollen productivity estimates and taxonomically harmonised RPP datasets for single continents and Northern Hemisphere extratropics

Mareike Wieczorek[1], Ulrike Herzschuh[1,2,3]

[1]Alfred-Wegener-Institut Helmholtz-Zentrum für Polar- und Meeresforschung, Telegrafenberg A45, 14473 Potsdam, Germany
[2]University of Potsdam, Institute of Environmental Sciences and Geography 14476 Potsdam, Germany
[3]University of Potsdam, Institute of Biochemistry and Biology, 14476 Potsdam, Germany

*Correspondence to*: Mareike Wieczorek (mareike.wieczorek@awi.de)

**Abstract** Relative pollen productivity (RPP) estimates are fractionate values, often in relation to Poaceae, that allow vegetation cover to be estimated from pollen counts with the help of models. RPPs are especially used in the scientific community in Europe and China, with a few studies in North America. Here we present a comprehensive compilation of available Northern Hemispheric RPP studies and their results arising from 51 publications with 60 sites and 131 taxa. This compilation allows scientists to identify data-gaps in need of further RPP analyses, but also can aid them in finding an RPP set for their study region. We also present a taxonomically harmonised, unified RPP dataset for the Northern Hemisphere and subsets for northern America (including Greenland), Europe (including arctic Russia) and China, which we generated from the available studies. The unified dataset gives the mean RPP for 55 harmonised taxa as well as fall speeds, which are necessary to reconstruct vegetation cover from pollen counts and RPP values. Data are openly available at https://doi.pangaea.de/10.1594/PANGAEA.922661 (Wieczorek and Herzschuh, 2020).

## 1 Introduction

Pollen records are widely used for the reconstruction of vegetation composition (e.g. Bartlein et al., 1984; Li et al., 2019). However, such records need to be interpreted carefully, as different taxa have different pollen productivities and dispersal abilities. While some taxa produce much and/or light pollen which is transported over large distances and thus overrepresented in the pollen records compared with vegetation, others produce little and/or heavy pollen which is hardly found in pollen records despite a high abundance of the taxon in the vegetation (e.g. Prentice, 1985; Prentice and Webb, 1986). To overcome these problems, relative pollen productivity (RPP) has been estimated and fall speed of pollen (FSP) measured or calculated for major plant taxa in several regions of the world (e.g. Baker et al., 2016; Broström et al., 2004; Commerford et al., 2013; Wang and Herzschuh, 2011). Most of these studies are limited to north/central Europe and China. Some major review studies provide RPPs for a number of sites and taxa (e.g. Broström et al., 2008; Li et al., 2018; Mazier et al., 2012), but a study compiling all available RPPs from the Northern Hemisphere - which would be useful to identify the most suitable dataset for a site-specific reconstruction - is not available. For an informed selection of the best fitting RPP values, a consistent overview of metadata and information on the RPP data assessment is required.

Combined large-scale RPP datasets are available for Europe (Mazier et al., 2012) and temperate China (Li et al, 2018). Such a compilation has, until now, not been available for northern America. By including recent studies,

we created new datasets for northern America (including Greenland), Europe (including Arctic Russia) and China (including subtropical regions). Combining these into one Northern Hemispheric RPP dataset might allow

for vegetation reconstructions using broad-scale pollen datasets by adopting a consistent approach.

Here we present a compilation of available RPP-publications, four large-scale datasets of RPP estimates and fall speeds (FSPs) for major Northern Hemispheric plant taxa.

## 2 Methods

### 2.1 Literature search

To find literature on relative pollen productivity estimates (RPP or PPE), we conducted internet searches in Google Scholar (https://scholar.google.de/) and the Web of Science (https://apps.webofknowledge.com/) for the terms "PPE", "RPP", "Pollen productivity", "Pollen productivity estimates", and various combinations of our search terms. Furthermore, we used literature cited in publications on RPPs to gain the most complete overview possible of existing literature about Northern Hemispheric RPPs. Of the resulting 63 publications from our

literature search, 12 were excluded a priori (e.g. if they did not provide RPPs or consisted only of compilations of previously available RPP data) and are marked with an x in Table 1.

### 2.2 RPP

#### 2.2.1 RPP Compilation

All RPP values and, if given, their standard deviation (SD) or standard error (SE) were collected from the

literature. If the data were only presented as figures, values were extracted with the help of Corel Draw X6. The studies of Ge et al. (2015), He et al. (2016), Li et al. (in prep), Wu et al. (2013) and Zhang et al. (2017) are only available in Chinese and RPP values where extracted from Li et al. (2018), while the study of Chen et al. (2019) was extracted from Jiang et al. (2020).

While different approaches exist to estimate RPP, the extended R-value (ERV) is the most common approach. Details on the ERV model and related assessment criteria can be found in, for example, Abraham and Kozáková (2012), Bunting et al. (2013) and Li et al. (2018). The maximum likelihood method (decreasing likelihood function score or increasing log-likelihood with distance) can be used to identify the relevant source area of pollen (RSAP) and should reach an asymptote with increasing sampling distance (Sugita 1994). For reliable

results, the vegetation sampling area should be ≥ RSAP (Sugita 1994). Unexpected behaviour of the maximum likelihood method can occur if assumptions of the ERV-model are not met (Li et al. 2018). Furthermore, a sufficient number of randomly selected sites (no of sites ≥ number of taxa for RPP-estimation) is necessary (Li et a. 2018). Last but not least, for the correct application of the REVEALS model, RPPs need to have a standard deviation provided, to allow for correct estimation of the vegetation cover.


To allow for further assessment of the presented RPP data, we collected information on, for example, the maximum likelihood, the vegetation sampling radius, and the site distribution used in the different studies. (Table A2, Wieczorek and Herzschuh (2020), https://doi.pangaea.de/10.1594/PANGAEA.922661). This will help researchers when creating customised RPP datasets. If RPP estimates for several models (e.g. ERV-

submodel 1, 2 or 3) were presented in the original study, we used all of them for the RPP compilation and added the information on which one was chosen as best fit by the original author and/or in the RPP-compilations of Mazier et al. (2012) and Li et al. (2018) (Tables A1, A3, Wieczorek and Herzschuh (2020), https://doi.pangaea.de/10.1594/PANGAEA.922661).

### 2.2.2 Continental RPP Datasets

To develop large-scale datasets for America (including Greenland), Europe (including Arctic Russia), China and the Northern Hemisphere, we confined ourselves to those studies in which the prerequisites for the ERV-model are met, i.e. a correct maximum likelihood curve, vegetation sampling radius ≥ RSAP, and number of sites ≥ number of taxa. Furthermore, we only used studies providing standard errors or standard deviations. However, some exceptions were made: studies without information on RSAP or likelihood, for example, were included if

they were previously found to be reliable by Mazier et al. (2012) or Li et al. (2018). In America particularly, only a few studies are available. We thus incorporated further studies and indicate which assumptions are not met. We followed the authors of the original publications in the choice of the most reliable ERV model, but included previous assessments of Li et al. (2018) and Mazier et al. (2012).

To be able to compare RPPs of different studies, it is necessary that all use the same reference; in our case

Poaceae in accordance with most other studies. It is possible to recalculate RPP values based on other reference taxa by setting the original reference taxon to the RPP value resulting from other studies and recalculating all other RPPs based on that ratio (Mazier et al. 2012, Li et al. 2018). Of those studies selected for the continental RPP datasets, three did not have Poaceae as the original reference and did not include an RPP for Poaceae. The study of Bunting et al. (2005, reference taxon *Quercus*) did not provide standard deviations, so we used the

values provided by Mazier et al. (2012) for this study, including the standard error. The RPPs of Li et al. (2015, reference taxon *Quercus*) were recalculated based on the mean *Quercus* RPP provided by Li et al. (2017), Zhang et al. (2017, Changbai), and Zhang et al. (2020). The RPPs of Matthias et al. (2012, reference taxon *Pinus*) were recalculated based on the mean *Pinus* RPP provided by Räsänen et al. (2007) and Abraham and Kozáková (2012). The study of Jiang et al. (2020) used *Quercus* as the reference taxon but included a value for Poaceae,

which was used as basis for recalculation.

With the remaining RPPs, two datasets of RPP were created. To obtain a reasonable taxonomic harmonisation, we assigned broader taxonomic levels to some taxa of the original publications. We kept all original values for the analyses, and calculated means per harmonised taxon for the final datasets if more than one value of finer

taxonomic levels was available (Table 2).

In the choice of reliable values, we mainly followed the strategy of Mazier et al. (2012) and Li et al. (2018). Dataset v1 includes all values of the chosen studies, except those RPPs which have an SD (or SE) > RPP. Dataset v2 is further reduced with the following steps:

- If N≥5, the highest and smallest RPPs are excluded

- If N=4, the most deviating value from the Taxa-specific mean is excluded. Exception: if two values are from the same study (they are generally similar), their mean is calculated and used for the overall mean (→ *Salix* in America; *Betula*, Fabaceae and *Larix* in China; *Rumex* in Europe). The most deviating value is chosen based on the resulting mean. Exception in America: *Betula* with 4 values from only two studies are all kept.

• If N=3, a value is only excluded if it is strongly deviating (>100% of the mean of all values) →
Caryophyllaceae of Li et al., in prep in China. Exceptions: in America Asteraceae and in Europe
Apiaceae with three values from only two studies are all kept, as the two similar ones came from the
same study.

• If N=2, all values are kept, except if one seems less reliable (*Larix*, Matthias et al. 2012)


Dataset v2 was created separately for each continent and is comparable to the Alt-1 dataset of Li et al. (2018)
and PPE.st2 of Mazier et al. (2012).

To calculate the SE of averaged RPPs, the delta method (Stuart and Ord, 1994, details in the supplement of Li et
al. 2020) was applied. For the calculation of an RPP from pollen counts, a variance-covariance matrix is created.
If only RPP ± SD (or SE) are available, the covariance is set to 0 and the final equation results in:

$$SE = \sqrt{\frac{\sum_{i=1}^{n}(var_i)}{(n * n)}}$$

Some problems arise from the labelling of standard errors and standard deviations. While some studies provide
standard deviations, others provide standard errors or give no information. Some studies provide standard
deviations, which are labelled as standard errors in other studies. Given this ambiguity, we used every value as it
is and noted if standard deviation or standard error are said to be given.

### 2.2.3 Northern Hemispheric dataset

The majority of RPP studies concentrates on China and Europe, with one study from Arctic Russia and few
studies from northern America. We thus decided to create a Northern Hemispheric dataset to be applied only for
broad-scale studies for which otherwise RPP data for various taxa would be lacking. The dataset for the whole
Northern Hemisphere was calculated with all data of the continental datasets.

We conducted Kruskal-Wallis tests on the dataset v2 between the continents for each taxon. Additionally, we
conducted the tests on the variability between taxa, once for the Northern Hemisphere and separately for each
continent, including only taxa with n>2.

### 2.3 Fall speeds

To use RPP values with, for example, the REVEALS model, fall speeds are necessary for the distance weighting
of pollen input. Fall speeds were extracted from the compiled literature of the RPP datasets. If several values
were available for one taxon (see Table A4), we calculated the mean with unique values, so if several studies had
the same fall speed for one taxon, we used only one of them. Taxonomic levels were combined according to
Table 2. Fall speeds for continental datasets were calculated based on studies used for RPP data.

### 3 Dataset description and results

#### 3.1 RPP Compilation

The compilation of RPP studies includes data from 49 studies, 43 of them using a form of the ERV-model (Tables A1, A2, A3, Wieczorek & Herzschuh (2020); https://doi.pangaea.de/10.1594/PANGAEA.922661). Twenty-nine studies used Poaceae as the reference taxon, while 20 studies used different taxa. The summary provides original RPP values with the given reference taxon. Only those used for the RPP datasets contain further RPP values recalibrated to Poaceae as the reference. An overview of all locations of the compiled RPP studies is given in Fig. 1, which clearly shows the absence of studies in Central Asia and large parts of Russia. Only a few studies have been conducted in North America. Not all studies provide information on the likelihood or RSAP, hampering the assessment of the reliability of the presented RPP values. Other studies do not provide standard deviations, leading to inaccurate results in subsequent applications.

#### 3.2 RPP Datasets

Of 60 RPP-datasets, 28 (coming from 23 studies) were excluded prior to the calculation of the combined RPP datasets.

Filipova-Marinova et al. (2010), Andersen (1967), Theuerkauf et al. (2015), Sjögren (2013), and Sjögren et al. (2008a, 2008b) do not present RPP-values based on ERV-models.

The likelihood function score should decrease and approach an asymptote when reaching the RSAP (see methods). Within the sampled vegetation area, the curve does not approach an asymptote in the studies of Calcote (1995) and Chaput & Gajewski (2018), meaning that vegetation composition is not studied up to the RSAP. As furthermore Poaceae was not used as the referenced taxon, we decided to not use these data despite the scarcity of studies in northern America. In the studies of Han et al. (2017) and Xu et al. (2014), the likelihood function score increases. We followed the assessment of Li et al. (2018) and did not incorporate these RPPs. The likelihood function score further increases in the study of Ge et al. (2017, year 2014 data). Data from He et al., (2016) are not used in accordance with Li et al. (2018), as pollen are sampled from a pollen trap, which might behave differently compared to moss pollsters or lakes. In the study of Hjelle and Sugita (2012), the likelihood function score does not approach an asymptote. Sugita et al. (1999, 2006) do not provide information on the likelihood and RPP values are given without information on standard deviation or standard error. The studies of Twiddle et al. (2012) and Li et al. (2011) do not provide standard deviations or errors for the presented RPP values. The study of Wu et al. (2013, original publication in Chinese) was rejected by Li et al. (2018) because of a too large sampling area and we followed this assessment. Theuerkauf et al. (2013) does not provide information on the maximum likelihood or the RSAP. Data from Chen et al. (2019) were extracted from Jiang et al. (2020) but included insufficient information on the study design and the ERV-approach. Data from the study of Qin et al. (2020) have been rejected has they had very high values for most taxa compared to other studies, which we assume was a systematic problem of the study. The study of Fang et al. (2019) was excluded because it was designed to test different methods for RPP estimation and was carried out in patchy vegetation without enough sites.

On the other hand, some studies were incorporated despite missing information or likelihood curves that did not meet our criteria:

Hjelle (1998) and Nielsen (2004) do not provide information on the likelihood but have been included in the dataset of Mazier et al. (2012, i.e. was assessed by an expert). Bunting et al. (2013) do not provide information on the likelihood nor do they sample vegetation up to the value of RSAP. The scarcity of data from northern America together with Poaceaea as a reference taxon led us to the decision to keep these RPPs. While the likelihood function score should decrease and reach an asymptote at the radius of the RSAP, the log-likelihood should increase before reaching the asymptote. This is not the case for the study of Commerford et al. (2013), but data have been included due to scarcity of American studies. At the boreal forest site of Hopla (2017), the likelihood function score does not reach an asymptote. Again, these data have been included due to the scarcity of American studies.

### 3.3 Continental and Northern Hemispheric RPP Datasets

All RPP data in the final dataset are given relative to Poaceae. Of 49 publications covering 60 sites, 27 publications and 31 sites are included in the final PPE datasets (10 studies and 11 datasets for China, 14 studies and 16 datasets for Europe, 3 studies and 4 datasets for America). We have RPP data for 33 taxa in China, 34 taxa in Europe and 25 taxa in northern America. The Northern Hemispheric dataset consists of RPP values and fall speeds for 55 taxa (Tables 3-6, Wieczorek and Herzschuh (2020); https://doi.pangaea.de/10.1594/PANGAEA.922661). Twenty-eight taxa are available in only one of the continental datasets (13 in China, 6 in America, 9 in Europe).

In Dataset v1, 11 RPP values have an SD <1 between the different datasets, while 15 have an SD >1 (Fig. 2). The size of RPP as well as the variability of RPP values between continents partly differs between Dataset v1 and v2 (Fig. 2, 3).

Testing the RPP values used to create the combined dataset on the variability between taxa shows that the taxa themselves are significantly different from each other (**Northern Hemisphere**: Kruskal-Wallis chi-squared = 99.337, df = 29, p <0.001 with *Acer*, *Alnus*, Apiaceae, *Artemisia*, Asteraceae, *Betula*, *Carpinus*, Caryophyllaceae, Cerealia, Chenopodiaceae, *Corylus*, Cyperaceae, Ericales, Fabaceae, *Fagus*, *Fraxinus*, *Juglans*, Lamiaceae, *Larix*, *Picea*, *Pinus*, Plataginaceae, *Populus*, *Quercus*, Ranunculaceae, Rosaceae, Rubiaceae, *Rumex*, *Salix*, *Tilia*; **China**: Kruskal-Wallis chi-squared = 27.599, df = 9, p <0.01, with *Artemisia*, Asteraceae, *Betula*, Chenopodiaceae, Cyperaceae, Fabaceae, *Juglans*, *Larix*, *Pinus*, *Quercus*; **Europe**: Kruskal-Wallis chi-squared = 56.5, df = 21, p <0.001, with *Acer*, *Alnus*, Apiaceae, Asteraceae, *Betula*, *Carpinus*, Cerealia, *Corylus*, Cyperaceae, Ericales, *Fagus*, *Fraxinus*, *Picea*, *Pinus*, Plataginaceae, *Quercus*, Ranunculaceae, Rosaceae, Rubiaceae, *Rumex*, *Salix*, *Tilia*; **America**: Kruskal-Wallis chi-squared = 6.7091, df = 2, p <0.05, with *Asteraceae*, *Betula*, *Salix*). Furthermore, while some taxa strongly differ between continents when looking at the absolute deviation (e.g. *Artemisia*, Fabaceae or *Larix*) others show no large deviation from the overall Northern Hemispheric mean (e.g. *Salix*, *Betula*; Fig. 4). And while we found overall significant differences between taxa (described above), we did not find significant differences between datasets for single taxa (n=6) from two continents when applying the Kruskal-Wallis test, except for Asteraceae (Fig. 4). This means the differences between continents are rather small compared to differences between taxa.

225     Comparison with taxa available in the compilations of Mazier et al. (2012, Europe) and Li et al. (2018, temperate China) clearly shows differences in absolute RPP values or a high absolute deviation for some taxa (Fig. 5, e.g. *Juniperus*, *Artemisia*, Rosaceae), while many others (e.g. *Alnus*, *Quercus* or Ranunculaceae) have a similar range of values, especially when considering the absolute deviation.

## 4 Discussion and data quality

### 4.1 RPP compilation

The compilation is, to our knowledge, the first overview of available RPP studies covering the whole Northern Hemisphere. It highlights data gaps with respect to certain regions and taxa and as such guides the design of future RPP studies. Good geographic coverage is, to date, limited to central/northern Europe and China (Fig. 1). RPP studies in Russian and North American boreal forests as well as in tropical regions are largely lacking. The
compilation covers most common taxa, mostly at the genus level, but the taxonomic resolution of available RPPs varies between studies and depends on the level to which pollen has been identified. Furthermore, while some taxa have a large number of available RPPs, for 24 taxa (i.e. ~40 %) only one or two datasets are available. By including additional metadata, our compilation is useful for the identification of available RPP sets at specific sites and regions and indicates how suitable they may be for further research. For many studies, however,
missing details needed for the evaluation (e.g. information on the maximum likelihood method) or use (e.g. standard deviation) of the RPP values lower their usefulness. It should therefore be stated clearly if data are presented with standard deviation or standard error.

### 4.2 Continental and hemispheric PPE datasets

Using RPPs for pollen-based quantitative vegetation reconstruction (Sugita, 2007; Theuerkauf et al., 2016) has
improved our understanding of environmental change (e.g. Marquer et al., 2014). In this paper, we present RPP datasets for three continents and one dataset of Northern Hemispheric extratropical RPPs and corresponding fall speeds, based on a compilation of studies.

We found that RPP values partly vary between the three continental datasets. Some uncertainty arises due to the
use of inconsistent reference taxa. Most studies used Poaceae, a widespread family, whose pollen is easy to identify and often preserved in a good state. However, as discussed by Broström et al. (2008), the pollen cannot be identified to species level and different studies may thus have used different species of Poaceae for the reference. Other taxa at higher taxonomic resolution such as Quercus or Acer are therefore sometimes used as the reference taxon (see Table A1, Wieczorek and Herzschuh (2020);
https://doi.pangaea.de/10.1594/PANGAEA.922661).

Reasons for variable RPP values have been discussed in depth by Broström et al. (2008) and Li et al. (2018), and are mainly methodological factors such as different sampling designs and environmental factors such as vegetation characteristics. Furthermore, pollen taxa from different sites can contain different species. Li et al.
(2018) discussed in detail for Pinus and Artemisia, that vegetation structure and climate of different Chinese study regions, but also methodological differences like the pollen sample type (moss vs. lake sediment) and

vegetation sampling method, can explain the variability of RPPs within one taxon even better than the occurrence of different taxa. This will be even more apparent when combining data for the whole Northern Hemisphere. However, our compilation clearly indicates that taxa have mostly characteristic RPP values (i.e. within-species variability is low compared to variability between species), while we found no significant differences between continents (i.e. variability within continents is not lower than variability between continents). This implies, when aiming to compare vegetation change between continents, that transformation of pollen data using RPP from another continent is better than keeping the data untransformed. While one has to keep in mind the limited amount of data influencing the statistical power, we conclude that there is no particular reason to not set up a Northern Hemispheric RPP dataset. Still, before applying one of the datasets presented, researchers should consult the original publication to be sure it fits their needs and standards and be aware of the rather problematic use of SD and SE, which might have influenced our presented SEs.

## 5 How to use the datasets

The RPP compilation can be used to get a good overview of existing RPP studies, to identify research gaps and to find RPPs to apply at one's study area. It is important (i) to use only those RPP data which have been evaluated by experts or the author as best fit and (ii) to look at the original publication for further information on how the RPP estimates have been generated.

The continental datasets can be applied to assess vegetation changes using broad-scale pollen datasets. It is important to keep in mind that different taxa with different pollen productivities and dispersal abilities are combined in one RPP value and the application to such broad-scale datasets can only be an approximation. This is especially important for the Northern Hemispheric dataset, which should not be applied to calculate site-specific vegetation compositions. This dataset fills data gaps of RPP values in various regions, but at the cost of accuracy. We consider the presented averaged RPP values as a tool for data transformation to be applied to broad-scale pollen datasets. Using the dataset in this way can account for differences in pollen productivities and transportation rather than obtaining fully reliable quantitative information about the vegetation cover around a specific site.

## 6 Data Availability

The RPP compilation as well as the taxonomically harmonised continental RPP datasets are available at https://doi.pangaea.de/10.1594/PANGAEA.922661 (Wieczorek and Herzschuh, 2020).

## 7 Author Contribution

MW and UH designed the study and wrote the Manuscript, MW carried out the analyses and produced tables and figures.

## 8 Competing interests

The authors declare that they have no conflict of interest.

## 9 Acknowledgements

The study was supported by and conducted as part of the ERC consolidator grant "GlacialLegacy" (Call: ERC-2017-COG, Project Reference: 772852) and PalMod Initiative (Grant 01LP1510C). We thank all scientists conducting research on pollen productivity, whose previous work and published data made our compilation possible. Many thanks Marie-José Gaillard and an anonymous referee for taking the time to give us very helpful and constructive reviews and suggestions to improve our dataset and manuscript. We thank C. Jenks for language editing.

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

TABLES

**Table 1: Publications returned by our literature research for relative pollen productivity (RPP) estimates. Literature not included in all further evaluations is given in italics and marked with an x. If a study has been further examined but did not use the ERV-model it is noted in brackets.**

| | |
|---|---|
| Abraham and Kozáková, 2012 | Li et al., 2017b |
| Andersen, 1967 (no ERV) | Li et al., 2018 (review) |
| Baker et al., 2016 | Li et al., in prep (from Li et al., 2018) |
| *x Binney et al., 2011 (no RPPs provided)* | Matthias et al., 2012 |
| Broström et al., 2004 | Mazier et al., 2008 |
| *x Broström et al., 2008 (review)* | Mazier et al., 2012 (review) |
| *x Broström, 2002 (PhD thesis, data given in publications)* | *x McLauchlan et al., 2011 (count data)* |
| *x Bunting and Hjelle, 2010 (comparison of different data collection methods)* | Nielsen, 2004 |
| Bunting et al., 2005 | Niemeyer et al., 2015 |
| Bunting et al., 2013 | Poska et al., 2011 |
| Calcote, 1995 | Qin et al., 2020 (from Jiang et al., 2020) |
| Chaput and Gajewski, 2018 | Räsänen et al., 2007 |
| Chen et al., 2019 | *x Sjögren et al., 2006 (pollen productivity, not PPEs)* |
| Commerford et al., 2013 | Sjögren et al., 2008a (no ERV) |
| *x Duffin and Bunting, 2008 (southern Africa - not our focus)* | Sjögren et al., 2008b (no ERV) |
| Fang et al., 2019 | Sjögren, 2013 (no ERV) |
| Filipova-Marinova et al., 2010 (no ERV) | Soepboer et al., 2007 |
| Ge et al., 2015 (from Li et al., 2018) | *x Soepboer et al., 2008 (no new PPEs)* |
| Ge et al., 2017 | Sugita et al., 1999 |
| Grindean et al., 2019 | Sugita et al., 2006 |
| Han et al., 2017 | *x Sugita et al., 2010 (absolute pollen values)* |
| He et al., 2016 (from Li et al., 2018) | Theuerkauf et al., 2013 |
| *x Heide and Bradshaw, 1982 (pollen percentages)* | Theuerkauf et al., 2015 (no ERV) |
| *x Hellman et al., 2008 (no new RPPs)* | *x Trondman et al., 2015 (uses PFTs)* |
| Hjelle and Sugita, 2012 | Twiddle et al., 2012 |
| Hjelle, 1998 | von Stedingk et al., 2008 |
| Hopla, 2017 | Wang and Herzschuh, 2011 |
| Jiang et al., 2020 | Wu et al., 2013 (from Li et al., 2018) |
| Kuneš et al., 2019 | Xu et al., 2014 |
| Li et al., 2011 | Zhang et al., 2017 (from Li et al., 2018) |
| Li et al., 2015 | Zhang et al., 2020 |
| Li et al., 2017a | |

**Table 2: Combination of taxonomic levels.**

| Pollen morphological taxon | Original morphological pollen taxa |
|---|---|
| *Abies* | *Abies + Abies alba* |
| *Acer* | *Acer + Acer rubrum + Acer saccharum* |
| *Alnus* | *Alnus + Alnus*_shrub + *Alnu*s_tree |
| Asteraceae | Asteraceae + *Achillea*-type + *Ambrosia* + Anthemis arvensis type + Asterac SF Cichor + *Aster/Anthemis* type + Compositae + *Leucanthemum vulgare* + *Saussurea* t + *Senecio* type + *Taraxacum* type |
| *Betula* | *Betula + Betula*_shrub + *Betula*_tree |
| Brassicaceae | Brassicaceae + *Sinapis type* |
| *Carpinus* | *Carpinus + Carpinus betulus + Carpinus orientalis* |
| Cerealia | *Avena triticum + Avena* type + *Avena* type b + Cerealia + *Hordeum* type + *Secale + Triticum* type |
| *Corylus* | *Corylus + Corylus avellana* |
| Elaeagnaceae | Elaeagnaceae + *Hippohae* |
| Ericales | Ericaceae + *Calluna + Calluna vulgaris + Empetrum + Vaccinium* |
| Fabaceae | Fabaceae + *Robinia/Sophora + Cercis* |
| *Fagus* | *Fagus + Fagus sylvatica* |
| *Fraxinus* | *Fraxinus + Fraxinus excelsior* |
| *Juglans* | *Juglans + Juglans regia* |
| *Juniperus* | *Juniperus + Juniperus communis* |
| Lamiaceae | Lamiaceae + *Mentha* type (*Thymus*) + *Thymus praecox* |
| *Larix* | *Larix + "Larix+Pseudotsuga"* |
| *Picea* | *Picea + Picea abies* |
| *Pinus* | *Pinus + Pinus cembra + Pinus sylvestris* |
| Plantaginaceae | *Plantago + Plantago lanceolata + Plantago media + Plantago montana* type + *Plantago maritima* |
| Poaceae | Poaceae + Graminae |
| Ranunculaceae | Ranunculaceae + *Ranunculus acris* type + *Trollius europaeus* |
| Rosaceae | Rosaceae + *Filipendula + Potentilla* t. |
| Rubiaceae | Rubiaceae + *Galium* type |
| *Rumex* | *Rumex + Rumex* sect. *acetosa + Rumex acetosella + Rumex acetosa* t |
| *Tilia* | *Tilia + Tilia begoniifolia + Tilia tomentosa + Tilia cordata* |

Table 3: Overview of continental and Northern Hemispheric relative pollen productivity (RPP) estimates for woody vegetation with their standard error (SE) (dataset v1) and fall speeds. All values are relative to Poaceae. See Table A1 for information on original RPP data, Table A4 for information on original fall speed values, and methods on the creation of dataset v1 (Wieczorek and Herzschuh (2020), https://doi.pangaea.de/10.1594/PANGAEA.922661).

| Type | Target taxon (pollen morphological) | China | | | | America | | | | Europe | | | | Northern Hemisphere | | | |
|---|---|---|---|---|---|---|---|---|---|---|---|---|---|---|---|---|---|
| | | n | RPP v1 | SE | FS (m/s) | n | RPP v1 | SE | FS (m/s) | n | RPP v1 | SE | FS (m/s) | n | RPP v1 | SE | FS (m/s) |
| woody | *Acer* | 0 | | | | 0 | | | | 3 | 0.23 | 0.043 | 0.056 | 3 | 0.23 | 0.043 | 0.056 |
| woody | Anacardiaceae | 1 | 0.45 | 0.07 | 0.027 | 0 | | | | 0 | | | | 1 | 0.45 | 0.070 | 0.027 |
| woody | Rosaceae | 2 | 0.53 | 0.05 | 0.017 | 1 | 0.35 | 0.030 | 0.015 | 6 | 1.08 | 0.159 | 0.012 | 9 | 0.88 | 0.107 | 0.014 |
| woody | *Tilia* | 1 | 0.40 | 0.10 | 0.030 | 0 | | | | 4 | 1.17 | 0.098 | 0.032 | 5 | 1.02 | 0.081 | 0.030 |
| woody | Moraceae | 0 | | | | 1 | 1.10 | 0.550 | 0.016 | 0 | | | | 1 | 1.10 | 0.550 | 0.016 |
| woody | Cupressaceae | 1 | 1.11 | 0.09 | 0.010 | 0 | | | | 1 | 1.11 | 0.090 | 0.010 | 1 | 1.11 | 0.090 | 0.010 |
| woody | *Salix* | 0 | | | | 4 | 2.02 | 0.188 | 0.016 | 4 | 0.59 | 0.053 | 0.028 | 8 | 1.30 | 0.098 | 0.022 |
| woody | *Populus* | 0 | | | | 2 | 0.67 | 0.085 | 0.026 | 1 | 3.42 | 1.600 | 0.026 | 3 | 1.59 | 0.536 | 0.026 |
| woody | Rubiaceae | 1 | 1.23 | 0.36 | 0.019 | 0 | | | | 5 | 1.75 | 0.138 | 0.019 | 6 | 1.67 | 0.129 | 0.019 |
| woody | *Corylus* | 1 | 3.17 | 0.20 | 0.012 | 0 | | | | 4 | 1.44 | 0.066 | 0.025 | 5 | 1.78 | 0.066 | 0.019 |
| woody | *Ulmus* | 2 | 2.24 | 0.46 | 0.024 | 0 | | | | 0 | | | | 2 | 2.24 | 0.462 | 0.026 |
| woody | *Fraxinus* | 2 | 1.05 | 0.18 | 0.020 | 0 | | | | 5 | 2.97 | 0.252 | 0.022 | 7 | 2.42 | 0.187 | 0.020 |
| woody | *Fagus* | 0 | | | | 0 | | | | 5 | 2.92 | 0.133 | 0.056 | 5 | 2.92 | 0.133 | 0.056 |
| woody | *Juglans* | 5 | 3.28 | 0.12 | 0.032 | 0 | | | | 0 | | | | 5 | 3.28 | 0.119 | 0.032 |
| woody | *Larix* | 4 | 2.31 | 0.16 | 0.119 | 0 | | | | 2 | 5.73 | 1.165 | 0.126 | 6 | 3.45 | 0.402 | 0.122 |
| woody | *Quercus* | 7 | 2.50 | 0.05 | 0.021 | 1 | 2.08 | 0.430 | 0.035 | 7 | 4.88 | 0.087 | 0.035 | 15 | 3.58 | 0.056 | 0.024 |
| woody | *Carpinus* | 0 | | | | 0 | | | | 5 | 4.31 | 0.216 | 0.042 | 5 | 4.31 | 0.216 | 0.042 |
| woody | *Castanea* | 2 | 5.87 | 0.25 | 0.014 | 0 | | | | 0 | | | | 2 | 5.87 | 0.245 | 0.014 |
| woody | *Picea* | 1 | 29.40 | 0.87 | 0.082 | 1 | 2.80 | | 0.056 | 6 | 2.57 | 0.114 | 0.056 | 8 | 5.96 | 0.138 | 0.065 |
| woody | *Abies* | 0 | | | | 0 | | | | 2 | 6.88 | 1.442 | 0.120 | 2 | 6.88 | 1.442 | 0.120 |
| woody | *Betula* | 4 | 11.29 | 0.17 | 0.016 | 4 | 6.19 | 0.149 | 0.051 | 8 | 5.67 | 0.335 | 0.024 | 16 | 7.21 | 0.177 | 0.028 |
| woody | *Alnus* | 0 | | | | 1 | 2.70 | 0.120 | 0.021 | 6 | 9.42 | 0.308 | 0.021 | 7 | 8.46 | 0.264 | 0.021 |
| woody | *Juniperus* | 0 | | | | 1 | 20.67 | 1.540 | 0.016 | 1 | 7.94 | 1.280 | 0.016 | 2 | 14.31 | 1.001 | 0.016 |
| woody | *Pinus* | 7 | 17.49 | 0.46 | 0.032 | 0 | | | | 6 | 11.32 | 0.539 | 0.036 | 13 | 14.64 | 0.352 | 0.033 |
| woody | Thymelaceae | 1 | 33.05 | 3.78 | 0.009 | 0 | | | | 0 | | | | 1 | 33.05 | 3.780 | 0.009 |



**Table 4:** Overview of continental and Northern Hemispheric relative pollen productivity (RPP) values for herbaceous vegetation with their standard error (SE) (dataset v1) and fall speeds. All values are relative to Poaceae. See Table A1 for information on original RPP data, Table A4 for information on original fall speed values, and methods on the creation of dataset v1 (Wieczorek and Herzschuh (2020), https://doi.pangaea.de/10.1594/PANGAEA.922661). The group of wild herbs is taken from the publication of Matthias et al. (2012) and consists of uncultivated terrestrial herb pollen, including Poaceae, Plantago lanceolata, Rumex acetosa, R. acemsella, and Chenopodiacea.

| Type | Target taxon (Pollen morphological) | China | | | | America | | | | Europe | | | | Northern Hemisphere | | | |
|---|---|---|---|---|---|---|---|---|---|---|---|---|---|---|---|---|---|
| | | n | RPP v1 | SE | FS (m/s) | n | RPP v1 | SE | FS (m/s) | n | RPP v1 | SE | FS (m/s) | n | RPP v1 | SE | FS (m/s) |
| herbaceous | wild.herbs | 0 | | | | 0 | | | | 1 | 0.07 | 0.070 | 0.034 | 1 | 0.07 | 0.070 | 0.034 |
| herbaceous | *Equisetum* | 0 | | | | 1 | 0.09 | 0.020 | 0.021 | 0 | | | | 1 | 0.09 | 0.020 | 0.021 |
| herbaceous | Convolvulaceae | 1 | 0.18 | 0.03 | 0.043 | 0 | | | | 0 | | | | 1 | 0.18 | 0.030 | 0.043 |
| herbaceous | Fabaceae | 4 | 0.35 | 0.04 | 0.020 | 1 | 0.02 | 0.020 | 0.021 | 1 | 0.40 | 0.070 | 0.021 | 6 | 0.30 | 0.029 | 0.020 |
| herbaceous | Orobanchaceae | 0 | | | | 1 | 0.33 | 0.040 | 0.038 | 0 | | | | 1 | 0.33 | 0.040 | 0.038 |
| herbaceous | Brassicaceae | 1 | 0.89 | 0.18 | 0.020 | 0 | | | | 1 | 0.07 | 0.040 | 0.021 | 2 | 0.48 | 0.092 | 0.021 |
| herbaceous | Ericales | 0 | | | | 1 | 0.53 | | 0.038 | 9 | 0.86 | 0.079 | 0.030 | 10 | 0.83 | 0.071 | 0.032 |
| herbaceous | Poaceae | 10 | 1.00 | 0.03 | 0.021 | 4 | 1.00 | 0.048 | 0.026 | 14 | 1.00 | | 0.035 | 28 | 1.00 | 0.012 | 0.023 |
| herbaceous | Lamiaceae | 2 | 1.24 | 0.19 | 0.015 | 1 | 0.72 | 0.080 | 0.031 | 0 | | | | 3 | 1.06 | 0.127 | 0.019 |
| herbaceous | *Sambucus nigra*-type | 0 | | | | 0 | | | | 1 | 1.30 | 0.120 | 0.013 | 1 | 1.30 | 0.120 | 0.013 |
| herbaceous | Asteraceae | 6 | 3.80 | 0.15 | 0.029 | 3 | 0.59 | 0.131 | 0.025 | 10 | 0.25 | 0.016 | 0.032 | 19 | 1.42 | 0.053 | 0.029 |
| herbaceous | Liliaceae | 1 | 1.49 | 0.11 | 0.014 | 0 | | | | 0 | | | | 1 | 1.49 | 0.110 | 0.014 |
| herbaceous | Amaryllidaceae | 1 | 1.64 | 0.09 | 0.013 | 0 | | | | 0 | | | | 1 | 1.64 | 0.090 | 0.013 |
| herbaceous | Cornaceae | 0 | | | | 1 | 1.72 | 0.140 | 0.044 | 0 | | | | 1 | 1.72 | 0.140 | 0.044 |
| herbaceous | Cyperaceae | 5 | 4.17 | 0.10 | 0.029 | 2 | 0.98 | 0.025 | 0.031 | 8 | 0.56 | 0.026 | 0.035 | 15 | 1.82 | 0.036 | 0.030 |
| herbaceous | *Rumex* | 0 | | | | 2 | 2.79 | 0.172 | 0.014 | 4 | 1.62 | 0.209 | 0.018 | 6 | 2.01 | 0.151 | 0.015 |
| herbaceous | Apiaceae | 0 | | | | 0 | | | | 3 | 2.13 | 0.410 | 0.042 | 3 | 2.13 | 0.410 | 0.042 |
| herbaceous | Campanulaceae | 0 | | | | 1 | 2.29 | 0.140 | 0.022 | 0 | | | | 1 | 2.29 | 0.140 | 0.022 |
| herbaceous | Ranunculaceae | 1 | 7.86 | 2.65 | 0.007 | 1 | 1.95 | 0.100 | 0.015 | 5 | 1.39 | 0.161 | 0.014 | 7 | 2.40 | 0.396 | 0.013 |
| herbaceous | Cerealia | 0 | | | | 0 | | | | 6 | 3.51 | 0.500 | 0.069 | 6 | 3.51 | 0.500 | 0.069 |
| herbaceous | Plantaginaceae | 0 | | | | 1 | 5.96 | 0.310 | 0.019 | 10 | 3.30 | 0.207 | 0.028 | 11 | 3.54 | 0.190 | 0.026 |
| herbaceous | *Thalictrum* | 0 | | | | 1 | 4.65 | 0.300 | 0.012 | 0 | | | | 1 | 4.65 | 0.300 | 0.013 |
| herbaceous | Chenopodiaceae | 5 | 7.57 | 0.64 | 0.014 | 0 | | | 0.011 | 1 | 4.28 | 0.270 | 0.019 | 6 | 7.02 | 0.532 | 0.014 |
| herbaceous | *Urtica* | 0 | | | | 0 | | | | 1 | 10.52 | 0.310 | 0.007 | 1 | 10.52 | 0.310 | 0.007 |
| herbaceous | *Artemisia* | 8 | 14.80 | 0.30 | 0.010 | 1 | 1.35 | 0.240 | 0.016 | 2 | 4.33 | 1.592 | 0.014 | 11 | 11.67 | 0.363 | 0.012 |
| herbaceous | Elaeagnaceae | 2 | 13.64 | 0.69 | 0.012 | 0 | | | | 0 | | | | 2 | 13.64 | 0.686 | 0.012 |
| herbaceous | *Humulus* | 1 | 16.43 | 1.00 | 0.010 | 0 | | | | 0 | | | | 1 | 16.43 | 1.000 | 0.010 |
| herbaceous | Amaranthaceae | 1 | 21.35 | 2.34 | 0.010 | 0 | | | | 0 | | | | 1 | 21.35 | 2.340 | 0.010 |
| herbaceous | Caryophyllaceae | 3 | 28.78 | 1.95 | 0.026 | 1 | 0.60 | 0.050 | 0.041 | 0 | | | | 4 | 21.74 | 1.463 | 0.032 |
| herbaceous | *Sanguisorba* | 1 | 24.07 | 3.50 | 0.012 | 0 | | | | 0 | | | | 1 | 24.07 | 3.500 | 0.012 |


Table 5: Overview of continental and Northern Hemispheric relative pollen productivity (RPP) values for woody vegetation with their standard error (SE) (dataset v2) and fall speeds. All values are relative to Poaceae. See Table A1 for information on original RPP data, Table A4 for information on original fall speed values, and methods on the creation of dataset v2 (Wieczorek and Herzschuh (2020), https://doi.pangaea.de/10.1594/PANGAEA.922661).

| Type | Target taxon (Pollen morphological) | China | | | | America | | | | Europe | | | | Northern Hemisphere | | | |
| --- | --- | --- | --- | --- | --- | --- | --- | --- | --- | --- | --- | --- | --- | --- | --- | --- | --- |
| | | n | RPP v2 | SE | FS (m/s) | n | RPP v2 | SE | FS (m/s) | n | RPP v2 | SE | FS (m/s) | n | RPP v2 | SE | FS (m/s) |
| woody | Acer | 0 | | | | 0 | | | | 3 | 0.23 | 0.04 | 0.056 | 3 | 0.23 | 0.043 | 0.056 |
| woody | Acardiaceae | 1 | 0.45 | 0.07 | 0.027 | 0 | | | | 0 | | | | 1 | 0.45 | 0.070 | 0.027 |
| woody | Salix | 0 | | | | 3 | 0.68 | 0.01 | 0.016 | 3 | 0.39 | 0.06 | 0.028 | 6 | 0.54 | 0.030 | 0.022 |
| woody | Rosaceae | 2 | 0.53 | 0.05 | 0.017 | 1 | 0.35 | 0.03 | 0.015 | 4 | 0.97 | 0.11 | 0.012 | 7 | 0.76 | 0.064 | 0.014 |
| woody | Tilia | 1 | 0.40 | 0.10 | 0.030 | | | | | 3 | 0.93 | 0.09 | 0.032 | 4 | 0.80 | 0.070 | 0.030 |
| woody | Moraceaea | 0 | | | | 1 | 1.10 | 0.55 | 0.016 | 0 | | | | 1 | 1.10 | 0.550 | 0.016 |
| woody | Cupressaceae | 1 | 1.11 | 0.09 | 0.010 | 0 | | | | 0 | | | | 1 | 1.11 | 0.090 | 0.010 |
| woody | Larix | 3 | 1.60 | 0.20 | 0.119 | 1 | 0.16 | 0.05 | 0.126 | 0 | | | | 4 | 1.24 | 0.153 | 0.122 |
| woody | Rubiaceae | 1 | 1.23 | 0.36 | 0.019 | 0 | | | | 3 | 1.56 | 0.12 | 0.019 | 4 | 1.48 | 0.126 | 0.019 |
| woody | Corylus | 1 | 3.17 | 0.20 | 0.012 | 0 | | | | 3 | 1.05 | 0.03 | 0.025 | 4 | 1.58 | 0.055 | 0.019 |
| woody | Populus | 0 | | | | 2 | 0.67 | 0.09 | 0.026 | 1 | 3.42 | 1.60 | 0.025 | 3 | 1.59 | 0.536 | 0.026 |
| woody | Ulmus | 2 | 2.24 | 0.46 | 0.024 | 0 | | | | 0 | | | | 2 | 2.24 | 0.462 | 0.026 |
| woody | Fagus | 0 | | | | 0 | | | | 3 | 2.35 | 0.11 | 0.056 | 3 | 2.35 | 0.107 | 0.056 |
| woody | Fraxinus | 2 | 1.05 | 0.18 | 0.020 | 0 | | | | 5 | 2.97 | 0.25 | 0.022 | 7 | 2.42 | 0.187 | 0.020 |
| woody | Quercus | 5 | 2.28 | 0.07 | 0.021 | 1 | 2.08 | 0.43 | 0.035 | 5 | 2.92 | 0.10 | 0.035 | 11 | 2.56 | 0.068 | 0.024 |
| woody | Juglans | 3 | 2.80 | 0.11 | 0.032 | 0 | | | | 0 | | | | 3 | 2.80 | 0.113 | 0.032 |
| woody | Carpinus | 0 | | | | 0 | | | | 3 | 3.09 | 0.28 | 0.042 | 3 | 3.09 | 0.284 | 0.042 |
| woody | Castanea | 2 | 5.87 | 0.25 | 0.014 | 0 | | | | 0 | | | | 2 | 5.87 | 0.245 | 0.014 |
| woody | Picea | 1 | 29.40 | 0.87 | 0.082 | 1 | 2.80 | | 0.056 | 4 | 1.65 | 0.15 | 0.056 | 6 | 6.46 | 0.177 | 0.065 |
| woody | Abies | 0 | | | | 0 | | | | 2 | 6.88 | 1.44 | 0.120 | 2 | 6.88 | 1.442 | 0.120 |
| woody | Betula | 3 | 12.45 | 0.15 | 0.016 | 4 | 6.19 | 0.15 | 0.051 | 6 | 4.94 | 0.44 | 0.024 | 13 | 7.06 | 0.212 | 0.028 |
| woody | Alnus | 0 | | | | 1 | 2.70 | 0.12 | 0.021 | 4 | 8.49 | 0.22 | 0.021 | 5 | 7.33 | 0.174 | 0.021 |
| woody | Pinus | 5 | 16.68 | 0.51 | 0.032 | 0 | | | | 4 | 10.86 | 0.80 | 0.036 | 9 | 14.10 | 0.454 | 0.033 |
| woody | Juniperus | 0 | | | | 1 | 20.67 | 1.54 | 0.016 | 1 | 7.94 | 1.28 | 0.016 | 2 | 14.31 | 1.001 | 0.016 |
| woody | Thymelaceae | 1 | 33.05 | 3.78 | 0.009 | 0 | | | | 0 | | | | 1 | 33.05 | 3.780 | 0.009 |

**Table 6: Overview of continental and Northern Hemispheric relative pollen productivity (RPP) values for herbaceous vegetation with their standard error (SE) (dataset v2) and fall speeds. All values are relative to Poaceae. See Table A1 for information on original RPP data, Table A4 for information on original fall speed values, and methods on the creation of dataset v2 (Wieczorek and Herzschuh (2020), https://doi.pangaea.de/10.1594/PANGAEA.922661). The group of wild herbs is taken from the publication of Matthias et al. (2012).**

| Type | Target taxon (Pollen morphological) | China | | | | America | | | | Europe | | | | Northern Hemisphere | | | |
|---|---|---|---|---|---|---|---|---|---|---|---|---|---|---|---|---|---|
| | | n | RPP v2 | SE | FS (m/s) | n | RPP v2 | SE | FS (m/s) | n | RPP v2 | SE | FS (m/s) | n | RPP v2 | SE | FS (m/s) |
| herbaceous | wild.herbs | 0 | | | | 0 | | | | 1 | 0.07 | 0.07 | 0.034 | 1 | 0.07 | 0.07 | 0.034 |
| herbaceous | *Equisetum* | 0 | | | | 1 | 0.09 | 0.02 | 0.021 | 0 | | | | 1 | 0.09 | 0.02 | 0.021 |
| herbaceous | Convolvulaceae | 1 | 0.18 | 0.03 | 0.043 | 0 | | | | 0 | | | | 1 | 0.18 | 0.03 | 0.043 |
| herbaceous | Fabaceae | 3 | 0.20 | 0.05 | 0.020 | 1 | 0.02 | 0.02 | 0.021 | 1 | 0.40 | 0.07 | 0.021 | 5 | 0.21 | 0.03 | 0.020 |
| herbaceous | Orobanchaceae | 0 | | | | 1 | 0.33 | 0.04 | 0.038 | 0 | | | | 1 | 0.33 | 0.04 | 0.038 |
| herbaceous | Ericales | 0 | | | | 1 | 0.53 | | 0.038 | 7 | 0.44 | 0.02 | 0.030 | 8 | 0.45 | 0.01 | 0.032 |
| herbaceous | Brassicaceae | 1 | 0.89 | 0.18 | 0.020 | 0 | | | | 1 | 0.07 | 0.04 | 0.022 | 2 | 0.48 | 0.09 | 0.021 |
| herbaceous | Poaceae | 10 | 1.00 | 0.03 | 0.021 | 4 | 1.00 | 0.05 | 0.026 | 14 | 1.00 | | 0.035 | 28 | 1.00 | 0.01 | 0.023 |
| herbaceous | Lamiaceae | 2 | 1.24 | 0.19 | 0.015 | 1 | 0.72 | 0.08 | 0.031 | 0 | | | | 3 | 1.06 | 0.13 | 0.019 |
| herbaceous | Asteraceae | 4 | 3.27 | 0.19 | 0.029 | 3 | 0.59 | 0.13 | 0.025 | 8 | 0.22 | 0.02 | 0.032 | 15 | 1.11 | 0.06 | 0.029 |
| herbaceous | *Sambucus nigra*-type | 0 | | | | 0 | | | | 1 | 1.30 | 0.12 | 0.013 | 1 | 1.30 | 0.12 | 0.013 |
| herbaceous | Cyperaceae | 3 | 3.37 | 0.13 | 0.029 | 2 | 0.98 | 0.03 | 0.031 | 6 | 0.56 | 0.02 | 0.035 | 11 | 1.40 | 0.04 | 0.030 |
| herbaceous | *Rumex* | 0 | | | | 2 | 2.79 | 0.17 | 0.014 | 3 | 0.58 | 0.03 | 0.018 | 5 | 1.46 | 0.07 | 0.015 |
| herbaceous | Liliaceae | 1 | 1.49 | 0.11 | 0.014 | 0 | | | | 0 | | | | 1 | 1.49 | 0.11 | 0.014 |
| herbaceous | Amaryllidaceae | 1 | 1.64 | 0.09 | 0.013 | 0 | | | | 0 | | | | 1 | 1.64 | 0.09 | 0.013 |
| herbaceous | Cornaceae | 0 | | | | 1 | 1.72 | 0.14 | 0.044 | 0 | | | | 1 | 1.72 | 0.14 | 0.044 |
| herbaceous | Apiaceae | 0 | | | | 0 | | | | 3 | 2.13 | 0.41 | 0.042 | 3 | 2.13 | 0.41 | 0.042 |
| herbaceous | Campanulaceae | 0 | | | | 1 | 2.29 | 0.14 | 0.022 | 0 | | | | 1 | 2.29 | 0.14 | 0.022 |
| herbaceous | Cerealia | 0 | | | | 0 | | | | 4 | 2.36 | 0.42 | 0.069 | 4 | 2.36 | 0.42 | 0.069 |
| herbaceous | Ranunculaceae | 1 | 7.86 | 2.65 | 0.007 | 3 | 1.95 | 0.10 | 0.015 | 5 | 0.99 | 0.12 | 0.014 | 9 | 2.56 | 0.54 | 0.013 |
| herbaceous | Plantaginaceae | 0 | | | | 1 | 5.96 | 0.31 | 0.019 | 8 | 2.49 | 0.11 | 0.028 | 9 | 2.87 | 0.11 | 0.026 |
| herbaceous | Caryophyllaceae | 2 | 4.08 | 0.10 | 0.026 | 1 | 0.60 | 0.05 | 0.041 | 0 | | | | 3 | 2.92 | 0.07 | 0.032 |
| herbaceous | Chenopodiaceae | 3 | 5.56 | 0.66 | 0.014 | 0 | | | | 1 | 4.28 | 0.27 | 0.019 | 4 | 5.24 | 0.50 | 0.014 |
| herbaceous | *Thalictrum* | 0 | | | | 1 | 4.65 | 0.30 | 0.012 | 0 | | | | 1 | 4.65 | 0.30 | 0.013 |
| herbaceous | *Artemisia* | 6 | 15.07 | 0.38 | 0.010 | 1 | 1.35 | 0.24 | 0.016 | 2 | 4.33 | 1.59 | 0.014 | 9 | 11.16 | 0.44 | 0.012 |
| herbaceous | *Urtica* | 0 | | | | 0 | | | | 1 | 10.52 | 0.31 | 0.007 | 1 | 10.52 | 0.31 | 0.007 |
| herbaceous | Elaeagnaceae | 2 | 13.64 | 0.69 | 0.012 | 0 | | | | 0 | | | | 2 | 13.64 | 0.686 | 0.012 |
| herbaceous | *Humulus* | 1 | 16.43 | 1.00 | 0.010 | 0 | | | | 0 | | | | 1 | 16.43 | 1.000 | 0.010 |
| herbaceous | Amaranthaceae | 1 | 21.35 | 2.34 | 0.010 | 0 | | | | 0 | | | | 1 | 21.35 | 2.340 | 0.010 |
| herbaceous | *Sanguisorba* | 1 | 24.07 | 3.50 | 0.012 | 0 | | | | 1 | 24.07 | 3.500 | 0.012 | 1 | 24.07 | 3.500 | 0.012 |

FIGURES

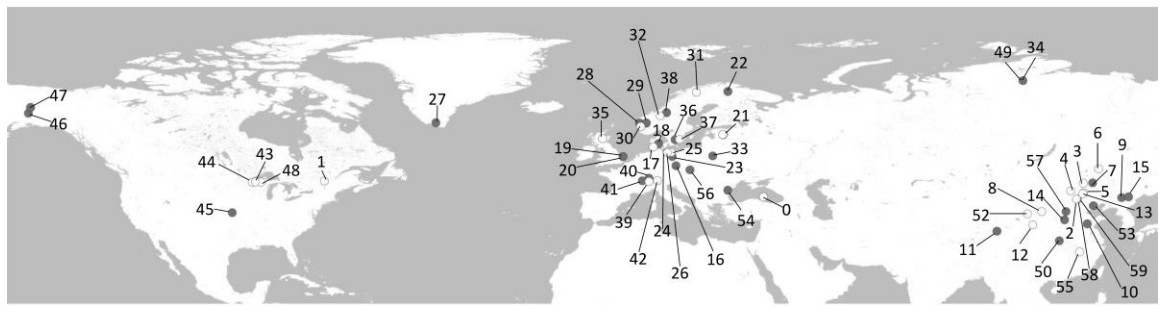

Publication included in composite datasets ○ Publication not included in composite datasets

*0 Filipova-Marinova et al., 2010 (no ERV)*
*1 Chaput & Gajewski 2018*
*2 Ge et al., 2015°*
*3 Han et al., 2017°*
*4 He et al., 2016° (Hulunbeier)*
*5 He et al., 2016° (Xilinhaote)*
*6 He et al., 2016° (Sunitezuoq)*
*7 Li et al. in prep°*
*8 Li et al., 2011*
9 Li et al., 2015
10 Li et al., 2017a
11 Wang and Herzschuh, 2011
*12 Wu et al., 2013°*
*13 Xu et al., 2014*
14 Zhang et al., 2017° (Taiyue)
*15 Zhang et al., 2017° (Changbai)*
16 Abraham and Kozáková, 2012
*17 Andersen, 1967 (no ERV)*
18 Nielsen 2004
19 Bunting et al., 2005 (Calthorpe)

20 Bunting et al., 2005 (Wheatfen)
*21 Poska et al., 2011*
22 Räsänen et al., 2007
23 Matthias et al., 2012
*24 Theuerkauf et al., 2013 (small lake)*
Theuerkauf et al., 2013 (medium lake)
*26 Theuerkauf et al., 2015 (no ERV)*
Bunting et al., 2013
Hjelle, 1998 (Coast)
Hjelle, 1998 (Inland)
*30 Hjelle and Sugita, 2012*
*31 Sjögren, 2013 Dividalen (no ERV)*
*32 Sjögren, 2013 Budalen (no ERV)*
Baker et al., 2016
Niemeyer et al., 2015 (moss)
*35 Twiddle et al., 2012*
Broström et al., 2004
*37 Sugita et al., 1999*
von Stedingk et al., 2008
*39 Sjögren et al., 2008b*

Soepboer et al., 2007
Mazier et al., 2008
*42 Sjögren et al., 2008a*
*43 Calcote, 1995 Sylvania*
*44 Calcote 1995 Wisconsin*
Commerford et al. 2013
Hopla, 2017 (Alaska, Denali)
Hopla 2017 (Alaska, Boreal Forest)
*48 Sugita et al., 2006*
Niemeyer et al., 2015 (lake)
Jiang et al., 2020
*51 Chen et al., 2019✦ (no coordinates)*
*52 Qin et al., 2020*
Zhang et al., 2020
Grindean et al. 2019
*55 Fang et al., 2019*
Kuneš et al., 2019
Li et al., 2017b
Ge et al., 2017 (year 2013 data)
*59 Ge et al., 2017 (year 2014 data)*

° Data from Li et al., 2018
✦ Data from Jiang et al., 2020

**Figure 1: Map of Northern Hemisphere studies on relative pollen productivity estimates. Studies in italics are not included in the continental relative pollen productivity datasets.**



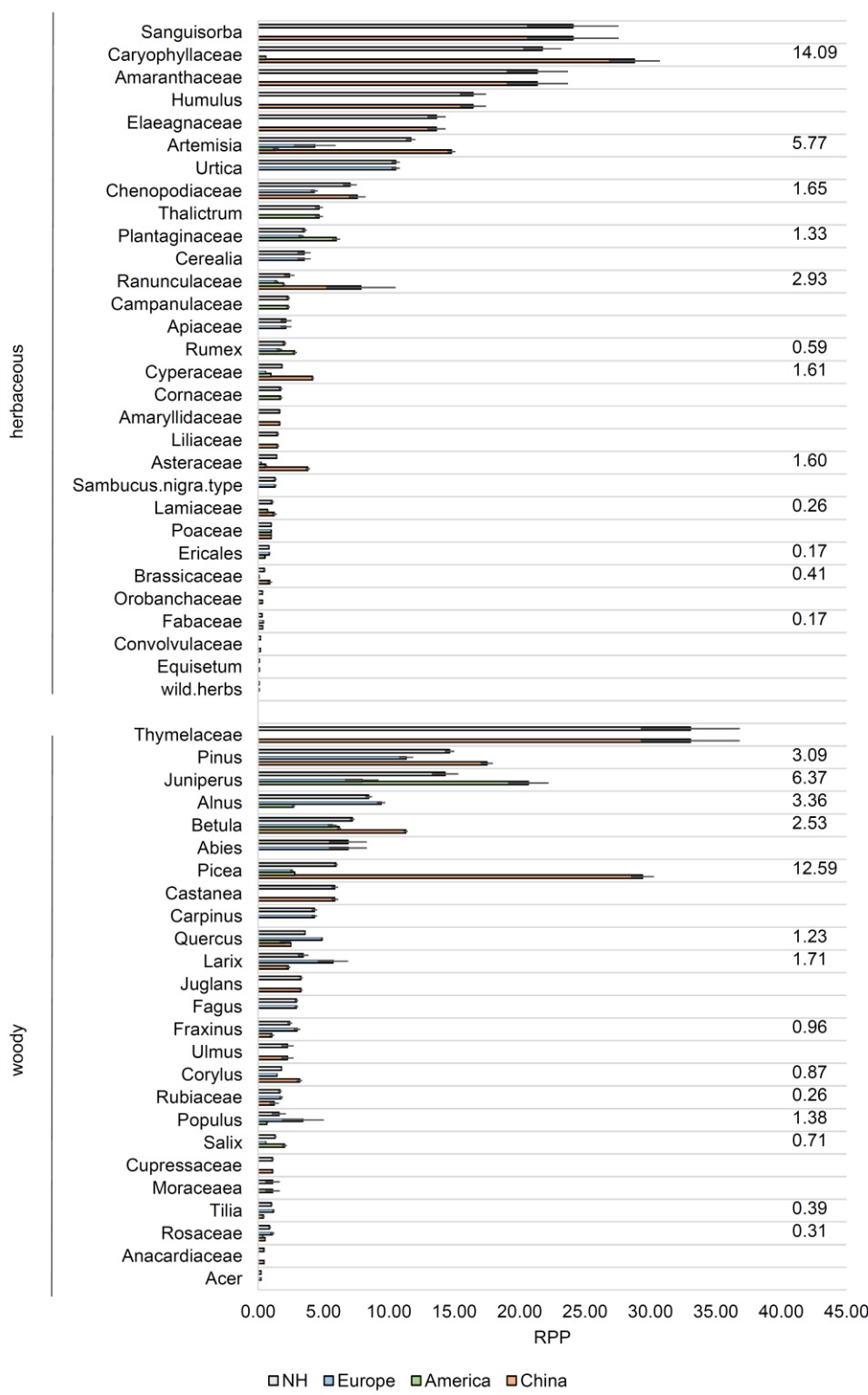


**Figure 2: Relative pollen productivity (RPP) dataset v1 including all continental mean RPP values with their standard error (SE), calculated with the delta method (see methods). Numbers to the right are the standard deviation (SD) between continental datasets, NH is Northern Hemisphere.**


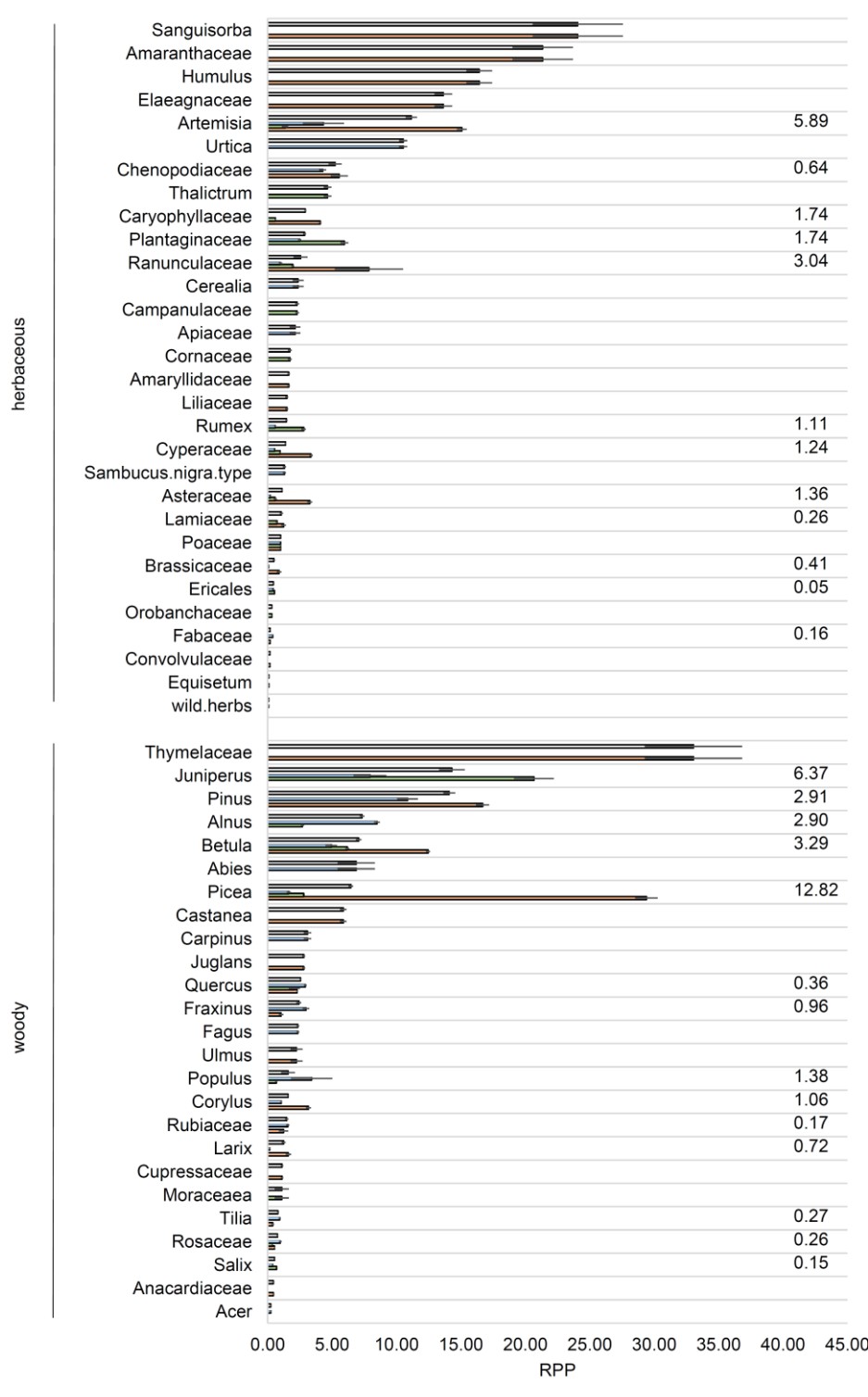

**Figure 3: Relative pollen productivity (RPP) dataset v2 including subsetted continental mean RPP values with their standard error (SE), calculated with the delta method (see methods). Numbers to the right are the standard deviation (SD) between continental datasets, NH is Northern Hemisphere.**


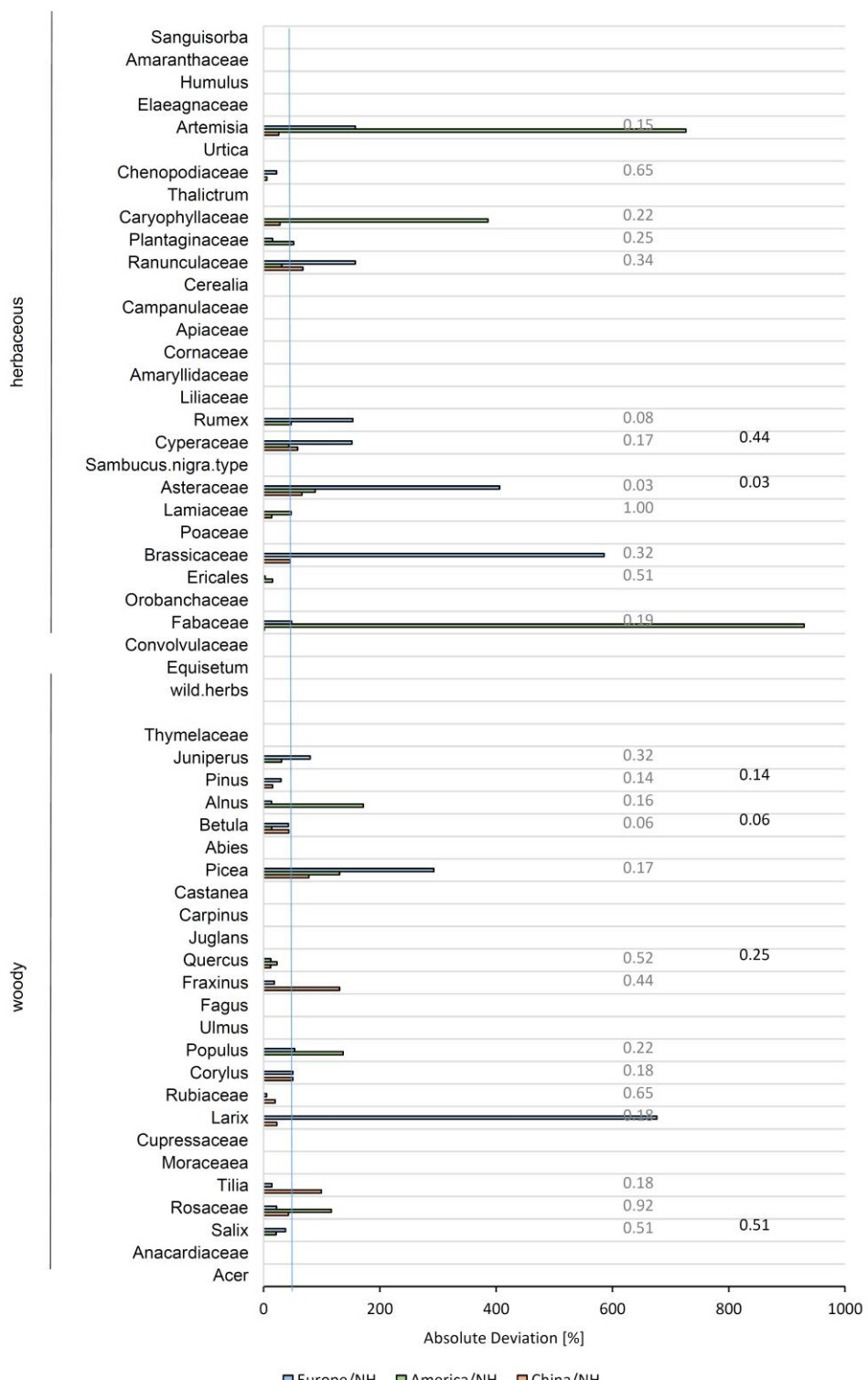

**Figure 4: Absolute percent deviation of the Northern Hemispheric relative pollen productivity (RPP) dataset v2 to each continental RPP dataset. Deviation is calculated by ABS((RPP$_{continent}$ - RPP$_{NH}$)/RPP$_{continent}$)*100. The blue line indicates an absolute deviation of 50%. Numbers on the right are p-values of a Kruskal-Wallis test of each taxon between the three continents. Results shown in grey included each RPP set with data, black coloured values only those with N>2 RPP values in at least two continents.**


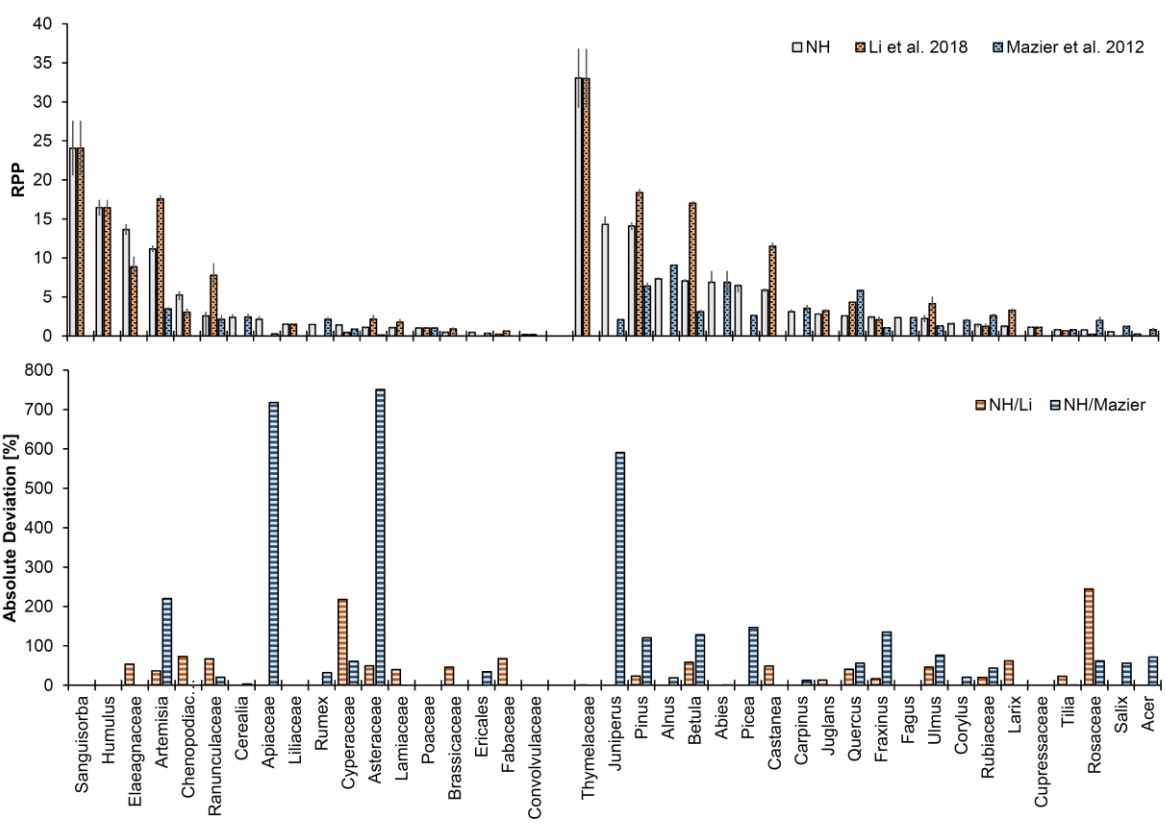


**Figure 5: Relative pollen productivity (RPP) values for selected taxa from different studies (upper panel) and absolute percentage deviation of the RPP Northern Hemispheric (NH) v2 dataset to previously published datasets (lower panel, calculated by ABS((RPP$_{study}$ - RPP$_{NH}$)/RPP$_{study}$)*100). Previously published datasets are the Alt-1 dataset of Li et al. (2018) and PPE.st2 of Mazier et al. (2012).**
