# Peer review of "Compilation of relative pollen productivity estimates and taxonomically harmonised RPP datasets for single continents and Northern Hemisphere extratropics"

_Earth System Science Data, 2019_

## Referee Comment (RC1) · Marie-Jose Gaillard (Referee) · 31 Mar 2020

General Comment

This compilation is useful as a summary of the studies that have attempted to estimate pollen productivity of plant taxa with the aim to apply models of pollen-vegetation relationships such as REVEALS and LOVE (Sugita et al. 2007a and b, The Holocene). However, the way in which the values of relative pollen productivity (RPP, earlier abbreviated PPEs) found in these publications have been handled is neither adequate

nor useful. My major concern with this synthesis is that it handle RPPs as they were measurements, but they are estimates calculated using a model (ERV model) that has assumptions. When these assumptions are badly met in the studies, the results do not make sense. Unfortunately several RPP studies were published although they are theoretically not sound. This is due to the fact that few palynologists understand the theory of the ERV model, its sub-models (1,2,3), and the likelihood method used to estimate RPPs. Therefore reviewers did not notice that these studies present values of RPP that are not correct.

The compilation of M. Wieczorek and U. Herzschuh does not take into account earlier expert evaluations of the RPP estimates (e.g. Mazier et al., 2014 for Europe; Li et al., 2018 for China). It therefore disregards careful evaluations that were meant to help palynologists in their choice of values to be applied. It implies that the database includes a mix of reliable and unreliable values of pollen productivity. The database also excludes reliable values for plant taxa that have been harmonized into higher taxonomic groups and, therefore, mixed with values representing a larger number of species or genus and often different ones. Moreover, the different ERV sub-models used are not explained, and the reason why the results from all ERV sub-models are not included in the database is not provided. In the excel file that can be uploaded in PANGAEA, the column "model" includes a mix of information on the ERV model submodels (1, 2, 3) and on models of dispersion and deposition (e.g. Langrarian model), although these are two difference things. There should be one column for the ERV sub-model chosen by the authors of the original publication (1, 2 or 3), one column for the dispersion and deposition model (Gaussian Plume Model or Lagrangian model), and a third column for the vegetation distance weighting model (1/d or Prentice's model (bog)).

In my opinion, a database should either include all RPP values published (i.e. values obtained with all ERV sub-models and all distance-weighting methods used by the authors) OR a selection of RPP values based on a proper expert evaluation. The database as it stands is neither nor. If it is a database including all values, the user

should be referred to former evaluations and warned on not using the database as a "black box", but as a source of data for further evaluation and testing of the values. If it is a database including only part of the published values, the reason for including and excluding values need to be argued for on scientific/theoretical grounds. I am of course fully open to new and different evaluations from those by Mazier et al. (2012) and Li et al. (2018). However, these evaluations should be based on expert knowledge, which is not the case in the current compilation submitted to ESSD. The database is now including values from studies that are theoretically not sound because most of the assumptions of the ERV model are not met. For instance, Li et al 2011 and Han et al. 2017 (included in the compilation) are theoretically not sound studies (see evaluation in Li et al., 2018). In contrast, Zhang et al. 2017 (not included in the compilation) is a study performed following the correct standards and that meets the most important assumptions of the ERV model (see evaluation in Li et al., 2018). The same can be said of several European studies that are either included or excluded without relevant reasons. It should be also noted that there is a new synthesis and evaluation of RPPs in Europe soon to be submitted (Gaillard et al., in progress), and a synthesis and evaluation of RPPs in N America-Canada-Alaska (Dawson et al. in progress). These studies will further help palynologists to choose RPP values for applications in those regions.

I do not want to reject this paper, because it would not be constructive. I require instead, for the sake of high quality science, that the revisions I am suggesting be considered with care. In order to be useful for the scientific community, the database should include all published RPP values. The authors then have to warn the users on the importance of choosing their RPP values after thorough evaluation of the RPP studies, considering the aim of their application (reconstruction of local, regional, or continental vegetation cover), and having strong arguments for their choice. One has to remember that most published RPP values have not been tested/validated. There is therefore an enormous need of test/validation studies using various alternatives of RPP values (see validations in Hellman et al., 2018 a and b; Cui et al., 2014; Mazier et al., 2015). For the sake of

such studies, a complete database of RPP values would be most valuable.

There is, however, one part of the paper that cannot be accepted: the "harmonized RPP dataset" of means of RPPs throughout the northern hemisphere. There are some limits to mixing the RPP values over geographical space!. When there are strong reasons to think that differences between obtained RPPs are due to differences in species represented in different continents (such as for Artemisia and Pinus in China versus in Europe/N America), it does not make sense to use a mean of those values in all parts of the Northern hemisphere. If we have used a mean of values within NW Europe (Trondman et al., 2015) and within temperate China (Li et al., 2020), it was motivated by the fact that there were too little RPP values to demonstrate that the difference between RPPs within Europe (or within China) could be explained by climate and/or vegetation composition alone. However, it is clear that the RPP values of e.g. Pinus and Artemisia are generally higher in temperate China than in NW Europe (based on the theoretically soundest studies). Therefore, it is not appropriate to mix them.

In summary, I request the following major revisions:

1. Delete the "harmonized RPP dataset" with mean RPP values for the entire N Hemisphere, it does not make sense, as long as it is not tested/validated to produce realistic REVEALS reconstructions of land cover. The danger is to let people believe that this RPP dataset is the best possible for the N Hemisphere and can be used right away for the best possible results. This neither true nor tested.

2. Add to the database all RPP values available in all RPP studies that are not included so far.

3. Indicate in the database the studies and RPP values that were evaluated as not reliable due to theoretical problems (often assumptions of ERV model not met, and/or strange behaviour of log likelihood values while estimating RPP values) in earlier expert evaluations.

4. Warn the user: the database should not be used as a "black box". Values should be selected carefully based on sound arguments.

Detailed comments

Please read comments made directly in the pdf file and implement the required revisions.

I am open to discuss the issues I explain above directly with the authors in case something is unclear or seems not correct in the view of the authors.

Marie-José Gaillard, March 21st 2020

Please also note the supplement to this comment:
https://www.earth-syst-sci-data-discuss.net/essd-2019-242/essd-2019-242-RC1-supplement.pdf

---

## Referee Comment (RC2) · Anonymous Referee #2 · 2 Apr 2020

Synthesis of relative pollen productivity estimates (RPP) is useful to achieve pollen-based quantitative reconstructions of plant cover for the purpose of palaeo-environmental and -climate studies taking plant cover into consideration. RPP is one of the most important parameters in the models of quantitative vegetation reconstruction (e.g. REVEALS and LOVE model, Sugita, 2007a and b). The reliability of RPP determines the reliability of the vegetation reconstruction. Therefore, it is important to check the theory and methodology behind each original publication to include only reliable pollen productivity before calculating a mean of such values. My major concern of this

study is that it does not take into consideration of earlier evaluation of RPPs (Mazier et al., 2012 for Europe; Li et al., 2018 for temperate China), which is not good for the reliability of future quantitative reconstruction if the unreliable RPP values are used. My second concern is, so far there is no test about whether the RPPs of one continent are reliable for application in quantitative reconstruction of another continent available so far, so it is better to handle them separately. With the reasons mentioned above, I would recommend a major revision.

I suggest following revising strategy:

1.List all available relative pollen productivity estimates, indicate the ones that evaluated by experts or tested for reliability in the original publication.

2.Check the reliability of each study through following steps:

1) There are several assumptions behind the ERV model, the reliability of the RPP values depend on whether the assumptions of the ERV model in the study are meet, check each study and keep the ones meet the assumptions.

2) In theory, log-likelihood will increase as the distance from sampling site increases and gradually reach an asymptote at the distance of relevant source area of pollen (RSAP). Check and keep only the studies with theoretically correct log-likelihood against distance curve.

3) Check the SE and RPP, retain the ones that SE<RPP

4) The RPPs from different continents can be very different mainly due to different plant species involved for same pollen type. Test of the reliability of sharing the RPPs among continents with observations (e.g. Hellman et al., 2008, Journal of Quaternary Science) or historical vegetation maps (Cui et al., 2014, Ecology and Evolution) is very important, but will be very time consuming and difficult to collect such data, therefore no such tests available so far. It is therefore important to prepare the RPP dataset of each continent separately for this study.

5) Calculate the mean of the retain values from above and do box plot of the PPE-means by excluding values defined as values outside the range of $\pm 1.5$ interquartile-range for each continent separately.

3. It is important to warn the readers the importance of using only the reliable RPPs.

4. In the dataset file, please do not mix the ERV sub-models and dispersion functions, they are totaly different things. Please indicate the distance weighting method of each study.

To achieve the goal of a more constructive and useful dataset of this synthesis for future application in quantitative reconstruction, I would recommend a second review of revised version by experts in ERV model and quantitative vegetation reconstruction models.

---

## Author Comment (AC1) · 20 May 2020

Author Response

We thank both reviewers for their helpful and constructive suggestions. From our answers you can see that we are ready to revise the dataset and adjust the manuscript.

Anonymous Referee

1.1 Synthesis of relative pollen productivity estimates (RPP) is useful to achieve pollen-

based quantitative reconstructions of plant cover for the purpose of palaeoenvironmental and -climate studies taking plant cover into consideration. RPP is one of the most important parameters in the models of quantitative vegetation reconstruction (e.g. REVEALS and LOVE model, Sugita, 2007a and b). The reliability of RPP determines the reliability of the vegetation reconstruction. Therefore, it is important to check the theory and methodology behind each original publication to include only reliable pollen productivity before calculating a mean of such values. My major concern of this study is that it does not take into consideration of earlier evaluation of RPPs (Mazier et al., 2012 for Europe; Li et al., 2018 for temperate China), which is not good for the reliability of future quantitative reconstruction if the unreliable RPP values are used. My second concern is, so far there is no test about whether the RPPs of one continent are reliable for application in quantitative reconstruction of another continent available so far, so it is better to handle them separately. With the reasons mentioned above, I would recommend a major revision.

1.2 We thank the anonymous referee for the critical and very constructive review, giving precise advises to create a better dataset. When preparing a revised manuscript we would be happy to acknowledge you by name.

——

2.1 I suggest following revising strategy: 1. List all available relative pollen productivity estimates, indicate the ones that evaluated by experts or tested for reliability in the original publication.

2.2 We gratefully thank the anonymous referee for this suggestion. In the process of revision, we will follow this advice, by compiling a dataset with all available RPP-values, indicating within the dataset whether a study/RPP value was evaluated by the original authors or by other experts (especially taking into account evaluations of Li et al. 2018 and Mazier et al. 2012).

——

3.1 2. Check the reliability of each study through following steps: 2.1 There are several assumptions behind the ERV model, the reliability of the RPP values depend on whether the assumptions of the ERV model in the study are meet, check each study and keep the ones meet the assumptions.

3.2 For a revised version, we will collect the main assumptions (increasing log-likelihood + SE>RPP) and indicate for each study, whether they are met.

———

4.1 2.2 In theory, log-likelihood will increase as the distance from sampling site increases and gradually reach an asymptote at the distance of relevant source area of pollen (RSAP). Check and keep only the studies with theoretically correct log-likelihood against distance curve.

4.2 We will follow this advice.

———

5.1 2.3 Check the SE and RPP, retain the ones that SE<RPP

5.2 As described in lines 56, we retained only those which meet this condition ("Afterwards, all PPEs with SD>PPE and non-plausible PPEs >50 were excluded from the dataset").

———

6.1 2.4 The RPPs from different continents can be very different mainly due to different plant species involved for same pollen type. Test of the reliability of sharing the RPPs among continents with observations (e.g. Hellman et al., 2008, Journal of Quaternary Science) or historical vegetation maps (Cui et al., 2014, Ecology and Evolution) is very important, but will be very time consuming and difficult to collect such data, therefore no such tests available so far. It is therefore important to prepare the RPP dataset of each continent separately for this study.

6.2 We acknowledge your concerns regarding the taxonomically harmonised RPP dataset over all continents. We will conduct further statistical analyses on the variability of RPP values within and between continents. We plan to present the RPP continent-wise, depending to the analyses results, we will present in addition also the hemisphere-wide results. We consider the presented averaged RPP values not as a tool for site-specific coverage reconstruction but rather as a tool for data transformation to be applied to large-scale pollen data sets.

——

7.1 2.5 Calculate the mean of the retain values from above and do box plot of the PPE-means by excluding values defined as values outside the range of $\pm 1.5$ interquartile range for each continent separately

7.2 Please see our answer 6.2.

——

8.1 3. It is important to warn the readers the importance of using only the reliable RPPs

8.2. Please see our answer 2.2 and 3.2. We will indicate the reliability of the RPP values and will warn readers to only use these, as you suggest.

——

9.1 4. In the dataset file, please do not mix the ERV sub-models and dispersion functions, they are totaly different things. Please indicate the distance weighting method of each study.

9.2 When going again through each study, we will follow your recommendation.

——

10.1 To achieve the goal of a more constructive and useful dataset of this synthesis for future application in quantitative reconstruction, I would recommend a second review

of revised version by experts in ERV model and quantitative vegetation reconstruction models.

10.2 We hope we will get the chance to upload a completely revised dataset and manuscript and will be happy to receive further recommendations.

Marie-Jose Gaillard (Referee)

General comments

1.1 This compilation is useful as a summary of the studies that have attempted to estimate pollen productivity of plant taxa with the aim to apply models of pollen-vegetation relationships such as REVEALS and LOVE (Sugita et al. 2007a and b, The Holocene). However, the way in which the values of relative pollen productivity (RPP, earlier abbreviated PPEs) found in these publications have been handled is neither adequate nor useful. My major concern with this synthesis is that it handle RPPs as they were measurements, but they are estimates calculated using a model (ERV model) that has assumptions. When these assumptions are badly met in the studies, the results do not make sense. Unfortunately several RPP studies were published although they are theoretically not sound. This is due to the fact that few palynologists understand the theory of the ERV model, its sub-models (1,2,3), and the likelihood method used to estimate RPPs. Therefore reviewers did not notice that these studies present values of RPP that are not correct. The compilation of M. Wieczorek and U. Herzschuh does not take into account earlier expert evaluations of the RPP estimates (e.g. Mazier et al., 2014 for Europe; Li et al., 2018 for China). It therefore disregards careful evaluations that were meant to help palynologists in their choice of values to be applied. It implies that the database includes a mix of reliable and unreliable values of pollen productivity. The database also excludes reliable values for plant taxa that have been harmonized into higher taxonomic groups and, therefore, mixed with values representing a larger number of species or genus and often different ones. Moreover, the different ERV sub-models used are not explained, and the reason why the results from all ERV submodels are not included in the database is not provided. In the excel file that can be uploaded in PANGAEA, the column "model" includes a mix of information on the ERV model submodels (1, 2, 3) and on models of dispersion and deposition (e.g. Langrarian model), although these are two difference things. There should be one column for the ERV sub-model chosen by the authors of the original publication (1, 2 or 3), one column for the dispersion and deposition model (Gaussian Plume Model or Lagrangian model), and a third column for the vegetation distance weighting model (1/d or Prentice's model (bog)). In my opinion, a database should either include all RPP values published (i.e. values obtained with all ERV sub-models and all distance-weighting methods used by the authors) OR a selection of RPP values based on a proper expert evaluation. The database as it stands is neither nor. If it is a database including all values, the user should be referred to former evaluations and warned on not using the database as a "black box", but as a source of data for further evaluation and testing of the values. If it is a database including only part of the published values, the reason for including and excluding values need to be argued for on scientific/theoretical grounds. I am of course fully open to new and different evaluations from those by Mazier et al. (2012) and Li et al. (2018). However, these evaluations should be based on expert knowledge, which is not the case in the current compilation submitted to ESSD. The database is now including values from studies that are theoretically not sound because most of the assumptions of the ERV model are not met. For instance, Li et al 2011 and Han et al. 2017 (included in the compilation) are theoretically not sound studies (see evaluation in Li et al., 2018). In contrast, Zhang et al. 2017 (not included in the compilation) is a study performed following the correct standards and that meets the most important assumptions of the ERV model (see evaluation in Li et al., 2018). The same can be said of several European studies that are either included or excluded without relevant reasons. It should be also noted that there is a new synthesis and evaluation of RPPs in Europe soon to be submitted (Gaillard et al., in progress), and a synthesis and evaluation of RPPs in N America-Canada-Alaska (Dawson et al. in progress). These studies will further help palynologists to choose RPP values for applications in those

regions. I do not want to reject this paper, because it would not be constructive. I require instead, for the sake of high quality science, that the revisions I am suggesting be considered with care. In order to be useful for the scientific community, the database should include all published RPP values. The authors then have to warn the users on the importance of choosing their RPP values after thorough evaluation of the RPP studies, considering the aim of their application (reconstruction of local, regional, or continental vegetation cover), and having strong arguments for their choice. One has to remember that most published RPP values have not been tested/validated. There is therefore an enormous need of test/validation studies using various alternatives of RPP values (see validations in Hellman et al., 2018 a and b; Cui et al., 2014; Mazier et al., 2015). For the sake of such studies, a complete database of RPP values would be most valuable.

1.2 Thank you very much for your honest and detailed revision of our dataset and manuscript. As described in our answers to the Anonymous Referee, we will compile a dataset containing all available RPP values. We will indicate which studies/values have been evaluated by the original authors or by other experts (especially taking into account evaluations of Li et al. 2018 and Mazier et al. 2012). We will furthermore collect the main assumptions of the ERV model (increasing log-likelihood + SE>RPP) and indicate, whether they are met.

——

2.1 There is, however, one part of the paper that cannot be accepted: the "harmonized RPP dataset" of means of RPPs throughout the northern hemisphere. There are some limits to mixing the RPP values over geographical space!. When there are strong reasons to think that differences between obtained RPPs are due to differences in comment species represented in different continents (such as for Artemisia and Pinus in China versus in Europe/N America), it does not make sense to use a mean of those values in all parts of the Northern hemisphere. If we have used a mean of values within NW Europe (Trondman et al., 2015) and within temperate China (Li et al., 2020),

it was motivated by the fact that there were too little RPP values to demonstrate that the difference between RPPs within Europe (or within China) could be explained by climate and/or vegetation composition alone. However, it is clear that the RPP values of e.g. Pinus and Artemisia are generally higher in temperate China than in NW Europe (based on the theoretically soundest studies). Therefore, it is not appropriate to mix them.

2.2 We acknowledge your concerns regarding the harmonised RPP dataset of means of RPPs throughout the northern hemisphere. Please also see our answer 6.2 to the anonymous referee. For a revised version, we will conduct tests on the variability of (reliable) RPPs within and between continents. Depending on the results, we will re-consider to calculate mean RPPs per continent or for the entire Northern Hemisphere.

——

3.1 In summary, I request the following major revisions: 1. Delete the "harmonized RPP dataset" with mean RPP values for the entire N Hemisphere, it does not make sense, as long as it is not tested/validated to produce realistic REVEALS reconstructions of land cover. The danger is to let people believe that this RPP dataset is the best possible for the N Hemisphere and can be used right away for the best possible results. This neither true nor tested.

3.2 Please see our answer above (2.2): Depending on the results of variation within and between continents, we will decide on how to proceed with the harmonised RPP dataset. In either way, we will warn the reader to not use the dataset without further own assessments of the suitability of the dataset for the respective study.

——

4.1 2. Add to the database all RPP values available in all RPP studies that are not included so far.

4.2 We will follow this advice and provide such a table, to give a most complete

overview on all data available.

———

5.1 3. Indicate in the database the studies and RPP values that were evaluated as not reliable due to theoretical problems (often assumptions of ERV model not met, and/or strange behaviour of log likelihood values while estimating RPP values) in earlier expert evaluations.

5.2 Please see our answer to your comment 1.1: In a revised version, we will compile and present all RPP values. We will indicate which studies/values have been evaluated and whether the main assumptions of the ERV model are met.

———

6.1 4. Warn the user: the database should not be used as a "black box". Values should be selected carefully based on sound arguments.

6.2 We will follow this recommendation.

———

Specific comments

7.1 Line 10: "Pollen productivity estimates (PPEs)" Please use the more useful term Relative Pollen Productivity (RPP) estimates (or RPPs) that is now more commonly used than PPEs in order to avoid any misunderstanding

7.2 Thank you for this advice, which we will follow in our revision.

———

8.1 Line 14-15: This compilation allows scientists to identify the best PPE for their own studies and to identify data-gaps in need of further PPE analyses. This sounds good; but HOW will scientists identify the "best" PPEs??? How will they evaluate all these values?? See my more detailed separate comments about this

8.2 In the revised version, we will include all RPP-values of the original studies. The - to the best of our knowledge - most complete overview of studies and their data can help scientists to identify RPP studies in their region, which can be applied to their data. We will point out in more detail, that before using these data, the original publications need to be read in detail for final decisions.

——

9.1 Line 28-19: , pollen productivity estimates (PPE) and their fall speeds have been calculated for various regions and taxa revise sentence; ".... relative pollen productivity (RPP) have been estimated and fall speed of pollen (FSP) measured or calculated for major plant taxa in several regions of the world.

9.2 We will follow this suggestion.

——

10.1 Line 32: Mazier 2008 Mazier 2012

10.2 Thank you, will be changed accordingly.

——

11.1 Line 34-36: an easy-to-apply unified PPE dataset for Northern Hemispheric pollen would help to reduce the bias of pollen dispersal and pollen productivities in vegetation reconstructions using broad-scale pollen datasets by adopting a consistent approach. It sounds nice, but I do not see how you can get a single RPP dataset for the entire NHemisphere given that there are obvious differences in RPP for plant taxa (such as Pinus, Artemisia, Chenopodiaceae) between e.g. Europe and China. Using a mean value for such taxa does not make sense; see my more detailed separate comment

11.2 Please see our explanation above (reply 2.2 and 3.2 to your review and reply 6.2 to the anonymous referee). The taxonomically harmonised, northern hemispheric dataset should mainly be applied to pollen datasets over large scale regions. Available

studies on pollen productivity are available in a rather clustered format in some regions of the world, but largely missing in others. We will check for pollen-variability within a continent and between continents, to finally decide whether a northern hemispheric dataset will be kept or if three datasets (Europe, Asia, North America) will be presented.

——

12.1 Line 37: PPE and fall speed Reword: "... a unified dataset of RPP estimates and FSPs for major Northern Hemispheric plant taxa.". Note however that I reject the concept of a single RPP and FSP dataset for entire NH

12.2 We respect your position concerning the single dataset, but believe that there are areas of application where it is relevant. These areas of application comprise, as described above, mainly large scale applications to big data sets. Please also see our other replies (e.g. 11.2).

——

13.1 Line 44: profound overview reword: ... to gain the most complete overview possible of ....

13.2 Will be changed accordingly.

——

14.1 Line 46: correction factor Don't use this term. A correction factor is another concept and is known in the literature mainly from the correction factors of S.T. Andersen. These were used to roughly correct pollen %. They were not proper RPP (PPE), and were not used together with models

14.1 Thank you for clarifying this. We will reword the manuscript based on your remark.

——

15.1 Line 47: provide such fractionate correction factors What is meant here?

15.2 This is meant for example for publications investigating pollen productivity but providing e.g. count data instead of RPPs. Will be rephrased to: 15.3 Publications which did not provide RPPs or consisted only of compilations of previously available PPE data were excluded from all further analyses.

–––

16.1 Line 50: The review of Li et al. (2018) does not contain newly calculated PPE It is a shame that you ignore the careful work made in Mazier et al. 2012 and Li et al. 2018 to select the most reliable RPP values on the basis of the method used and thorough evaluation of the values. RPP studies using the ERV model are not straight forward to conduct, and some of the values published are not reliable at all because the authors have made theoretical errors in their use of the ERV model. Moreover, most of the RPP published have not been tested/validated

16.2 Please see our above replies to your review (1.1 and 5.2).

–––

17.1 Line 51: which of we incorporated Reword: "....from which we selected only those published in Chinese."

17.2 Will be changed accordingly.

–––

18.1 Line 54: In a first step, all PPE values and, if given, their standard deviation (SD) Does this mean that you included RPP values that did not have SDs? Why? Do you think it is a good idea to use RPP without SDs? It will provide an error on the REVEALS or LOVE reconstructions that will be misleading.

18.2 We will largely remove RPPs without SD. However, we consider e.g. Larix, of which the only available RPP based on Poaceae is given without SD, as an indispensable taxon. In the revised version we will point to this problematic taxa and indicate the

knowledge gap.

——

19.1 Line 56-57: a reasonable taxonomic harmonisation It might be reasonable in terms of plant taxanomy, but it might not be reasonable in terms of RPPs, i.e. you might mix species with totally different RPP values, and species that do not exist together on a particular continent.

19.2 As stated above, we will test for the variability within and between continents to decide whether we keep the up with our concept or only present different RPPs for each continent. As well, we consider the presented RPPs more as a tool for pollen-data transformation rather than for site-specific quantitative vegetation cover reconstruction. We will better indicate this in the improved version of the manuscript.

——

20.1 Line 61: we confined our analysis to publications with Poaceae as the reference taxon. such publications according to what you are saying below! My initial comment was: Why? It is possible to convert RPP relative to other taxa (Pinus or Quercus for instance) into RPP relative to Poaceae, at least if the authors have calculated a RPP for Poaceae as well. It would be useful to indicate what studies you excluded on this basis.

20.2 The information of which studies we excluded is given in Figure 1 and in lines 74 to 77 of the manuscript.

——

21.1 Line 62: boxplots of PPE per taxon were calculated with the help of R (version 3.5.3, R Core Team, 2019). Please motivate/argue for the choice of this particular method.

21.2 Boxplots calculate the quartiles of data. Defining values outside the of $\pm 1.5$ *

interquartile-range (range between the 0.25 and 0.75 quartile) is a common approach for outlier identification.

‾‾

22.1 Line 64-65: excluding outliers (defined as values outside the range of $\pm 1.5$ * interquartile-range). Could you please motivate/argue for the choice of this criteria to define outliers

22.2 Please see our explanation above.

‾‾

23.1 Line 65: delta method (Stuart and Ord, 1994). Add: (...., 1994; see Li et al., 2020 for details on the method).

23.2 We will add this reference.

‾‾

24.1 Line 65-68: Subsequently, we looked at those taxa for which PPEs are available but do not have Poaceae as the reference taxon. These comprise Eleaganaceae, Nitraria, Tsuga, wild herbs and PoaceaeCrop. For studies that included Poaceae in the analysis set, we set Poaceae as 1 and recalculated the other PPEs based on that ratio This comes too late! See my comment above. From your text above it seems you excluded those studies with other taxa than Poaceae as reference taxon.... Restructure the paragraph! - Other comment: "wild herbs" does not help us so much if we do not know what they include!

24.2 Wild herbs will be checked again for included taxa. The paragraph will be restructured as follows: 24.3 Line 59 and following: Some publications did not use Poaceae as the reference taxon: while it is possible to recalculate values relative to Poaceae (cf. Li et al., 2018; Mazier et al., 2012), we confined our analysis to publications with Poaceae as the reference taxon. However, some taxa are not available with Poaceae

as reference – for these, recalculations were conducted.

———

25.1 Line 67: PoaceaeCrop I guess this is "Cerealia type"? Why not call it cereals?

25.2 Will be changed accordingly.

———

26.1 Line 68: (applies to…) What do you mean? "following (or according to) Sugita et al....."?

26.2 This means that the recalculation based on available Poaceae RPP could only be applied to this study. Will be changed to: 26.3 For studies that included Poaceae in the analysis set, we set Poaceae as 1 and recalculated the other RPPs based on that ratio (was the case for RPPs of Sugita et al., 1999).

———

27.1 Line 69-72: For all other studies (Calcote, 1995; Chaput and Gajewski, 2018; Li et al., 2015; Matthias et al., 2012; Theuerkauf et al., 2015), we recalculated the PPEs based on the original reference taxon. If, for example, Acer was used as the reference taxon, we assumed the Poaceae-to-Acer PPE to be the same as our calculated mean PPE for Acer and recalculated the other values based on that ratio Same comment as above. And, add a reference for this way to recalculate RPPs (I think Mazier et al., 2012 was one of the first using this approach).

27.2 We are not sure which of the above comments is meant. The reference to this way of recalculation is given in line 60/61 but will be added here as well.

———

28.1 Line 74-75: Publications are not considered if neither Poaceae was used as a reference nor species included for which no PPE would be available without recalculation

Whar do you mean? This sentence does not make sense! Rewrite!

28.2 This means, that studies are not included if (i) they do not have Poaceae as reference taxon and (ii) include only taxa of which we have RPP values based on Poaceae from other studies and will be rephrased (28.3). However, the whole paragraph might change after new analysis and evaluation of ERV in the different studies (see replies 1.2 to your review and 3.2 to the anonymous referee). 28.3 Nine studies (Andersen, 1967; Bunting et al., 2005; He et al., 2016; Li et al., 2015; Sjögren et al., 2008b; Theuerkauf et al., 2013; Twiddle et al., 2012; Wu et al., 2013; Zhang et al., 2017 (Changbai Mountains)) are excluded from the as they did not use Poaceae as reference taxon and include only taxa of which RPP values based on Poaceae are available in studies

——

29.1 Line 68-92: Changbai site from Zhang et al. (2017) were excluded from calculations because Poaceae was not the reference taxon. Eight studies did not have Poaceae as the reference, but did include PPE values for taxa which would otherwise not be represented in the final PPE-dataset (applies to Calcote, 1995; Filipova-Marinova et al., 2010; Li et al., 2011; Matthias et al., 2012; Sugita et al., 1999, 2006; Theuerkauf et al., 2015; Zhang et al., 2017 (site Taiyue)). These taxa are Aesculus, Elaeagnaceae, Nitraria, PoaceaeCrop, Pterocrya, Tsuga and wild.herbs (Table 3). The final dataset consists of PPE values for 58 taxa and fall speeds for 57 taxa, with 54 taxa having both PPE values and fall speeds available (Table 3). This is already said in Methods. I would recommend to have this only in Methods and to explain this better. It is very confusing as it is written now. AS far as I understand you have 3 categories of studies with reference taxa different than Poaceae: a) reference taxon different than Poaceae, but RPP for Poaceae available: conversion is simple; b) reference taxon different than Poaceae, and RPP for Poaceae not available: conversion requires assumption "we recalculated the PPEs based on the original reference taxon. If, for example, Acer was used as the reference taxon, we assumed the Poaceae-to-Acer PPE to be the same as our calculated mean PPE for Acer and recalculated the other

values based on that ratio"; c) reference taxon different than Poaceae, and xxxxxx (I don't understand what is characteristic for this third group): conversion is not possible if we want to use the same assumption/approach as for b). Once these 3 groups are well defined, make a Table with the studies and taxa that are related to those 3 groups.

29.2 We have the following four criteria, which we will move into the Methods for better description: 1) Poaceae is the reference taxon 2) another species is the reference, but a value for Poaceae available → recalculation for those taxa, of which we do not have an RPP-value with Poaceae as original reference taxon 3) another species is the reference and no value for Poaceae available → recalculation for those taxa, of which we do not have an RPP-value with Poaceae as original reference taxon 4) only values already recalculated to Poaceae as reference taxon available → no own recalculation, but inclusion only of those taxa of which we do not have an RPP-value with Poaceae as original reference taxon.

——

30.1 Line 98: as well as subtropical regions are largely lacking Did you include the subtropics in your search of literature? Seems to me that some studies are lacking in that case. There are at least studies in press. If you want to include the subtropics in this study/paper you should tell what is in progress as well.

30.2 We conducted an open-ended search (see search terms in lines 42-44), which resulted in the literature given in Table 1.

——

31.1 Line 100: can be Has been

31.2 Will be changed accordingly.

——

32.1 Line 101: broad scale what do you mean? A large number? Or values from

different parts of the world, or both? Clarify!

32.2 Large number. Will be changed accordingly.

——

31.1 Line 105: show Shows

31.2 Will be changed accordingly.

——

32.1 Line 105-106: While Cyperaceae or Fraxinus Alnus, Quercus and Chenopodi-aceae would be better examples. Low values have a tendency to "look" more similar than large values, but you should think about what a RPP is used for. A RPP of 2 is very different than a RPP of 1, even if it does not look to be very different. It means that one of the value is double the other. A RPP of 10 seems much different than a RPP of 20, but in fact it's the same difference, one is double the other. The difference it will imply in the application of these values in reconstructions will be the same in both cases.

33.1 We will reconsider our examples based on your remarks.

——

33.1 Line 106: similar range Do you mean? similar values?

33.2 Yes.

——

34.1 Line 106-107: strongly vary between the publications In fact it is mainly a large difference between Europe and China for Artemisia, Betula and Pinus, as discussed in Li et al. 2018

34.2 As described above, we will compare in detail the variability within and between continents and present the results and implications here.

——

35.1 Line 115-119: such as vegetation characteristics, and some uncertainty due to the use of inconsistent reference taxa. Most studies used Poaceae, a widespread family, whose pollen is easy to identify and often preserved in a good state. However, as discussed by Broström et al. (2008), the pollen cannot be identified to species level and different studies may thus have used different species of Poaceae for the reference. Other taxa such as Quercus or Acer are therefore sometimes used as the reference taxon These explanations are the general ones that have been discussed in all syntheses so far. However, for Artemisia and Pinus in particular, Li et al. (2018) have discussed the differences within China with more specific arguments. Please revise accordingly.

35.2 We will provide more examples including Li et al (2018) how previews authors dealt with this problem.

——

36.1 Line 124-125: This open access dataset can be used to improve our understanding of past vegetation dynamics for a broad range of taxa rather than interpreting pollen counts or percentages alone. This is perhaps my most serious concern about this paper. If the dataset made open access in PANGAEA is used as a "black box" in will really NOT improve our understanding of past vegetation cover. It may provide results that are nonsense. Please read my detailed separate comment about this, and revise accordingly.

36.2 We assumed with our revised manuscript including all the detailed suggestions from both reviewers our data set does not come as a "black box". Please also read our replies above. We will reanalyse the data to decide on how to proceed with the taxonomically harmonised dataset. Furthermore, we will warn the reader to not use the dataset without further own assessments on the suitability for the planned application.

37.1 Line 127: The compilation is useful for the identification of available PPE sets at specific sites and regions. Yes. But I don't think it adds so terribly much to the syntheses of Li et al. 2018 and Mazier et al. (2012). There are moreover syntheses in progress for northern America (Dawson et al., personal communication) and a new one for Europe (Gaillard et al.).

37.2 Thank you very much for this information. However, we cannot take a position on these syntheses as they have not yet been published. We acknowledge the work, Mazier et al. and Li et al. put in their compilations. However, new studies have been published since then and furthermore, both do not include the studies in Northern Asian and North America. We consider our study mainly as providing a basic and most complete overview on data access of RPP studies and are convinced, that a complete compilation is of value for the scientific community.

38.1 Line 129-130: The unified PPE dataset of Northern Hemispheric extratropical taxa allows a consistent approach to be applied to synthesised pollen data at a continental to hemispherical scale Not the best way to do it, the values need to be evaluated!

39.2 As described above we will add some evaluation in particular with respect to regional differences.

40.1 Line 132-133: and PPEs from a nearby local study would not necessarily be best suited for their interpretation. Please develop what you mean here. It is true that one can think of using RPPs from particular modern climate zones for time periods of the past with similar climate conditions at other geographical locations than today. However, doing so imply that we first can demonstrate that differences in RPP between studies are indeed due to climatic parameters. It is certainly useful to have all these

values in a database. BUT, all scientists that will use them will HAVE TO go back to the original publications to evaluate the values. Anybody can do alternative syntheses to those already published by Mazier et al 2012 or Li et al 2018. But what is MOST needed to day is VALIDATION of these values.

40.2 An in-depth assessment is out of the scope of our manuscript, which focuses on the provision of (to the best of our knowledge) all available RPP studies in the northern hemisphere. We will however provide further information on whether the data were evaluated by the original authors or other experts, with main focus on Mazier et al 2012 and Li et al 2018.

——

41.1 Line 146-147: This study is a contribution to the Past Global Changes (PAGES) LandCover6k working group project. It won't be accepted as a contribution to Land-Cover6k if relevant revisions are not implemented. This study has not been discussed with any of the responsible coordinators of PAGES LandCover6k. PAGES LandCover6k has a certain number of quality requirements for the studies that are performed as contributions. We can of course use different methods and have other ideas in terms of interpretation of results. But we can't accept scientific/theoretical mistakes as it is the case here.

41.2 We hope that an overview of all available RPP data that comes along with a thoroughly revised manuscript according to your and the anonymous reviewer constructive suggestion will be considered as an contribution to PAGES LandCover6k. We also specifically submitted this data to ESSD, as we know that the entire community can join the revision of data and manuscript.

——

42.1 Table 1: Broström et al., 2008, Mazier et al., 2012 Would be useful to know from the table that this is a review
42.2 The information will be added.

–––

43.1 Table 2: Taxon Are these thought as pollen morphological taxa? In that case, specify! It is not the same as plant taxa

43.2 Yes, these are pollen morphological taxa. The information will be added.

–––

44.1 Table 2: Original Taxa Here again you should specify that these are the pollen-morphological taxa for which RPP were estimated in the literature; for some of these taxa it is not relevant to have lost the information on RPP for particular species or genus in the case of such a synthesis. To group pollen-morphological types is sometimes needed if those taxa were not identified in the fossil pollen records used for reconstruction. But in other cases these RPPs can be used. I am thinking of e.g. Compositae SF Cichoriodae, Aster T. /Anthemis T, Calluna vulgaris, Pinus cembra, Secale cerealia, Trollius europaeus, Filipendula, Potentilla type, different Plantago species, P. lanceolata in particular, etc.

44.2 The grouping presented in Table 2 is particularly important for the harmonised data set. Since we do not think that the northern hemispheric RPP-dataset will be applied for site-specific, but for large, taxonomically harmonised datasets, we regard the groups as useful taxonomic units. We will however check again our groups with regard to the proposed taxa.

–––

45.1 Table 2: Aster-Anthemis type italic and write Aster/Anthemis type

45.2 Will be changed accordingly.

–––

46.1 Table 2: Leucanthemum vulgate vulgare?

46.2 Yes, will be changed accordingly.

——

47.1 Table 2: Robinia Sophora write Robinia/Sophora

47.2 Will be changed accordingly.

——

48.1 Table 2: Cercis italic!

48.2 Will be changed accordingly.

——

49.1 Table 2: Lamiaceae + Mentha type Thymus + Thymus praecox + Vitex negundo very different pollen-morphological type; can't be harmonized in this way

49.2 Please see our answer 44.2 – we will reconsider the grouping of pollen-morphological types.

——

50.1 Table 2 : Mentha type Thymus + Thymus +

50.2 Will be changed to Mentha type (Thymus).

——

51.1 Table 2: PoaceaeCrop In the table of mean RPPs you have also cereals, what is the difference between PoaceaeCrop and Cereals?

51.2 PoaceaeCrop and Cerealia will be combined.

——

52.1 Table 3: Have genus names in italic

52.2 Will be changed accordingly.

——

53.1 Table 3: PoaceaeCrop Are these wild Poaceae that are cultivated? Or Cerealia type? Isn't it a "synonym" for "Cerealia"? What is this taxon good for?

53.2 PoaceaeCrop and Cerealia will be combined.

——

54.1 Table 3: wild herbs What wild herbs? What is this taxon good for if we do not know what plant taxa it includes?

54.2 Wild herbs will be checked again for included taxa.

——

55.1 Figure 1: What means the asterix? What is (from Gaillard 1994)? What publication is it? Not in the list of publication. And why? What is wrong with Sugita et al. 1999 as reference in this case?

55.2 We are not sure which asterisk is meant. Some references have a ° indicating that they are extracted from Li et al. 2018, as stated in the figure caption. We used Sugita et al. 1999, who stated to have their data from Gaillard et al. 1994 – the reference will be added to the list of publications.

——

56.1 Figure 2: which replace by "that"

56.2 Will be changed accordingly.

——

57.1 Figure 3: values Add SEs for all values!!!!

57.2 Will be added.

——

58.1 Figure 3: species Taxa

58.2 Will be changed accordingly.

——

59.1 Figure 3: upper panel Would be good to also add the grey mean in the upper panel

59.2 Will be added.

——

60.1 Figure 3: lower panel Specify that the grey bars are acoording to "this paper, mean values"!

60.2 We will indicate, that the values in the lower panel are means.

——

61.1 Figure 3: showing similar values for some and a high variability for other taxa. delete; not relevant in a Figure caption

61.2 Will be removed.

---

## Author Response (AR1)

**Author Response**

We thank both reviewers for their helpful and constructive suggestions. We substantially revised the manuscript and the data to meet the required updates.

**Anonymous Referee**

*1.1    Synthesis of relative pollen productivity estimates (RPP) is useful to achieve pollen-based quantitative reconstructions of plant cover for the purpose of palaeoenvironmental and - climate studies taking plant cover into consideration. RPP is one of the most important parameters in the models of quantitative vegetation reconstruction (e.g. REVEALS and LOVE model, Sugita, 2007a and b). The reliability of RPP determines the reliability of the vegetation reconstruction. Therefore, it is important to check the theory and methodology behind each original publication to include only reliable pollen productivity before calculating a mean of such values. My major concern of this study is that it does not take into consideration of earlier evaluation of RPPs (Mazier et al., 2012 for Europe; Li et al., 2018 for temperate China), which is not good for the reliability of future quantitative reconstruction if the unreliable RPP values are used. My second concern is, so far there is no test about whether the RPPs of one continent are reliable for application in quantitative reconstruction of another continent available so far, so it is better to handle them separately. With the reasons mentioned above, I would recommend a major revision.*

1.2    RESPONSE:
       We thank the referee for their critical and very constructive review, giving good advice as to how to create a better dataset. When preparing a revised manuscript we would be happy to acknowledge you by name.
* * *
*2.1    I suggest following revising strategy:*
       *1. List all available relative pollen productivity estimates, indicate the ones that evaluated by experts or tested for reliability in the original publication.*

2.2    RESPONSE:
       We are grateful to the referee for this suggestion. We revised the compilation including all published RPPs (Table A1). In the new metadata table (Table A3) we indicate if a model was chosen by authors and/or experts as best fit.
* * *
*3.1    2. Check the reliability of each study through following steps:*
       *2.1 There are several assumptions behind the ERV model, the reliability of the RPP values depend on whether the assumptions of the ERV model in the study are meet, check each study and keep the ones meet the assumptions.*

3.2    RESPONSE:
       We collected the main assumptions (e.g. increasing log-likelihood, SE > RPP, vegetation sampling > RSAP) and indicate for each study and model, whether they are met (Table A3)
* * *
*4.1   2.2  In theory, log-likelihood will increase as the distance from sampling site increases and gradually reach an asymptote at the distance of relevant source area of pollen (RSAP). Check and keep only the studies with theoretically correct log-likelihood against distance curve.*

4.2   RESPONSE:
We followed this advice and only kept correct maximum likelihood curves for our final combined RPP datasets. However, if a study had been evaluated and used by experts before, we relied on their decision. Because so few studies are available from the American continent, we relaxed our criteria in this respect for this continent. All information is now available in the tables (Table A1, A2, A3) so users of the data can easily create their own customised dataset applying their own selection criteria.

4.3   TEXT (lines 165-196):
The likelihood function score should decrease and approach an asymptote when reaching the RSAP (see methods). Within the sampled vegetation area, the curve does not approach an asymptote in the studies of Calcote (1995) and Chaput & Gajewski (2018), meaning that vegetation composition is not studied up to the RSAP. As furthermore Poaceae was not used as the referenced taxon, we decided to not use these data despite the scarcity of studies in northern America. In the studies of Han et al. (2017) and Xu et al. (2014), the likelihood function score increases. We followed the assessment of Li et al. (2018) and did not incorporate these RPPs. The likelihood function score further increases in the study of Ge et al. (2017, year 2014 data). Data from He et al., (2016) are not used in accordance with Li et al. (2018), as pollen are sampled from a pollen trap, which might behave differently compared to moss pollsters or lakes. In the study of Hjelle and Sugita (2012), the likelihood function score does not approach an asymptote. Sugita et al. (1999, 2006) do not provide information on the likelihood and RPP values are given without information on standard deviation or standard error. The studies of Twiddle et al. (2012) and Li et al. (2011) do not provide standard deviations or errors for the presented RPP values. The study of Wu et al. (2013, original publication in Chinese) was rejected by Li et al. (2018) because of a too large sampling area and we followed this assessment. Theuerkauf et al. (2013) does not provide information on the maximum likelihood or the RSAP. Data from Chen et al. (2019) were extracted from Jiang et al. (2020) but included insufficient information on the study design and the ERV-approach. Data from the study of Qin et al. (2020) have been rejected has they had very high values for most taxa compared to other studies, which we assume was a systematic problem of the study. The study of Fang et al. (2019) was excluded because it was designed to test different methods for RPP estimation and was carried out in patchy vegetation without enough sites.

On the other hand, some studies were incorporated despite missing information or likelihood curves that did not meet our criteria:

Hjelle (1998) and Nielsen (2004) do not provide information on the likelihood but have been included in the dataset of Mazier et al. (2012, i.e. was assessed by an expert). Bunting et al. (2013) do not provide information on the likelihood nor do they sample vegetation up to the value of RSAP. The scarcity of data from northern America together with Poaceaea as a reference taxon led us to the decision to keep these RPPs. While the likelihood function score should decrease and reach an asymptote at the radius of the RSAP, the log-likelihood should

increase before reaching the asymptote. This is not the case for the study of Commerford et al. (2013), but data have been included due to scarcity of American studies. At the boreal forest site of Hopla (2017), the likelihood function score does not reach an asymptote. Again, these data have been included due to the scarcity of American studies.

5.1    *2.3  Check the SE and RPP, retain the ones that SE<RPP*

5.2    RESPONSE:
We retained only those which meet this criterion.

5.3    TEXT (line 106):
Dataset v1 includes all values of the chosen studies, except those RPPs which have an SD (or SE) > RPP
* * *
6.1    *2.4 The RPPs from different continents can be very different mainly due to different plant species involved for same pollen type. Test of the reliability of sharing the RPPs among continents with observations (e.g. Hellman et al., 2008, Journal of Quaternary Science) or historical vegetation maps (Cui et al., 2014, Ecology and Evolution) is very important, but will be very time consuming and difficult to collect such data, therefore no such tests available so far. It is therefore important to prepare the RPP dataset of each continent separately for this study.*

6.2    RESPONSE:
We acknowledge your concerns regarding the taxonomically harmonised RPP dataset over all continents. We conducted further statistical analyses on the variability of RPP values within and between continents. We present the RPP continent-wise and in addition hemisphere-wide results. We consider the presented averaged RPP values not as a tool for site-specific coverage reconstruction but rather as a tool for data transformation to be applied to broad-scale pollen datasets. Kruskal-Wallis tests on the differences of RPP values between the continents show that a significant difference between continents can only be found for Asteraceae.

6.3    TEXT:
Lines 139-141
We conducted Kruskal-Wallis tests on the dataset v2 between the continents for each taxon. Additionally, we conducted the tests on the variability between taxa, once for the Northern Hemisphere and separately for each continent, including only taxa with n>2.

Lines 210-225
Testing the RPP values used to create the combined dataset on the variability between taxa shows that the taxa themselves are significantly different from each other (**Northern Hemisphere**: Kruskal-Wallis chi-squared = 99.337, df = 29, p <0.001 with *Acer*, *Alnus*, Apiaceae, *Artemisia*, Asteraceae, *Betula*, *Carpinus*, Caryophyllaceae, Cerealia, Chenopodiaceae, *Corylus*, Cyperaceae, Ericales, Fabaceae, *Fagus*, *Fraxinus*, *Juglans*, Lamiaceae, *Larix*, *Picea*, *Pinus*, Plataginaceae, *Populus*, *Quercus*, Ranunculaceae, Rosaceae, Rubiaceae, *Rumex*, *Salix*, *Tilia*; **China**: Kruskal-Wallis chi-squared = 27.599, df = 9, p <0.01, with *Artemisia*, Asteraceae, *Betula*, Chenopodiaceae, Cyperaceae, Fabaceae, *Juglans*, *Larix*, *Pinus*, *Quercus*; **Europe**: Kruskal-Wallis

chi-squared = 56.5, df = 21, p <0.001, with *Acer*, *Alnus*, Apiaceae, Asteraceae, *Betula*, *Carpinus*, Cerealia, *Corylus*, Cyperaceae, Ericales, *Fagus*, *Fraxinus*, *Picea*, *Pinus*, Plataginaceae, *Quercus*, Ranunculaceae, Rosaceae, Rubiaceae, *Rumex*, *Salix*, *Tilia*; **America**: Kruskal-Wallis chi-squared = 6.7091, df = 2, p <0.05, with *Asteraceae*, *Betula*, *Salix*). Furthermore, while some taxa strongly differ between continents when looking at the absolute deviation (e.g. *Artemisia*, Fabaceae or *Larix*) others show no large deviation from the overall Northern Hemispheric mean (e.g. *Salix*, *Betula*; Figure 4). And while we found overall significant differences between taxa (described above), we did not find significant differences between datasets for single taxa (n=6) from two continents when applying the Kruskal-Wallis test, except for Asteraceae (Figure 4). This means the differences between continents are rather small compared to differences between taxa.
* * *
**7.1** *2.5 Calculate the mean of the retain values from above and do box plot of the PPEmeans by excluding values defined as values outside the range of ±1.5 interquartile range for each continent separately*

7.2 RESPONSE:
We revised our analyses and skipped the boxplots. Instead, we partly followed the methods of Li et al. (2018) and Mazier et al. (2012) in the creation of combined RPP datasets. We did that for each continent separately and used the resultant data to calculate the Northern Hemisphere extratropics dataset.

7.3 TEXT:
Lines 106-122:
In the choice of reliable values, we mainly followed the strategy of Mazier et al. (2012) and Li et al. (2018).
Dataset v1 includes all values of the chosen studies, except those RPPs which have an SD (or SE) > RPP.
Dataset v2 is further reduced with the following steps:
- If N≥5, the highest and smallest RPPs are excluded
- If N=4, the most deviating value from the Taxa-specific mean is excluded. Exception: if two values are from the same study (they are generally similar), their mean is calculated and used for the overall mean (→ *Salix* in America; *Betula*, Fabaceae and *Larix* in China; *Rumex* in Europe). The most deviating value is chosen based on the resulting mean. Exception in America: *Betula* with 4 values from only two studies are all kept.
- If N=3, a value is only excluded if it is strongly deviating (>100% of the mean of all values) → Caryophyllaceae of Li et al., in prep in China. Exceptions: in America Asteraceae and in Europe Apiaceae with three values from only two studies are all kept, as the two similar ones came from the same study.
- If N=2, all values are kept, except if one seems less reliable (*Larix*, Matthias et al. 2012)

Dataset v2 was created separately for each continent and is comparable to the Alt-1 dataset of Li et al. (2018) and PPE.st2 of Mazier et al. (2012).

Lines 134-137:

The majority of RPP studies concentrates on China and Europe, with one study from Arctic Russia and few studies from northern America. We thus decided to create a Northern Hemispheric dataset to be applied only for broad-scale studies for which otherwise RPP data for various taxa would be lacking. The dataset for the whole Northern Hemisphere was calculated with all data of the continental datasets.
* * *
8.1    *3. It is important to warn the readers the importance of using only the reliable RPPs*

8.2.   RESPONSE:
       We indicated the reliability of the RPP values (Table A2, A3) and warn readers to only use these, as you suggest.

8.3    TEXT
       lines 269-273:
       While one has to keep in mind the limited amount of data influencing the statistical power, we conclude that there is no particular reason to not set up a Northern Hemispheric RPP dataset. Still, before applying one of the datasets presented, researchers should consult the original publication to be sure it fits their needs and standards and be aware of the rather problematic use of SD and SE, which might have influenced our presented SEs.

       Lines 279-287:
       The continental datasets can be applied to assess vegetation changes using broad-scale pollen datasets. It is important to keep in mind that different taxa with different pollen productivities and dispersal abilities are combined in one RPP value and the application to such broad-scale datasets can only be an approximation. This is especially important for the Northern Hemispheric dataset, which should not be applied to calculate site-specific vegetation compositions. This dataset fills data gaps of RPP values in various regions, but at the cost of accuracy. We consider the presented averaged RPP values as a tool for data transformation to be applied to broad-scale pollen datasets. Using the dataset in this way can account for differences in pollen productivities and transportation rather than obtaining fully reliable quantitative information about the vegetation cover around a specific site.
* * *
9.1    *4. In the dataset file, please do not mix the ERV sub-models and dispersion functions, they are totally different things. Please indicate the distance weighting method of each study.*

9.2    RESPONSE:
       We followed your recommendation and indicate for each study which model and which distance weighting were applied (e.g. Tables A1, A3). While we present RPPs estimated from various models, we only include those estimated with the ERV in our continental datasets (see Methods).

9.3    TEXT:
       Lines 151-152

The compilation of RPP studies includes data from 49 studies, 43 of them using a form of the ERV-model (Tables A1, A2, A3).

Lines 161-164
Of 60 RPP-datasets, 28 (coming from 23 studies) were excluded prior to the calculation of the combined RPP datasets.
Filipova-Marinova et al. (2010), Andersen (1967), Theuerkauf et al., (2015), Sjögren (2013), and Sjögren et al. (2008a, 2008b) do not present RPP-values based on ERV-models.
* * *
10.1   *To achieve the goal of a more constructive and useful dataset of this synthesis for future application in quantitative reconstruction, I would recommend a second review of revised version by experts in ERV model and quantitative vegetation reconstruction models.*

10.2   We will be happy to receive further recommendations.

**Marie-Jose Gaillard (Referee)**

General comments

1.1   *This compilation is useful as a summary of the studies that have attempted to estimate pollen productivity of plant taxa with the aim to apply models of pollen-vegetation relationships such as REVEALS and LOVE (Sugita et al. 2007a and b, The Holocene). However, the way in which the values of relative pollen productivity (RPP, earlier abbreviated PPEs) found in these publications have been handled is neither adequate nor useful. My major concern with this synthesis is that it handle RPPs as they were measurements, but they are estimates calculated using a model (ERV model) that has assumptions. When these assumptions are badly met in the studies, the results do not make sense. Unfortunately several RPP studies were published although they are theoretically not sound. This is due to the fact that few palynologists understand the theory of the ERV model, its sub-models (1,2,3), and the likelihood method used to estimate RPPs. Therefore reviewers did not notice that these studies present values of RPP that are not correct. The compilation of M. Wieczorek and U. Herzschuh does not take into account earlier expert evaluations of the RPP estimates (e.g. Mazier et al., 2014 for Europe; Li et al., 2018 for China). It therefore disregards careful evaluations that were meant to help palynologists in their choice of values to be applied. It implies that the database includes a mix of reliable and unreliable values of pollen productivity. The database also excludes reliable values for plant taxa that have been harmonized into higher taxonomic groups and, therefore, mixed with values representing a larger number of species or genus and often different ones. Moreover, the different ERV sub-models used are not explained, and the reason why the results from all ERV sub-models are not included in the database is not provided. In the excel file that can be uploaded in PANGAEA, the column "model" includes a mix of information on the ERV model submodels (1, 2, 3) and on models of dispersion and deposition (e.g. Langrarian model), although these are two difference things. There should be one column for the ERV sub-model chosen by the authors of the original publication (1, 2 or 3), one column for the dispersion and*

*deposition model (Gaussian Plume Model or Lagrangian model), and a third column for the vegetation distance weighting model (1/d or Prentice's model (bog)).*

*In my opinion, a database should either include all RPP values published (i.e. values obtained with all ERV sub-models and all distance-weighting methods used by the authors) OR a selection of RPP values based on a proper expert evaluation. The database as it stands is neither nor. If it is a database including all values, the user should be referred to former evaluations and warned on not using the database as a "black box", but as a source of data for further evaluation and testing of the values. If it is a database including only part of the published values, the reason for including and excluding values need to be argued for on scientific/theoretical grounds. I am of course fully open to new and different evaluations from those by Mazier et al. (2012) and Li et al. (2018). However, these evaluations should be based on expert knowledge, which is not the case in the current compilation submitted to ESSD. The database is now including values from studies that are theoretically not sound because most of the assumptions of the ERV model are not met. For instance, Li et al 2011 and Han et al. 2017 (included in the compilation) are theoretically not sound studies (see evaluation in Li et al., 2018). In contrast, Zhang et al. 2017 (not included in the compilation) is a study performed following the correct standards and that meets the most important assumptions of the ERV model (see evaluation in Li et al., 2018). The same can be said of several European studies that are either included or excluded without relevant reasons. It should be also noted that there is a new synthesis and evaluation of RPPs in Europe soon to be submitted (Gaillard et al., in progress), and a synthesis and evaluation of RPPs in N America-Canada-Alaska (Dawson et al. in progress). These studies will further help palynologists to choose RPP values for applications in those regions.*

*I do not want to reject this paper, because it would not be constructive. I require instead, for the sake of high quality science, that the revisions I am suggesting be considered with care. In order to be useful for the scientific community, the database should include all published RPP values. The authors then have to warn the users on the importance of choosing their RPP values after thorough evaluation of the RPP studies, considering the aim of their application (reconstruction of local, regional, or continental vegetation cover), and having strong arguments for their choice. One has to remember that most published RPP values have not been tested/validated. There is therefore an enormous need of test/validation studies using various alternatives of RPP values (see validations in Hellman et al., 2018 a and b; Cui et al., 2014; Mazier et al., 2015). For the sake of such studies, a complete database of RPP values would be most valuable.*

1.2    RESPONSE:

Thank you very much for your honest and detailed revision of our dataset and manuscript. We now present a substantially revised RPP compilation and manuscript. As described in our answers to the Referee 1, we compiled a dataset containing all available RPP values of the studies presented. We indicate which studies have been evaluated by the original authors or by other experts (especially taking into account evaluations of Li et al. 2018 and Mazier et al. 2012). We furthermore collect the main requirements for the ERV model (e.g. likelihood curves reaching an asymptote, + SE>RPP) and indicate, whether they are met.

1.3    TEXT (lines 55-77)

All RPP values and, if given, their standard deviation (SD) or standard error (SE) were collected from the literature. If the data were only presented as figures, values were extracted with the help of Corel Draw X6. The studies of Ge et al. (2015), He et al. (2016), Li et al. (in prep), Wu et

al. (2013) and Zhang et al. (2017) are only available in Chinese and RPP values where extracted from Li et al. (2018), while the study of Chen et al. (2019) was extracted from Jiang et al. (2020).

While different approaches exist to estimate RPP, the extended R-value (ERV) is the most common approach. Details on the ERV model and related assessment criteria can be found in, for example, Abraham and Kozáková (2012), Bunting et al. (2013) and Li et al. (2018). The maximum likelihood method (decreasing likelihood function score or increasing log-likelihood with distance) can be used to identify the relevant source area of pollen (RSAP) and should reach an asymptote with increasing sampling distance (Sugita 1994). For reliable results, the vegetation sampling area should be ≥ RSAP (Sugita 1994). Unexpected behaviour of the maximum likelihood method can occur if assumptions of the ERV-model are not met (Li et al. 2018). Furthermore, a sufficient number of randomly selected sites (no of sites ≥ number of taxa for RPP-estimation) is necessary (Li et a. 2018). Last but not least, for the correct application of the REVEALS model, RPPs need to have a standard deviation provided, to allow for correct estimation of the vegetation cover.

To allow for further assessment of the presented RPP data, we collected information on, for example, the maximum likelihood, the vegetation sampling radius, and the site distribution used in the different studies. (Table A2). This will help researchers when creating customised RPP datasets. If RPP estimates for several models (e.g. ERV-submodel 1, 2 or 3) were presented in the original study, we used all of them for the RPP compilation and added the information on which one was chosen as best fit by the original author and/or in the RPP-compilations of Mazier et al. (2012) and Li et al. (2018) (Tables A1, A3).
* * *
2.1    *There is, however, one part of the paper that cannot be accepted: the "harmonized RPP dataset" of means of RPPs throughout the northern hemisphere. There are some limits to mixing the RPP values over geographical space!. When there are strong reasons to think that differences between obtained RPPs are due to differences in comment species represented in different continents (such as for Artemisia and Pinus in China versus in Europe/N America), it does not make sense to use a mean of those values in all parts of the Northern hemisphere. If we have used a mean of values within NW Europe (Trondman et al., 2015) and within temperate China (Li et al., 2020), it was motivated by the fact that there were too little RPP values to demonstrate that the difference between RPPs within Europe (or within China) could be explained by climate and/or vegetation composition alone. However, it is clear that the RPP values of e.g. Pinus and Artemisia are generally higher in temperate China than in NW Europe (based on the theoretically soundest studies). Therefore, it is not appropriate to mix them.*

2.2    RESPONSE:
We acknowledge your concerns regarding the harmonised RPP dataset of means of RPPs throughout the Northern Hemisphere. Please also see our answer 6.2 to referee 1. We conducted statistical tests on the variability of reliable RPPs within and between continents. While we found overall significant differences between taxa, we did not find significant differences between datasets for single taxa (n=6) from two continents when applying the Kruskal-Wallis test, except for Asteraceae. This means the differences between continents are rather small compared to differences between taxa. This implies, when aiming to compare

vegetation change between continents, that transformation of pollen data using an RPP from another continent is better than keeping the data untransformed. So we agree, that when possible, only data from the same continent should be used, but if not possible (i.e. because a taxon is not yet covered), the values from another continent should be applied. Based on these results we decided to create four different RPP datasets: for America (including Greenland), Europe (including Arctic Russia), China, and one for the entire Northern Hemisphere. However, we explicitly state the limitations of this approach.

2.3    TEXT
Lines 269-273:
While one has to keep in mind the limited amount of data influencing the statistical power, we conclude that there is no particular reason to not set up a Northern Hemispheric RPP dataset. Still, before applying one of the datasets presented, researchers should consult the original publication to be sure it fits their needs and standards and be aware of the rather problematic use of SD and SE, which might have influenced our presented SEs.

Lines 279-287:
The continental datasets can be applied to assess vegetation changes using broad-scale pollen datasets. It is important to keep in mind that different taxa with different pollen productivities and dispersal abilities are combined in one RPP value and the application to such broad-scale datasets can only be an approximation. This is especially important for the Northern Hemispheric dataset, which should not be applied to calculate site-specific vegetation compositions. This dataset fills data gaps of RPP values in various regions, but at the cost of accuracy. We consider the presented averaged RPP values as a tool for data transformation to be applied to broad-scale pollen datasets. Using the dataset in this way can account for differences in pollen productivities and transportation rather than obtaining fully reliable quantitative information about the vegetation cover around a specific site.
* * *
3.1    *In summary, I request the following major revisions:*
*1. Delete the "harmonized RPP dataset" with mean RPP values for the entire N Hemisphere, it does not make sense, as long as it is not tested/validated to produce realistic REVEALS reconstructions of land cover. The danger is to let people believe that this RPP dataset is the best possible for the N Hemisphere and can be used right away for the best possible results. This neither true nor tested.*

3.2    RESPONSE:
Please see our answer above (2). Based on our statistical analyses, we decided to keep the Northern Hemispheric dataset in addition to the three continental ones, but warn the reader to not use the dataset without their own assessments of the suitability of the dataset for the respective study. In particular, we think that these Northern Hemisphere values can be used when no RPP values are available for certain taxa from the region or continent of interest.
* * *
4.1    *2. Add to the database all RPP values available in all RPP studies that are not included so far.*

4.2    RESPONSE:

We followed this advice and provide such a table, to give a thorough overview of all data available (Table A1, A2, A3).
* * *
5.1 *3. Indicate in the database the studies and RPP values that were evaluated as not reliable due to theoretical problems (often assumptions of ERV model not met, and/or strange behaviour of log likelihood values while estimating RPP values) in earlier expert evaluations.*

5.2 RESPONSE:
Please also see our answer to your comment 1: We compiled and present all RPP values available in the presented studies and indicate which studies have been evaluated by experts and whether the main requirements of the ERV model are met (Table A1, A2, A3).

5.3 TEXT (lines 55-77)
All RPP values and, if given, their standard deviation (SD) or standard error (SE) were collected from the literature. If the data were only presented as figures, values were extracted with the help of Corel Draw X6. The studies of Ge et al. (2015), He et al. (2016), Li et al. (in prep), Wu et al. (2013) and Zhang et al. (2017) are only available in Chinese and RPP values where extracted from Li et al. (2018), while the study of Chen et al. (2019) was extracted from Jiang et al. (2020).

While different approaches exist to estimate RPP, the extended R-value (ERV) is the most common approach. Details on the ERV model and related assessment criteria can be found in, for example, Abraham and Kozáková (2012), Bunting et al. (2013) and Li et al. (2018). The maximum likelihood method (decreasing likelihood function score or increasing log-likelihood with distance) can be used to identify the relevant source area of pollen (RSAP) and should reach an asymptote with increasing sampling distance (Sugita 1994). For reliable results, the vegetation sampling area should be ≥ RSAP (Sugita 1994). Unexpected behaviour of the maximum likelihood method can occur if assumptions of the ERV-model are not met (Li et al. 2018). Furthermore, a sufficient number of randomly selected sites (no of sites ≥ number of taxa for RPP-estimation) is necessary (Li et a. 2018). Last but not least, for the correct application of the REVEALS model, RPPs need to have a standard deviation provided, to allow for correct estimation of the vegetation cover.

To allow for further assessment of the presented RPP data, we collected information on, for example, the maximum likelihood, the vegetation sampling radius, and the site distribution used in the different studies. (Table A2). This will help researchers when creating customised RPP datasets. If RPP estimates for several models (e.g. ERV-submodel 1, 2 or 3) were presented in the original study, we used all of them for the RPP compilation and added the information on which one was chosen as best fit by the original author and/or in the RPP-compilations of Mazier et al. (2012) and Li et al. (2018) (Tables A1, A3).
* * *
6.1 *4. Warn the user: the database should not be used as a "black box". Values should be selected carefully based on sound arguments.*

6.2    RESPONSE:
Please also see our answers to your comment 2 and to comment 8 of referee 1. We warn the readers to use the dataset only after careful evaluation of their needs and will add information on PANGAEA to not use it without reading this manuscript.

6.3    TEXT
Lines 266-271:
While one has to keep in mind the limited amount of data influencing the statistical power, we conclude that there is no particular reason to not set up a Northern Hemispheric RPP dataset. Still, before applying one of the datasets presented, researchers should consult the original publication to be sure it fits their needs and standards and be aware of the rather problematic use of SD and SE, which might have influenced our presented SEs.

Lines 277-286:
The continental datasets can be applied to assess vegetation changes using broad-scale pollen datasets. It is important to keep in mind that different taxa with different pollen productivities and dispersal abilities are combined in one RPP value and the application to such broad-scale datasets can only be an approximation. This is especially important for the Northern Hemispheric dataset, which should not be applied to calculate site-specific vegetation compositions. This dataset fills data gaps of RPP values in various regions, but at the cost of accuracy. We consider the presented averaged RPP values as a tool for data transformation to be applied to broad-scale pollen datasets. Using the dataset in this way can account for differences in pollen productivities and transportation rather than obtaining fully reliable quantitative information about the vegetation cover around a specific site.
* * *
Specific comments

7.1    *Line 10: "**Pollen productivity estimates (PPEs)**"*
*Please use the more useful term Relative Pollen Productivity (RPP) estimates (or RPPs) that is now more commonly used than PPEs in order to avoid any misunderstanding*

7.2    Thank you for this advice, which we have followed in our revision.
* * *
8.1    *Line 14-15: **This compilation allows scientists to identify the best PPE for their own studies and to identify data-gaps in need of further PPE analyses.***
*This sounds good; but HOW will scientists identify the "best" PPEs??? How will they evaluate all these values?? See my more detailed separate comments about this*

8.2    RESPONSE:
We included all RPP-values of the original studies (Table A1). The most complete - to the best of our knowledge - overview of studies and their data (Table A2, A3) can help scientists to identify RPP studies in their region, which can be applied to their data. However, we now emphasise that before using these data, the original publications need to be consulted before making a final decision about their use.

8.3    TEXT:
Lines 276-288:
The RPP compilation can be used to get a good overview of existing RPP studies, to identify research gaps and to find RPPs to apply at one's study area. It is important (i) to use only those RPP data which have been evaluated by experts or the author as best fit and (ii) to look at the original publication for further information on how the RPP estimates have been generated. The continental datasets can be applied to assess vegetation changes using broad-scale pollen datasets. It is important to keep in mind that different taxa with different pollen productivities and dispersal abilities are combined in one RPP value and the application to such broad-scale datasets can only be an approximation. This is especially important for the Northern Hemispheric dataset, which should not be applied to calculate site-specific vegetation compositions. This dataset fills data gaps of RPP values in various regions, but at the cost of accuracy. We consider the presented averaged RPP values as a tool for data transformation to be applied to broad-scale pollen datasets. Using the dataset in this way can account for differences in pollen productivities and transportation rather than obtaining fully reliable quantitative information about the vegetation cover around a specific site. -----

9.1    *Line 28-29: **, pollen productivity estimates (PPE) and their fall speeds have been calculated for various regions and taxa***
*revise sentence; ".... relative pollen productivity (RPP) have been estimated and fall speed of pollen (FSP) measured or calculated for major plant taxa in several regions of the world.*

9.2    RESPONSE:
We followed this suggestion.

9.3    TEXT (lines 27-30)
To overcome these problems, relative pollen productivity (RPP) has been estimated and fall speed of pollen (FSP) measured or calculated for major plant taxa in several regions of the world (e.g. Baker et al., 2016; Broström et al., 2004; Commerford et al., 2013; Wang and Herzschuh, 2011).
* * *
10.1    *Line 32: **Mazier 2008***
*Mazier 2012*

10.2    Thank you, has been changed accordingly.
* * *
11.1    *Line 34-36: **an easy-to-apply unified PPE dataset for Northern Hemispheric pollen would help to reduce the bias of pollen dispersal and pollen productivities in vegetation reconstructions using broad-scale pollen datasets by adopting a consistent approach.***
*It sounds nice, but I do not see how you can get a single RPP dataset for the entire NHemisphere given that there are obvious differences in RPP for plant taxa (such as Pinus, Artemisia, Chenopodiaceae) between e.g. Europe and China. Using a mean value for such taxa does not make sense; see my more detailed separate comment*

11.2 RESPONSE:
Please also see our explanation above (reply to comments 2 and 3 of your review and reply to comment 6 of the referee 1). We checked for RPP variability within and between continents and decided to keep a Northern Hemispheric dataset in addition to the three continental datasets. Such data could be used to fill gaps (i.e. if an RPP is not available) and can be used for transformation of hemispheric-wide datasets. However, we agree that it should not be used for studies that focus on specific sites or regions. We state the potential usage of the data now more clearly in the text.

11.3 TEXT:
Lines 276-288:
The RPP compilation can be used to get a good overview of existing RPP studies, to identify research gaps and to find RPPs to apply at one's study area. It is important (i) to use only those RPP data which have been evaluated by experts or the author as best fit and (ii) to look at the original publication for further information on how the RPP estimates have been generated. The continental datasets can be applied to assess vegetation changes using broad-scale pollen datasets. It is important to keep in mind that different taxa with different pollen productivities and dispersal abilities are combined in one RPP value and the application to such broad-scale datasets can only be an approximation. This is especially important for the Northern Hemispheric dataset, which should not be applied to calculate site-specific vegetation compositions. This dataset fills data gaps of RPP values in various regions, but at the cost of accuracy. We consider the presented averaged RPP values as a tool for data transformation to be applied to broad-scale pollen datasets. Using the dataset in this way can account for differences in pollen productivities and transportation rather than obtaining fully reliable quantitative information about the vegetation cover around a specific site.
* * *
12.1 *Line 37: **PPE and fall speed***
*Reword: "... a unified dataset of RPP estimates and FSPs for major Northern Hemispheric plant taxa.". Note however that I reject the concept of a single RPP and FSP dataset for entire NH*

12.2 RESPONSE:
We respect your position concerning the single dataset, but believe that there are areas of application where it is relevant. These areas of application comprise, as described above, mainly broad-scale applications to big datasets. Please also see our other replies (reply to your comments 2, 3 and 11 and to comment 6 of referee 1) explaining why we stick to our concept. The sentence has been rephrased:

12.3 TEXT (lines 41-42)
Here we present a compilation of available RPP-publications, four large-scale datasets of RPP estimates and fall speeds (FSPs) for major Northern Hemispheric plant taxa.
* * *
13.1 *Line 44: **profound overview***
*reword: ... to gain the most complete overview possible of ....*

13.2 RESPONSE:
Changed accordingly (now line 49).
* * *
14.1 *Line 46: **correction factor***
*Don't use this term. A correction factor is another concept and is known in the literature mainly from the correction factors of S.T. Andersen. These were used to roughly correct pollen %. They were not proper RPP (PPE), and were not used together with models*

14.2 RESPONSE:
Thank you for clarifying this. We deleted this passage.
* * *
15.1 *Line 47: **provide such fractionate correction factors***
*What is meant here?*

15.2 RESPONSE:
This is meant for publications investigating pollen productivity but providing, for example, count data instead of RPPs. Rephrased to:

15.3 TEXT (lines 50-52):
Of the resulting 63 publications from our literature search, 12 were excluded a priori (e.g. if they did not provide RPPs or consisted only of compilations of previously available RPP data) and are marked with an x in Table 1.
* * *
16.1 *Line 50: **The review of Li et al. (2018) does not contain newly calculated PPE***
*It is a shame that you ignore the careful work made in Mazier et al. 2012 and Li et al. 2018 to select the most reliable RPP values on the basis of the method used and thorough evaluation of the values. RPP studies using the ERV model are not straight forward to conduct, and some of the values published are not reliable at all because the authors have made theoretical errors in their use of the ERV model. Moreover, most of the RPP published have not been tested/validated*

16.2 RESPONSE:
We are sorry that our statement was misunderstandable and hope that it is now clear that we appreciate the work of Li et al. (2018). Please see our above replies to your comments (1 and 5). We compiled an updated dataset containing all available RPP values of the studies presented. We indicate which studies have been evaluated by the original authors or by other experts (especially taking into account evaluations by Li et al. 2018 and Mazier et al. 2012). We furthermore mention the main requirements for the ERV model (e.g. likelihood curves reaching an asymptote, + SE>RPP) and indicate whether they are met (Table A1, A2, A3).

16.3 TEXT (lines 61-77)

While different approaches exist to estimate RPP, the extended R-value (ERV) is the most common approach. Details on the ERV model and related assessment criteria can be found in, for example, Abraham and Kozáková (2012), Bunting et al. (2013) and Li et al. (2018). The maximum likelihood method (decreasing likelihood function score or increasing log-likelihood with distance) can be used to identify the relevant source area of pollen (RSAP) and should reach an asymptote with increasing sampling distance (Sugita 1994). For reliable results, the vegetation sampling area should be ≥ RSAP (Sugita 1994). Unexpected behaviour of the maximum likelihood method can occur if assumptions of the ERV-model are not met (Li et al. 2018). Furthermore, a sufficient number of randomly selected sites (no of sites ≥ number of taxa for RPP-estimation) is necessary (Li et a. 2018). Last but not least, for the correct application of the REVEALS model, RPPs need to have a standard deviation provided, to allow for correct estimation of the vegetation cover.

To allow for further assessment of the presented RPP data, we collected information on, for example, the maximum likelihood, the vegetation sampling radius, and the site distribution used in the different studies. (Table A2). This will help researchers when creating customised RPP datasets. If RPP estimates for several models (e.g. ERV-submodel 1, 2 or 3) were presented in the original study, we used all of them for the RPP compilation and added the information on which one was chosen as best fit by the original author and/or in the RPP-compilations of Mazier et al. (2012) and Li et al. (2018) (Tables A1, A3).
* * *
17.1 *Line 51:* **which of we incorporated**
*Reword: "....from which we selected only those published in Chinese."*

17.2 RESPONSE:
We rephrased the sentence

17.3 TEXT (lines 55-59):
The studies of Ge et al. (2015), He et al. (2016), Li et al. (in prep), Wu et al. (2013) and Zhang et al. (2017) are only available in Chinese and RPP values where extracted from Li et al. (2018), while the study of Chen et al. (2019) was extracted from Jiang et al. (2020).
* * *
18.1 *Line 54:* **In a first step, all PPE values and, if given, their standard deviation (SD)**
*Does this mean that you included RPP values that did not have SDs? Why? Do you think it is a good idea to use RPP without SDs? It will provide an error on the REVEALS or LOVE reconstructions that will be misleading.*

18.2 RESPONSE:
In the revised dataset we removed RPPs without SD/SE and added a paragraph on the problematic use of SD/SE in some studies (see also our answer to your comment 23).

18.3 TEXT :
Line 82
Furthermore, we only used studies providing standard errors or standard deviations

Lines 124-131

To calculate the SE of averaged RPPs, the delta method (Stuart and Ord, 1994, details in the supplement of Li et al. 2020) was applied. For the calculation of an RPP from pollen counts, a variance-covariance matrix is created. If only RPP ± SD (or SE) are available, the covariance is set to 0 and the final equation results in:

$$SE \sqrt{\frac{= \sum_{i=1}^{n}(var_i)}{(n*n)}}$$

Some problems arise from the labelling of standard errors and standard deviations. While some studies provide standard deviations, others provide standard errors or give no information. Some studies provide standard deviations, which are labelled as standard errors in other studies. Given this ambiguity, we used every value as it is and noted if standard deviation or standard error are said to be given.
* * *
19.1  *Line 56-57:* **a reasonable taxonomic harmonisation**
*It might be reasonable in terms of plant taxanomy, but it might not be reasonable in terms of RPPs, i.e. you might mix species with totally different RPP values, and species that do not exist together on a particular continent.*

19.2  RESPONSE:
We consider the presented continent-wide and Northern Hemisphere-wide RPPs more as a tool for pollen-data transformation and as a base to fill data gaps in regional RPP datasets rather than for site-specific quantitative vegetation cover reconstruction. This is now clearly indicated in the text. We conducted statistical tests on the variability of reliable RPPs within and between continents and while we found overall significant differences between taxa, we found no significant differences between datasets for single taxa (n=6) from two continents when applying the Kruskal-Wallis test, except for Asteraceae. This suggests the differences between continents are rather small compared to differences between taxa, with the implication that, when aiming to compare vegetation change between continents, transformation of pollen data using an RPP from other continent is better than keeping the data untransformed. So we agree, that when possible, only data from the same continent should be used; but if not possible (i.e. because a taxon is not yet covered), values from another continent should be applied.

19.3  TEXT:
Lines 276-288:
The RPP compilation can be used to get a good overview of existing RPP studies, to identify research gaps and to find RPPs to apply at one's study area. It is important (i) to use only those RPP data which have been evaluated by experts or the author as best fit and (ii) to look at the original publication for further information on how the RPP estimates have been generated. The continental datasets can be applied to assess vegetation changes using broad-scale pollen datasets. It is important to keep in mind that different taxa with different pollen productivities and dispersal abilities are combined in one RPP value and the application to such broad-scale datasets can only be an approximation. This is especially important for the Northern Hemispheric dataset, which should not be applied to calculate site-specific vegetation

compositions. This dataset fills data gaps of RPP values in various regions, but at the cost of accuracy. We consider the presented averaged RPP values as a tool for data transformation to be applied to broad-scale pollen datasets. Using the dataset in this way can account for differences in pollen productivities and transportation rather than obtaining fully reliable quantitative information about the vegetation cover around a specific site. -----

*20.1* *Line 61: **we confined our analysis to publications with Poaceae as the reference taxon**. such publications according to what you are saying below! My initial comment was: Why? It is possible to convert RPP relative to other taxa (Pinus or Quercus for instance) into RPP relative to Poaceae, at least if the authors have calculated a RPP for Poaceae as well. It would be useful to indicate what studies you excluded on this basis.*

20.2 RESPONSE:
We adjusted our approach and provide detailed information and arguments of how we handled each single study.

20.3 TEXT (lines 89-100):
To be able to compare RPPs of different studies, it is necessary that all use the same reference; in our case Poaceae in accordance with most other studies. It is possible to recalculate RPP values based on other reference taxa by setting the original reference taxon to the RPP value resulting from other studies and recalculating all other RPPs based on that ratio (Mazier et al. 2012, Li et al. 2018). Of those studies selected for the continental RPP datasets, three did not have Poaceae as the original reference and did not include an RPP for Poaceae. The study of Bunting et al. (2005, reference taxon Quercus) did not provide standard deviations, so we used the values provided by Mazier et al. (2012) for this study, including the standard error. The RPPs of Li et al. (2015, reference taxon Quercus) were recalculated based on the mean Quercus RPP provided by Li et al. (2017), Zhang et al. (2017, Changbai), and Zhang et al. (2020). The RPPs of Matthias et al (2012, reference taxon Pinus) were recalculated based on the mean Pinus RPP provided by Räsänen et al. (2007) and Abraham and Kozáková (2012). The study of Jiang et al. (2020) used Quercus as the reference taxon but included a value for Poaceae, which was used as basis for recalculation.
* * *
*21.1* *Line 62: **boxplots of PPE per taxon were calculated with the help of R (version 3.5.3, R Core Team, 2019).***
*Please motivate/argue for the choice of this particular method.*

21.2 RESPONSE:
We adjusted our approach and oriented on the procedure by Mazier et al. (2012) and Li et al. (2018), by, amongst others, excluding the minimum and maximum value.

21.3 TEXT:
Lines 106-122:
In the choice of reliable values, we mainly followed the strategy of Mazier et al. (2012) and Li et al. (2018).
Dataset v1 includes all values of the chosen studies, except those RPPs which have an SD (or SE) > RPP.

Dataset v2 is further reduced with the following steps:

- If N≥5, the highest and smallest RPPs are excluded
- If N=4, the most deviating value from the Taxa-specific mean is excluded. Exception: if two values are from the same study (they are generally similar), their mean is calculated and used for the overall mean (→ *Salix* in America; *Betula*, Fabaceae and *Larix* in China; *Rumex* in Europe). The most deviating value is chosen based on the resulting mean. Exception in America: *Betula* with 4 values from only two studies are all kept.
- If N=3, a value is only excluded if it is strongly deviating (>100% of the mean of all values) → Caryophyllaceae of Li et al., in prep in China. Exceptions: in America Asteraceae and in Europe Apiaceae with three values from only two studies are all kept, as the two similar ones came from the same study.
- If N=2, all values are kept, except if one seems less reliable (*Larix*, Matthias et al. 2012)

Dataset v2 was created separately for each continent and is comparable to the Alt-1 dataset of Li et al. (2018) and PPE.st2 of Mazier et al. (2012).

Lines 133-136:
The majority of RPP studies concentrates on China and Europe, with one study from Arctic Russia and few studies from northern America. We thus decided to create a Northern Hemispheric dataset to be applied only for broad-scale studies for which otherwise RPP data for various taxa would be lacking. The dataset for the whole Northern Hemisphere was calculated with all data of the continental datasets.
* * *
22.1 *Line 64-65: **excluding outliers (defined as values outside the range of ±1.5 \* interquartile-range)**.*
*Could you please motivate/argue for the choice of this criteria to define outliers*

22.2 RESPONSE:
Please see our explanation just above (21) on our changed approach.
* * *
23.1 *Line 65: **delta method (Stuart and Ord, 1994).***
*Add: (...., 1994; see Li et al., 2020 for details on the method).*

23.2 RESPONSE:
We added this reference, but detected inconsistencies in the provision and naming of standard errors and standard deviations, which we commented as follows. We furthermore state in our metadata and the compilation (Table A1, A3) if SD or SE is given, if it is not clear what is given or if is named differently in different publications.

23.3 TEXT (Lines 123-130)
Lines 124-131
To calculate the SE of averaged RPPs, the delta method (Stuart and Ord, 1994, details in the supplement of Li et al. 2020) was applied. For the calculation of an RPP from pollen counts, a variance-covariance matrix is created. If only RPP ± SD (or SE) are available, the covariance is set to 0 and the final equation results in:

$$SE \sqrt{\frac{= \sum_{i=1}^{n}(var_i)}{(n*n)}}$$

Some problems arise from the labelling of standard errors and standard deviations. While some studies provide standard deviations, others provide standard errors or give no information. Some studies provide standard deviations, which are labelled as standard errors in other studies. Given this ambiguity, we used every value as it is and noted if standard deviation or standard error are said to be given.
* * *
24.1 *Line 65-68: **Subsequently, we looked at those taxa for which PPEs are available but do not have Poaceae as the reference taxon. These comprise Eleaganaceae, Nitraria, Tsuga, wild herbs and PoaceaeCrop. For studies that included Poaceae in the analysis set, we set Poaceae as 1 and recalculated the other PPEs based on that ratio***
*This comes too late! See my comment above. From your text above it seems you excluded those studies with other taxa than Poaceae as reference taxon.... Restructure the paragraph! - Other comment: "wild herbs" does not help us so much if we do not know what they include!*

24.2 RESPONSE:
We adjusted our approach on RPP recalculations as stated in answer to your comment 20. The "taxon" wild.herbs now comes with information on the included taxa.

24.3 TEXT (lines 89-100):
To be able to compare RPPs of different studies, it is necessary that all use the same reference; in our case Poaceae in accordance with most other studies. It is possible to recalculate RPP values based on other reference taxa by setting the original reference taxon to the RPP value resulting from other studies and recalculating all other RPPs based on that ratio (Mazier et al. 2012, Li et al. 2018). Of those studies selected for the continental RPP datasets, three did not have Poaceae as the original reference and did not include an RPP for Poaceae. The study of Bunting et al. (2005, reference taxon Quercus) did not provide standard deviations, so we used the values provided by Mazier et al. (2012) for this study, including the standard error. The RPPs of Li et al. (2015, reference taxon Quercus) were recalculated based on the mean Quercus RPP provided by Li et al. (2017), Zhang et al. (2017, Changbai), and Zhang et al. (2020). The RPPs of Matthias et al (2012, reference taxon Pinus) were recalculated based on the mean Pinus RPP provided by Räsänen et al. (2007) and Abraham and Kozáková (2012). The study of Jiang et al. (2020) used Quercus as the reference taxon but included a value for Poaceae, which was used as basis for recalculation.
* * *
25.1 *Line 67: **PoaceaeCrop***
*I guess this is "Cerealia type"? Why not call it cereals?*

25.2 RESPONSE:
Yes, changed accordingly.
* * *
*26.1  Line 68: **(applies to…)***
*What do you mean? "following (or according to) Sugita et al....."?*

26.2  RESPONSE:
This paragraph has been deleted
* * *
*27.1  Line 69-72: **For all other studies (Calcote, 1995; Chaput and Gajewski, 2018; Li et al., 2015; Matthias et al., 2012; Theuerkauf et al., 2015), we recalculated the PPEs based on the original reference taxon. If, for example, Acer was used as the reference taxon, we assumed the Poaceae-to-Acer PPE to be the same as our calculated mean PPE for Acer and recalculated the other values based on that ratio***
*Same comment as above.*
*And, add a reference for this way to recalculate RPPs (I think Mazier et al., 2012 was one of the first using this approach).*

27.2  RESPONSE:
We are not sure which of the above comments is meant. However, we adjusted our approach on recalculating the reference taxon (see our responses to your comments 20 and 24).

27.3  TEXT (lines 89-100):
To be able to compare RPPs of different studies, it is necessary that all use the same reference; in our case Poaceae in accordance with most other studies. It is possible to recalculate RPP values based on other reference taxa by setting the original reference taxon to the RPP value resulting from other studies and recalculating all other RPPs based on that ratio (Mazier et al. 2012, Li et al. 2018). Of those studies selected for the continental RPP datasets, three did not have Poaceae as the original reference and did not include an RPP for Poaceae. The study of Bunting et al. (2005, reference taxon Quercus) did not provide standard deviations, so we used the values provided by Mazier et al. (2012) for this study, including the standard error. The RPPs of Li et al. (2015, reference taxon Quercus) were recalculated based on the mean Quercus RPP provided by Li et al. (2017), Zhang et al. (2017, Changbai), and Zhang et al. (2020). The RPPs of Matthias et al (2012, reference taxon Pinus) were recalculated based on the mean Pinus RPP provided by Räsänen et al. (2007) and Abraham and Kozáková (2012). The study of Jiang et al. (2020) used Quercus as the reference taxon but included a value for Poaceae, which was used as basis for recalculation.
* * *
*28.1  Line 74-75: **Publications are not considered if neither Poaceae was used as a reference nor species included for which no PPE would be available without recalculation***
*Whar do you mean? This sentence does not make sense! Rewrite!*

28.2  RESPONSE:
We adjusted our approach on recalculating the reference taxon and this paragraph has been deleted (see our responses to your comments 20, 24 and 27)
* * *
29.1 *Line 68-92:* **Changbai site from Zhang et al. (2017) were excluded from calculations because Poaceae was not the reference taxon. Eight studies did not have Poaceae as the reference, but did include PPE values for taxa which would otherwise not be represented in the final PPE-***dataset* **(applies to Calcote, 1995; Filipova-Marinova et al., 2010; Li et al., 2011; Matthias et al., 2012; Sugita et al., 1999, 2006; Theuerkauf et al., 2015; Zhang et al., 2017 (site Taiyue))***. These taxa are Aesculus, Elaeagnaceae, Nitraria, PoaceaeCrop, Pterocrya, Tsuga and wild.herbs (Table 3). The final dataset consists of PPE values for 58 taxa and fall speeds for 57 taxa, with 54 taxa having both PPE values and fall speeds available (Table 3).*
*This is already said in Methods. I would recommend to have this only in Methods and to explain this better. It is very confusing as it is written now. AS far as I understand you have 3 categories of studies with reference taxa different than Poaceae: a) reference taxon different than Poaceae, but RPP for Poaceae available: conversion is simple; b) reference taxon different than Poaceae, and RPP for Poaceae not available: conversion requires assumption "we recalculated the PPEs based on the original reference taxon. If, for example, Acer was used as the reference taxon, we assumed the Poaceae-to-Acer PPE to be the same as our calculated mean PPE for Acer and recalculated the other values based on that ratio"; c) reference taxon different than Poaceae, and xxxxxx (I don't understand what is characteristic for this third group): conversion is not possible if we want to use the same assumption/approach as for b). Once these 3 groups are well defined, make a Table with the studies and taxa that are related to those 3 groups.*

29.2 RESPONSE:
As stated above, we adjusted our approach on recalculating the reference taxon and this paragraph has been deleted (see our responses to your comments 20, 24 and 27). We hope that the procedure is clearer now.
* * *
30.1 *Line 98:* **as well as subtropical regions are largely lacking**
*Did you include the subtropics in your search of literature? Seems to me that some studies are lacking in that case. There are at least studies in press. If you want to include the subtropics in this study/paper you should tell what is in progress as well.*

30.2 RESPONSE:
We conducted an open search (see search terms in lines 42-44), which resulted in the literature given in Table 1. However, a new search brought some additional publications (including those in the subtropics), which we added to the compilation and analyses.

30.3 TEXT:
Lines 46-50
To find literature on relative pollen productivity estimates (RPP or PPE), we conducted internet searches in Google Scholar (https://scholar.google.de/) and the Web of Science (https://apps.webofknowledge.com/) for the terms "PPE", "RPP", "Pollen productivity", "Pollen productivity estimates", and various combinations of our search terms. Furthermore, we used literature cited in publications on RPPs to gain the most complete overview possible of existing literature about Northern Hemispheric RPPs.

Line 236
RPP studies in Russian and North American boreal forests as well as in tropical regions are largely lacking.
* * *
*31.1   Line 100: **can be***
*Has been*

31.2   RESPONSE:
Changed accordingly.
* * *
*32.1   Line 101: **broad scale***
*what do you mean? A large number? Or values from different parts of the world, or both? Clarify!*

32.2   RESPONSE:
Large number. Changed accordingly.
* * *
*31.1   Line 105: **show***
*Shows*

31.2   RESPONSE:
Changed accordingly.
* * *
*32.1   Line 105-106: **While Cyperaceae or Fraxinus***
*Alnus, Quercus and Chenopodiaceae would be better examples. Low values have a tendency to "look" more similar than large values, but you should think about what a RPP is used for. A RPP of 2 is very different than a RPP of 1, even if it does not look to be very different. It means that one of the value is double the other. A RPP of 10 seems much different than a RPP of 20, but in fact it's the same difference, one is double the other. The difference it will imply in the application of these values in reconstructions will be the same in both cases.*

33.1   RESPONSE:
We reconsidered our analyses and examples based on your remarks and our analysis of absolute deviation between the Northern Hemispheric dataset and those published by Mazier et al. 2012 and Li et al. 2018.

33.2   TEXT (lines 227-230):
Comparison with taxa available in the compilations of Mazier et al. (2012, Europe) and Li et al. (2018, temperate China) clearly shows differences in absolute RPP values or a high absolute deviation for some taxa (Figure 5, e.g. *Juniperus*, *Artemisia*, Rosaceae), while many others (e.g.

*Alnus*, *Quercus* or Ranunculaceae) have a similar range of values, especially when considering the absolute deviation
* * *
33.1  *Line 106: **similar range***
*Do you mean? similar values?*

33.2  RESPONSE:
Yes

33.3  TEXT line 230:
similar range of values
* * *
34.1  *Line 106-107: **strongly vary between the publications***
*In fact it is mainly a large difference between Europe and China for Artemisia, Betula and Pinus, as discussed in Li et al. 2018*

34.2  RESPONSE:
As described above, we compared in detail the variability within and between continents. Among the 6 taxa that have enough RPP values available (at least two values in two continents), a significant variability has only been found for Asteraceae. We furthermore present the absolute deviation of the Northern Hemispheric dataset compared to continental datasets to inform the reader/user which taxa show most variation. (see Figure 4)

34.3  TEXT:
TEXT (lines 227-230):
Comparison with taxa available in the compilations of Mazier et al. (2012, Europe) and Li et al. (2018, temperate China) clearly shows differences in absolute RPP values or a high absolute deviation for some taxa (Figure 5, e.g. *Juniperus*, *Artemisia*, Rosaceae), while many others (e.g. *Alnus*, *Quercus* or Ranunculaceae) have a similar range of values, especially when considering the absolute deviation.
* * *
35.1  *Line 115-119: **such as vegetation characteristics, and some uncertainty due to the use of inconsistent reference taxa. Most studies used Poaceae, a widespread family, whose pollen is easy to identify and often preserved in a good state. However, as discussed by Broström et al. (2008), the pollen cannot be identified to species level and different studies may thus have used different species of Poaceae for the reference. Other taxa such as Quercus or Acer are therefore sometimes used as the reference taxon***
*These explanations are the general ones that have been discussed in all syntheses so far. However, for Artemisia and Pinus in particular, Li et al. (2018) have discussed the differences within China with more specific arguments. Please revise accordingly.*

35.2  RESPONSE:

We added more information from the discussion of Li et al. (2018)

35.3 TEXT (lines 252 – 274):
We found that RPP values partly vary between the three continental datasets. Some uncertainty arises due to the use of inconsistent reference taxa. Most studies used Poaceae, a widespread family, whose pollen is easy to identify and often preserved in a good state. However, as discussed by Broström et al. (2008), the pollen cannot be identified to species level and different studies may thus have used different species of Poaceae for the reference. Other taxa at higher taxonomic resolution such as *Quercus* or *Acer* are therefore sometimes used as the reference taxon (see Table A1 in the Appendix).

Reasons for variable RPP values have been discussed in depth by Broström et al. (2008) and Li et al. (2018), and are mainly methodological factors such as different sampling designs and environmental factors such as vegetation characteristics. Furthermore, pollen taxa from different sites can contain different species. Li et al. (2018) discussed in detail for *Pinus* and *Artemisia*, that vegetation structure and climate of different Chinese study regions, but also methodological differences like the pollen sample type (moss vs. lake sediment) and vegetation sampling method, can explain the variability of RPPs within one taxon even better than the occurrence of different taxa. This will be even more apparent when combining data for the whole Northern Hemisphere. However, our compilation clearly indicates that taxa have mostly characteristic RPP values (i.e. within-species variability is low compared to variability between species), while we found no significant differences between continents (i.e. variability within continents is not lower than variability between continents). This implies, when aiming to compare vegetation change between continents, that transformation of pollen data using RPP from another continent is better than keeping the data untransformed. While one has to keep in mind the limited amount of data influencing the statistical power, we conclude that there is no particular reason to not set up a Northern Hemispheric RPP dataset. Still, before applying one of the datasets presented, researchers should consult the original publication to be sure it fits their needs and standards and be aware of the rather problematic use of SD and SE, which might have influenced our presented SEs.
* * *
36.1 *Line 124-125:* **This open access dataset can be used to improve our understanding of past vegetation dynamics for a broad range of taxa rather than interpreting pollen counts or percentages alone.**
*This is perhaps my most serious concern about this paper. If the dataset made open access in PANGAEA is used as a "black box" in will really NOT improve our understanding of past vegetation cover. It may provide results that are nonsense. Please read my detailed separate comment about this, and revise accordingly.*

36.2 RESPONSE:
We hope that our revised dataset will not be used as a "black box". We hope that our reanalysis of the data has improved the dataset and we strongly warn the reader not to use the dataset without their own assessment on its suitability for their planned application. We will furthermore add a note in PANGAEA that the dataset should not be used without first reading this accompanying data description paper (this manuscript). We consider open access to be the most useful way of providing information.

36.3 TEXT (lines 279-287):
The continental datasets can be applied to assess vegetation changes using broad-scale pollen datasets. It is important to keep in mind that different taxa with different pollen productivities and dispersal abilities are combined in one RPP value and the application to such broad-scale datasets can only be an approximation. This is especially important for the Northern Hemispheric dataset, which should not be applied to calculate site-specific vegetation compositions. This dataset fills data gaps of RPP values in various regions, but at the cost of accuracy. We consider the presented averaged RPP values as a tool for data transformation to be applied to broad-scale pollen datasets. Using the dataset in this way can account for differences in pollen productivities and transportation rather than obtaining fully reliable quantitative information about the vegetation cover around a specific site.
* * *
37.1 *Line 127:* **The compilation is useful for the identification of available PPE sets at specific sites and regions.**
*Yes. But I don't think it adds so terribly much to the syntheses of Li et al. 2018 and Mazier et al. (2012). There are moreover syntheses in progress for northern America (Dawson et al., personal communication) and a new one for Europe (Gaillard et al.).*

37.2 RESPONSE:
Thank you very much for this information. However, we cannot take a position on these syntheses as they have not yet been published. We acknowledge the work Mazier et al. 2012 and Li et al. 2018 put into their compilations. Several new studies have been published since then but do not include studies in Arctic Russia or North America. We consider our study mainly as providing a basic and currently complete overview of RPP studies and are convinced that a complete compilation, including a first overview on the background of the studies (Table A2, A3), is of value to the scientific community.
* * *
38.1 *Line 129-130:* **The unified PPE dataset of Northern Hemispheric extratropical taxa allows a consistent approach to be applied to synthesised pollen data at a continental to hemispherical scale**
*Not the best way to do it, the values need to be evaluated!*

39.2 RESPONSE:
As described above we added some evaluation in particular with respect to regional differences of the RPP values. A full evaluation is, however, beyond our scope.
* * *
40.1 *Line 132-133:* **and PPEs from a nearby local study would not necessarily be best suited for their interpretation.**
*Please develop what you mean here. It is true that one can think of using RPPs from particular modern climate zones for time periods of the past with similar climate conditions at other geographical locations than today. However, doing so imply that we first can demonstrate that differences in RPP between studies are indeed due to climatic parameters. It is certainly useful*

*to have all these values in a database. BUT, all scientists that will use them will HAVE TO go back to the original publications to evaluate the values. Anybody can do alternative syntheses to those already published by Mazier et al 2012 or Li et al 2018. But what is MOST needed to day is VALIDATION of these values.*

40.2 RESPONSE:
An in-depth assessment (i.e. requiring recalculation of RPPs) is out of the scope of our manuscript and is also not possible because most studies do not provide the original data. However, we systematically collected all the relevant information for each RPP study (Table A1, A2, A3 ) that can be used to assess the quality of the studies and further indicate whether or not the data were evaluated by the original authors or other experts (Mazier et al 2012 and Li et al 2018).
* * *
41.1 *Line 146-147:* **This study is a contribution to the Past Global Changes (PAGES) LandCover6k working group project.**
*It won't be accepted as a contribution to LandCover6k if relevant revisions are not implemented. This study has not been discussed with any of the responsible coordinators of PAGES LandCover6k. PAGES LandCover6k has a certain number of quality requirements for the studies that are performed as contributions. We can of course use different methods and have other ideas in terms of interpretation of results. But we can't accept scientific/theoretical mistakes as it is the case here.*

41.2 RESPONSE:
We hope that an overview of all available RPP data that comes along with a thoroughly revised manuscript according to your and reviewer 1's constructive suggestions will be considered as a contribution to PAGES LandCover6k. We also specifically submitted this data to ESSD, as we know that the entire community can engage with the revision of data and the manuscript.
* * *
42.1 *Table 1:* **Broström et al., 2008, Mazier et al., 2012**
*Would be useful to know from the table that this is a review*

42.2 RESPONSE:
The information was added.
* * *
43.1 *Table 2:* **Taxon**
*Are these thought as pollen morphological taxa? In that case, specify! It is not the same as plant taxa*

43.2 RESPONSE:
Yes, these are pollen morphological taxa. The information was added.
* * *
*44.1*   *Table 2: **Original Taxa***
*Here again you should specify that these are the pollen-morphological taxa for which RPP were estimated in the literature; for some of these taxa it is not relevant to have lost the information on RPP for particular species or genus in the case of such a synthesis. To group pollen-morphological types is sometimes needed if those taxa were not identified in the fossil pollen records used for reconstruction. But in other cases these RPPs can be used. I am thinking of e.g. Compositae SF Cichoriodae, Aster T. /Anthemis T, Calluna vulgaris, Pinus cembra, Secale cerealia, Trollius europaeus, Filipendula, Potentilla type, different Plantago species, P. lanceolata in particular, etc.*

44.2   RESPONSE:
The grouping presented in Table 2 is particularly important for the harmonised dataset. Since we do not think that the Northern Hemispheric RPP dataset will be applied for site-specific studies, but for large, taxonomically harmonised datasets, we regard the groups as useful taxonomic units. As the original values are presented as well, users have the possibility to harmonise the data according to their needs.
* * *
*45.1*   *Table 2: **Aster-Anthemis type***
*italic and write Aster/Anthemis type*

45.2   RESPONSE:
Changed accordingly.
* * *
46.1   *Table 2: **Leucanthemum vulgate***
*vulgare?*

46.2   RESPONSE:
Yes, changed accordingly.
* * *
47.1   *Table 2: **Robinia Sophora***
*write Robinia/Sophora*

47.2   RESPONSE:
Changed accordingly.
* * *
*48.1*   *Table 2: **Cercis***
*italic!*

48.2  RESPONSE:
Changed accordingly.
* * *
49.1  *Table 2: Lamiaceae + Mentha type Thymus + Thymus praecox + **Vitex negundo***
*very different pollen-morphological type; can't be harmonized in this way*

49.2  RESPONSE:
We took Vitex negundo out of the group of Lamiaceae
* * *
50.1  *Table 2 : Mentha type **Thymus***
*+ Thymus +*

50.2  RESPONSE:
Changed to Mentha type (Thymus).
* * *
51.1  *Table 2: **PoaceaeCrop***
*In the table of mean RPPs you have also cereals, what is the difference between PoaceaeCrop and Cereals?*

51.2  RESPONSE:
PoaceaeCrop and Cerealia are now combined.
* * *
52.1  *Table 3: Have genus names in italic*

52.2  RESPONSE:
Will be changed accordingly.
* * *
53.1  *Table 3: **PoaceaeCrop***
*Are these wild Poaceae that are cultivated? Or Cerealia type? Isn't it a "synonym" for "Cerealia"? What is this taxon good for?*

53.2  RESPONSE:
PoaceaeCrop and Cerealia are now combined.
* * *
54.1  *Table 3: **wild herbs***
*What wild herbs? What is this taxon good for if we do not know what plant taxa it includes?*

54.2 RESPONSE:
Wild herbs have information now on the included taxa.
* * *
55.1 *Figure 1: What means the asterix? What is (from Gaillard 1994)? What publication is it? Not in the list of publication. And why? What is wrong with Sugita et al. 1999 as reference in this case?*

55.2 RESPONSE:
We are not sure which asterisk is meant. Some references have a ° indicating that they are extracted from Li et al. 2018, as stated in the figure caption.
Sugita et al. 1999 stated their data were from Gaillard et al. 1994, however we have now cited only Sugita et al. 1999.
* * *
56.1 *Figure 2: **which***
*replace by "that"*

56.2 RESPONSE:
Figure has been changed
* * *
57.1 *Figure 3: **values***
*Add SEs for all values!!!!*

57.2 RESPONSE:
SEs are added in all graphs
* * *
58.1 *Figure 3: **species***
*Taxa*

58.2 RESPONSE:
Has been changed accordingly (now Figure 5)
* * *
59.1 *Figure 3: upper panel*
*Would be good to also add the grey mean in the upper panel*

59.2 RESPONSE:
Having calculated continental datasets as well, the data of the upper panel of old Figure 3 are now shown in Figures 2 and 3, including the mean.
* * *
60.1   *Figure 3: lower panel*
       *Specify that the grey bars are acoording to "this paper, mean values"!*

60.2   RESPONSE:
       The figure has changed slightly. All colours are explained and the grey bars are specified to be
       the Northern Hemispheric dataset
* * *
61.1   *Figure 3: **showing similar values for some and a high variability for other taxa.***
       *delete; not relevant in a Figure caption*

61.2   RESPONSE:
       Removed.

[revised manuscript text omitted]

65 ~~further analyses. Thus, of the resulting 54 publications from our literature search, 13 were excluded a priori and are marked with an x in Table 1. The review of Li et al. (2018) does not contain newly calculated PPE, but is a collection of Chinese PPE studies, which of we incorporated those only available in Chinese language into our study. In the end, we use a total of 40 publications for our analyses.~~

**2.2 RPP**

70 **RPP Compilation**

All RPP values and, if given, their standard deviation (SD) or standard error (SE)) were collected from the literature. If the data were only presented as figures, values were extracted with the help of Corel Draw X6. The studies of Ge et al. (2015), He et al. (2016), Li et al. (in prep), Wu et al. (2013) and Zhang et al. (2017) are only available in Chinese and RPP values where extracted from Li et al. (2018), while the study

75 of Chen et al. (2019) was extracted from Jiang et al. (2020).

assigned broader taxonomic levels to some taxa of the original publications and calculated means if more than one value of finer taxonomic levels was available (Table 2).

A complete compilation of PPE data and their references is given in the Appendix. Some publications did not use Poaceae as the reference taxon: while it is possible to recalculate values relative to Poaceae (cf. Li et al., 2018; Mazier et al., 2012), we confined our analysis to publications with Poaceae as the reference taxon.

While different approaches exist to estimate RPP, the extended R-value (ERV) is the most common approach. Details on the ERV model and related assessment criteria can be found in, for example, Abraham and Kozáková (2012), Bunting et al. (2013) and Li et al. (2018). The maximum likelihood method (decreasing likelihood function score or increasing log-likelihood with distance) can be used to identify the relevant source area of pollen (RSAP) and should reach an asymptote with increasing sampling distance (Sugita 1994). For reliable results, the vegetation sampling area should be ≥ RSAP (Sugita 1994). Unexpected behaviour of the maximum likelihood method can occur if assumptions of the ERV-model are not met (Li et al. 2018). Furthermore, a sufficient number of randomly selected sites (no of sites ≥ number of taxa for RPP-estimation) is necessary (Li et a. 2018). Last but not least, for the correct application of the REVEALS model, RPPs need to have a standard deviation provided, to allow for correct estimation of the vegetation cover.

To allow for further assessment of the presented RPP data, we collected information on, for example, the maximum likelihood, the vegetation sampling radius, and the site distribution used in the different studies. (Table A2). This will help researchers when creating customised RPP datasets. If RPP estimates for several models (e.g. ERV-submodel 1, 2 or 3) were presented in the original study, we used all of them for the RPP compilation and added the information on which one was chosen as best fit by the original author and/or in the RPP-compilations of Mazier et al. (2012) and Li et al. (2018) (Tables A1, A3).

To achieve our goal of a unified PPE dataset for the whole Northern Hemisphere, boxplots of PPE per taxon were calculated with the help of R (version 3.5.3, R Core Team, 2019). A set of PPE-means per taxon was then created by calculating the mean of all values excluding outliers (defined as values outside the range of ±1.5 * interquartile range). The SE was estimated using the delta method (Stuart and Ord, 1994). Subsequently, we looked at those taxa for which PPEs are available but do not have Poaceae as the reference taxon. These comprise Eleaganaceae, *Nitraria*, *Tsuga*, wild herbs and PoaceaeCrop. For studies that included Poaceae in the analysis set, we set Poaceae as 1 and recalculated the other PPEs based on that ratio (applies to Sugita et al., 1999). For all other studies (Calcote, 1995; Chaput and Gajewski, 2018; Li et al., 2015; Matthias et al., 2012; Theuerkauf et al., 2015), we recalculated the PPEs based on the original reference taxon. If, for example, Acer was used as the reference taxon, we assumed the Poaceae-to-Acer PPE to be the same as our calculated mean PPE for Acer and recalculated the other values based on that ratio. For Zhang et al. (2017) and Li et al. (2011) only values recalculated relative to Poaceae are available and extracted from Li et al. (2018), thus we did not conduct our own recalculations. Publications are not considered if neither Poaceae was used as a reference nor species included for which no PPE would be available without recalculation (applies to Andersen, 1967; Bunting et al., 2005; He et al., 2016; Li et al., 2015; Sjögren et al., 2008b; Theuerkauf et al., 2013; Twiddle et al., 2012; Wu et al., 2013; Zhang et al., 2017 (Changbai Mountains)).

[revised manuscript text omitted]
, a final number of 3127 publications and 3531 sites isare included in the final PPE dataset (https://doi.pangaea.de/10.1594/PANGAEA.908862, Wieczorek and Herzschuh, 2019). Nine publications + the Changbai site from Zhang et al. (2017) were excluded from calculations because Poaceae was not the reference taxon. Eight studies did not have Poaceae as the reference, but did include PPE values for taxa which would otherwise not be represented in the final PPE dataset (applies to Calcote, 1995; Filipova-Marinova et al., 2010; Li et al., 2011; Matthias et al., 2012; Sugita et al., 1999, 2006; Theuerkauf et al., 2015; Zhang et al., 2017 (site Taiyue)). These taxa are *Aesculus*, Elaeagnaceae, *Nitraria*, PoaceaeCrop, *Pterocrya*, *Tsuga* and wild.herbs (Tabledatasets (10 studies and 11 datasets for China, 14 studies and 16 datasets for Europe, 3). The final studies and 4 datasets for America). We have RPP data for 33 taxa in China, 34 taxa in Europe and 25 taxa in northern America. The Northern Hemispheric dataset consists of RPPE values for 58 taxa and fall speeds for 5755 taxa (Tables 3-6, https://doi.pangaea.de/10.1594/PANGAEA.908862). Twenty-eight taxa, with 54 taxa having both PPE values and fall speeds are available (Table 3). All PPE data in only one of the final dataset are given relative to Poaceea, markedcontinental datasets (13 in redChina, 6 in TableAmerica, 9 in Europe).

[revised manuscript text omitted]

335 **4.2 How to use the dataset**

~~Using PPEs for pollen-based quantitative vegetation reconstruction (Sugita, 2007; Theuerkauf et al., 2016) has improved our understanding of environmental change (e.g. Marquer et al., 2014). In this paper, we present a dataset of Northern Hemispheric extratropical PPEs and corresponding fall speeds based on a compilation of studies. This open access dataset can be used to improve our understanding of past vegetation dynamics for a~~
340

345

compositions and climate conditions, and PPEs from a nearby local study would not necessarily be best suited for their interpretation.

**5 How to use the datasets**

The RPP compilation can be used to get a good overview of existing RPP studies, to identify research gaps and to find RPPs to apply at one's study area. It is important (i) to use only those RPP data which have been evaluated by experts or the author as best fit and (ii) to look at the original publication for further information on how the RPP estimates have been generated.

The continental datasets can be applied to assess vegetation changes using broad-scale pollen datasets. It is important to keep in mind that different taxa with different pollen productivities and dispersal abilities are combined in one RPP value and the application to such broad-scale datasets can only be an approximation. This is especially important for the Northern Hemispheric dataset, which should not be applied to calculate site-specific vegetation compositions. This dataset fills data gaps of RPP values in various regions, but at the cost of accuracy. We consider the presented averaged RPP values as a tool for data transformation to be applied to broad-scale pollen datasets. Using the dataset in this way can account for differences in pollen productivities and transportation rather than obtaining fully reliable quantitative information about the vegetation cover around a specific site.

**6 Data Availability**

The RPPE compilation as well as the taxonomically harmonised PPE datasetcontinental RPP datasets are available at https://doi.pangaea.de/10.1594/PANGAEA.908862 (Wieczorek and Herzschuh 2019). The updating of the dataset on PANGAEA is currently in progress, the data are available as supplementary material

**67 Author Contribution**

MW and UH designed the study and wrote the Manuscript, MW carried out the analyses and produced tables and figures.

**78 Competing interests**

The authors declare that they have no conflict of interest.

**89 Acknowledgements**

The study was supported by and conducted as part of the ERC consolidator grant "GlacialLegacy" (Call: ERC-2017-COG, Project Reference: 772852) and PalMod Initiative (Grant 01LP1510C). We thank all scientists conducting research on pollen productivity, whose previous work and published data made our compilation possible. We thank C. Jenks for language correction. This study is a contribution to the Past Global Changes (PAGES) LandCover6k working group project.Many thanks  Marie-José Gaillard and an anonymous referee for

taking the time to give us very helpful and constructive reviews and suggestions to improve our dataset and
380 manuscript. We thank C. Jenks for language editing.

[revised manuscript text omitted]

Caryophyllaceae woody 0.74 0.07 0.034 | Populus 0 2 0.67 0.09085 0.026 1 3.42 1.600 0.025 3 1.59 0.536 0.026

Castanea 11.49 0.49 0.004 | *Pterocarya 26.84 0.00 0.029

Cerealia 1.13 0.23 0.065 | Quercus 3.33 0.11 0.023

Chenopodiaceae 4.01 0.32 0.016 | Ranunculaceae 1.66 0.16 0.014

Convolvulaceae 0.18 0.03 | Rosaceae 1.10 0.09 0.010

Cornaceae woody 1.72 0.14 0.044 | Rubiaceae 1 1.85 0.1636 0.019 0 5 1.75 0.138 0.019 6 1.67 0.129 0.019

woody | Corylus 1.97 0.133.17 0.20 0.012 0 4 1.44 0.06 0.025 Rumex5 1.6178 0.15066 0.0159

Cupressaceae 1.11 0.09 0.010 | Salix 0.57 0.03 0.022

Cyperaceae woody | Ulmus 0.76 0.032 2.24 0.46 0.024 Sambucus nigra-type 0 1.30 0.12 0.013 0 0.032 2 2.24 0.462 0.026

*Elaeagnaceae woody 8.88 | Fraxinus 1.302 0.0161.05 0.18 0.020 Sanguisorba 24.07 3.500 5 2.97 0.252 0.022 7 2.42 0.187 0.020

Ephedra woody 0.96 | Fagus 0.14 0.015 Selaginella 0 0.041 5 2.92 0.133 0.056

Equisetum 0.09 0.02 0.021 | Solanaceae 0.027

Ericales woody | Juglans 0.59 0.015 3.28 0.12 0.032 Thalictrum 0 5 3.8628 0.26119 0.009032

Fabaceae woody 0.33 | Larix 0.044 2.310.025 0.16 Thymelaeaceae 0.119 33.05 0 0.126 2 5.73 1.165 0.126 6 3.7845 0.03140 0.122

Fagus woody 1.96 | Quercus 7 2.50 0.05 0.021 1 02.08 0.05743 0.035 Tilia 71.17 34.88 0.17 0.03108 0.035 15 3.58 0.056 0.024

woody | Carpinus 0 0 5 4.31 0.216 0.042 5 4.31 0.216 0.042

Gelöschte Zellen …
Gelöschte Zellen …
Gelöschte Zellen …
Gelöschte Zellen …
Eingefügte Zellen …
Eingefügte Zellen …
Eingefügte Zellen …
Eingefügte Zellen …
Eingefügte Zellen …
Eingefügte Zellen …
Eingefügte Zellen …
Eingefügte Zellen …
Eingefügte Zellen …
Eingefügte Zellen …
Eingefügte Zellen …
Eingefügte Zellen …
Eingefügte Zellen …
Eingefügte Zellen …
Eingefügte Zellen …
Eingefügte Zellen …
Gelöschte Zellen …
Gelöschte Zellen …
Gelöschte Zellen …
Gelöschte Zellen …
Eingefügte Zellen …
Eingefügte Zellen …
Eingefügte Zellen …
Eingefügte Zellen …
Eingefügte Zellen …
Eingefügte Zellen …
Eingefügte Zellen …
Eingefügte Zellen …
Eingefügte Zellen …
Eingefügte Zellen …
Eingefügte Zellen …
Eingefügte Zellen …

[revised manuscript text omitted]

Figure 2: Overview of all pollen productivity estimate (PPE) values (left panel) which were included in the calculation of a mean PPE. The higher the PPE, the higher the overrepresentation of a taxon in a pollen sample and vice versa. Fall speeds (right panel) which have been used to calculate a mean fall speed per taxon. The lower the fall speed, the farther a pollen grain can be carried through the air and vice versa. Pollen with a low PPE and a high fall speed are thus highly underrepresented in pollen samples.

655

[Figure]

Figure 3: Pollen productivity estimate (PPE) values for selected species, compared between different regions within this paper (upper panel) and different studies (lower panel), showing similar values for some and a high variability for other taxa. We chose those taxa which are present in at least two study regions or two publications.

660

Appendix

*Table A1 Overview of available studies and their PPE values.*

665 See attached pdf "Table A1 or https://doi.pangaea.de/10.1594/PANGAEA.908862 - Compilation of pollen productivity estimates from Northern Hemisphere extratropics.

**Table** *A2 Raw fall speeds with original pollen type, target taxon as used for our dataset, fall speed in m/s, the original publication of the value and the publication we are citing from*

| Pollen morphological type | Target taxon | FS (m/s) | Original publication | Found in: |
|---|---|---|---|---|
| Abies | Abies | 0.12 | Eisenhut, 1961 | Cao et al., 2019 |
| Abies | Abies | 0.12 | Mazier et al., 2012 | Chaput and Gajewski, 2018 |
| Acer | Acer | 0.056 | Mazier et al., 2012 | Chaput and Gajewski, 2018 |
| Acer | Acer | 0.056 | Sugita, 1993, 1994 | Sugita et al., 1999 |
| Achillea | Asteraceae | 0.017 | - | Bunting et al., 2013 |
| Aesculus | Aesculus | 0.029 | Knoll, 1932 | Flilipova-Marinova et al. , 2010 |
| Alnus | Alnus | 0.021 | Eisenhut, 1961 | Cao et al., 2019 |
| Alnus | Alnus | 0.021 | Eisenhut, 1961 | Cao et al., 2019 |
| Alnus | Alnus | 0.021 | Gaillard et al., 2008 | Hjelle and Sugita, 2012 |
| Amaranthaceae/Chenopodiaceae | Chenopodiaceae | 0.027 | - | Li et al., 2017 |
| Ambrosia | Asteraceae | 0.019 | - | Commerford et al., 2013 |
| Artemisia | Artemisia | 0.007 | Han et al., 2017 | Li et al., 2018 |
| Artemisia | Artemisia | 0.007 | Zhang et al., 2017 | Li et al., 2018 |
| Artemisia | Artemisia | 0.009 | Ge et al., 2015 | Li et al., 2018 |
| Artemisia | Artemisia | 0.009 | Li et al., in prep. | Li et al., 2018 |
| Artemisia | Artemisia | 0.009 | Zhang et al., 2017 | Li et al., 2018 |
| Artemisia | Artemisia | 0.0093 | - | Xu et al., 2014 |
| Artemisia | Artemisia | 0.01 | Li et al., 2011 | Li et al., 2018 |
| Artemisia | Artemisia | 0.01 | Wu et al., 2013 | Li et al., 2018 |
| Artemisia | Artemisia | 0.0101 | - | Wang and Herzschuh, 2011 |
| Artemisia | Artemisia | 0.013 | - | Commerford et al., 2013 |
| Artemisia | Artemisia | 0.014 | - | Abraham and Kozáková, 2012 |
| Artemisia | Artemisia | 0.021 | He et al., 2016 | Li et al., 2018 |
| Artemisia | Artemisia | 0.021 | - | Poska et al., 2011 |
| Artemisia | Artemisia | 0.015 | - | Li et al., 2017 |
| Aster/Anthemis-type | Asteraceae | 0.025 | - | Li et al., 2017 |
| Asteraceae | Asteraceae | 0.0118 | - | Xu et al., 2014 |
| Asteraceae | Asteraceae | 0.014 | - | Commerford et al., 2013 |
| Asteraceae | Asteraceae | 0.051 | Broström, 2002 | Cao et al., 2019 |
| Asteraceae SF | Asteraceae | 0.028 | - | Li et al., 2017 |
| Avena-Triticum group (m) | PoaceaeCrop | 0.06 | Gregory, 1973 | Theuerkauf et al., 2015 |
| Avena-Typ (b) | PoaceaeCrop | 0.06 | Gregory, 1973 | Theuerkauf et al., 2015 |
| Avena-type | PoaceaeCrop | 0.078 | Soepboer et al., 2007 | Matthias et al., 2012 |

| Pollen morphological type | Target taxon | FS (m/s) | Original publication | Found in: |
|---|---|---|---|---|
| Betula | Betula | 0.011 | Wu et al., 2013 | Li et al., 2018 |
| Betula | Betula | 0.011 | Zhang et al., 2017 | Li et al., 2018 |
| Betula | Betula | 0.017 | Zhang et al., 2017 | Li et al., 2018 |
| Betula | Betula | 0.018 | - | Bunting et al., 2013 |
| Betula | Betula | 0.019 | Han et al., 2017 | Li et al., 2018 |
| Betula | Betula | 0.019 | - | Li et al., 2015 |
| Betula | Betula | 0.024 | Eisenhut, 1961 | Cao et al., 2019 |
| Betula | Betula | 0.024 | Eisenhut, 1961 | Cao et al., 2019 |
| Betula | Betula | 0.024 | Gaillard et al., 2008 | Hjelle and Sugita, 2012 |
| Betula | Betula | 0.024 | Jackson and Lyford, 1999 | Bunting et al., 2013 |
| Betula | Betula | 0.024 | Mazier et al., 2012 | Chaput and Gajewski, 2018 |
| Betula | Betula | 0.11 | - | Bunting et al., 2013 |
| Brassicaceae | Brassicaceae | 0.0034 | - | Xu et al., 2014 |
| Brassicaceae | Brassicaceae | 0.02 | - | Li et al., 2017 |
| Calluna vulgaris | Ericales | 0.038 | Gaillard et al., 2008 | Hjelle and Sugita, 2012 |
| Calluna vulgaris | Ericales | 0.038 | - | Broström et al., 2004 |
| Campanula | Campanulaceae | 0.022 | - | Bunting et al., 2013 |
| Cannabis/Humulus | Humulus | 0.01 | - | Li et al., 2017 |
| Carpinus | Carpinus | 0.042 | Eisenhut, 1961 | Cao et al., 2019 |
| Caryophyllaceae | Caryophyllaceae | 0.028 | - | Bunting et al., 2013 |
| Caryophyllaceae | Caryophyllaceae | 0.039 | - | Li et al., 2017 |
| Castanea | Castanea | 0.004 | - | Li et al., 2017 |
| Cercis | Fabaceae | 0.023 | Dyakowska, 1936 | Flilipova-Marinova et al. , 2010 |
| Cerealia | Cerealia | 0.06 | Broström et al., 2004 | Poska et al., 2011 |
| Cerealia | Cerealia | 0.06 | Gregory, 1961 | Abraham and Kozáková, 2012 |
| Cerealia | Cerealia | 0.078 | - | Soepboer et al., 2007 |
| Cerealia type | Cerealia | 0.06 | Gregory, 1973 | Sugita et al., 1999 |
| Chenopodiaceae | Chenopodiaceae | 0.0108 | - | Xu et al., 2014 |
| Chenopodiaceae | Chenopodiaceae | 0.011 | - | Commerford et al., 2013 |
| Chenopodiaceae | Chenopodiaceae | 0.0117 | - | Wang and Herzschuh, 2011 |
| Chenopodiaceae | Chenopodiaceae | 0.019 | - | Abraham and Kozáková, 2012 |
| Cichorioideae | - | - | - | - |
| Comp. SF Cichorioideae | Asteraceae | 0.051 | - | Broström et al., 2004 |
| Cornus | Cornaceae | 0.044 | - | Commerford et al., 2013 |
| Corylus | Corylus | 0.025 | Gregory, 1973 | Cao et al., 2019 |
| Corylus | Corylus | 0.025 | Knoll as cited in Gregory, 1973 | Soepboer et al., 2007 |
| Cupressaceae | Cupressaceae | 0.01 | Li et al., 2017 | Li et al., 2018 |
| Cupressaceae | Cupressaceae | 0.01 | - | Li et al., 2017 |
| Cyperaceae | Cyperaceae | 0.014 | Li et al., in prep. | Li et al., 2018 |

| Pollen morphological type | Target taxon | FS (m/s) | Original publication | Found in: |
|---|---|---|---|---|
| Cyperaceae | Cyperaceae | 0.0152 | - | Xu et al., 2014 |
| Cyperaceae | Cyperaceae | 0.017 | Han et al., 2017 | Li et al., 2018 |
| Cyperaceae | Cyperaceae | 0.019 | Zhang et al., 2017 | Li et al., 2018 |
| Cyperaceae | Cyperaceae | 0.023 | Wu et al., 2013 | Li et al., 2018 |
| Cyperaceae | Cyperaceae | 0.026 | - | Bunting et al., 2013 |
| Cyperaceae | Cyperaceae | 0.028 | He et al., 2016 | Li et al., 2018 |
| Cyperaceae | Cyperaceae | 0.0291 | - | Wang and Herzschuh, 2011 |
| Cyperaceae | Cyperaceae | 0.035 | - | Sugita et al., 1999 |
| Cyperaceae | Cyperaceae | 0.037 | - | Li et al., 2017 |
| Elaeagnaceae | Elaeagnaceae | 0.012 | Zhang et al., 2017 | Li et al., 2018 |
| Elaeagnaceae | Elaeagnaceae | 0.019 | Zhang et al., 2017 | Li et al., 2018 |
| Empetrum | Ericales | 0.019 | - | Räsänen et al., 2007 |
| Ephedra | Ephedra | 0.015 | - | Xu et al., 2014 |
| Equisetum | Equisetum | 0.021 | - | Bunting et al., 2013 |
| Ericaceae | Ericales | 0.034 | Broström et al., 2004 | Cao et al., 2019 |
| Fabaceae | Fabaceae | 0.021 | - | Commerford et al., 2013 |
| Fagus | Fagus | 0.055 | Knoll as cited in Gregory, 1973 | Soepboer et al., 2007 |
| Fagus | Fagus | 0.057 | Gregory, 1973 | Mazier et al., 2008 |
| Fagus | Fagus | 0.057 | Mazier et al., 2012 | Chaput and Gajewski, 2018 |
| Fagus | Fagus | 0.0603 | Dyakowska, 1936 | Filipova-Marinova et al. , 2010 |
| Filipendula | Rosaceae | 0.006 | - | Broström et al., 2004 |
| Fraxinus | Fraxinus | 0.017 | - | Li et al., 2015 |
| Fraxinus | Fraxinus | 0.022 | Eisenhut, 1961 | Cao et al., 2019 |
| Gramineae | Poaceae | 0.035 | - | Sugita et al., 1999 |
| Hordeum (m) | PoaceaeCrop | 0.06 | Gregory, 1973 | Theuerkauf et al., 2015 |
| Hordeum type | PoaceaeCrop | 0.06 | - | Matthias et al., 2012 |
| Iridaceae | Iridaceae | 0.0121 | - | Xu et al., 2014 |
| Juglans | Juglans | 0.028 | Zhang et al., 2017 | Li et al., 2018 |
| Juglans | Juglans | 0.03 | - | Li et al., 2015 |
| Juglans | Juglans | 0.031 | Zhang et al., 2017 | Li et al., 2018 |
| Juglans | Juglans | 0.037 | Bodmer, 1922 | Li et al. 2015 |
| Juglans regia | Juglans | 0.037 | - | Li et al., 2017 |
| Juniperus | Juniperus | 0.016 | Eisenhut, 1961 | Broström et al., 2004 |
| Larix | Larix | 0.027 | Zhang et al., 2017 | Li et al., 2018 |
| Larix | Larix | 0.117 | Zhang et al., 2017 | Li et al., 2018 |
| Larix | Larix | 0.126 | Eisenhut, 1961 | Cao et al., 2019 |
| Larix | Larix | 0.131 | Eisenhut, 1961 | Li et al., 2015 |
| Larix | Larix | 0.135 | - | Li et al., 2015 |
| Lespedeza type | Fabaceae | 0.036 | - | Li et al., 2017 |
| Maclura | Moraceae | 0.016 | - | Commerford et al., 2013 |
| Nitraria | Nitraria | 0.016 | Li et al., 2011 | Li et al., 2018 |
| Picea | Picea | 0.056 | Eisenhut, 1961 | Cao et al., 2019 |
| Picea | Picea | 0.056 | Mazier et al., 2012 | Chaput and Gajewski, 2018 |

| Pollen morphological type | Target taxon | FS (m/s) | Original publication | Found in: |
|---|---|---|---|---|
| Pinus | Pinus | 0.028 | Li et al., 2017 | Li et al., 2018 |
| Pinus | Pinus | 0.03 | Zhang et al., 2017 | Li et al., 2018 |
| Pinus | Pinus | 0.031 | Eisenhut, 1961 | Cao et al., 2019 |
| Pinus | Pinus | 0.039 | Han et al., 2017 | Li et al., 2018 |
| Pinus | Pinus | 0.039 | - | Li et al., 2015 |
| Pinus | Pinus | 0.041 | Dyakowska, 1936 | Soepboer et al., 2007 |
| Pinus | Pinus | 0.041 | Zhang et al., 2017 | Li et al., 2018 |
| Pinus | Pinus | 0.028 | - | Li et al., 2017 |
| Plantago | Plantaginaceae | 0.024 | - | Mazier et al., 2008 |
| Plantago | Plantaginaceae | 0.03 | - | Mazier et al., 2008 |
| Plantago lanceolata | Plantaginaceae | 0.019 | - | Bunting et al., 2013 |
| Plantago lanceolata | Plantaginaceae | 0.029 | - | Broström et al., 2004 |
| Poaceae | Poaceae | 0.016 | - | Xu et al., 2014 |
| Poaceae | Poaceae | 0.016 | Xu et al., 2014 | Li et al., 2018 |
| Poaceae | Poaceae | 0.017 | Li et al., in prep. | Li et al., 2018 |
| Poaceae | Poaceae | 0.017 | - | Bunting et al., 2013 |
| Poaceae | Poaceae | 0.017 | Zhang et al., 2017 | Li et al., 2018 |
| Poaceae | Poaceae | 0.0185 | - | Wang and Herzschuh, 2011 |
| Poaceae | Poaceae | 0.02 | Zhang et al., 2017 | Li et al., 2018 |
| Poaceae | Poaceae | 0.022 | Han et al., 2017 | Li et al., 2018 |
| Poaceae | Poaceae | 0.022 | - | Li et al., 2017 |
| Poaceae | Poaceae | 0.023 | Li et al., 2011 | Li et al., 2018 |
| Poaceae | Poaceae | 0.032 | He et al., 2016 | Li et al., 2018 |
| Poaceae | Poaceae | 0.034 | Wu et al., 2013 | Li et al., 2018 |
| Poaceae | Poaceae | 0.035 | Sugita et al., 1999 | Mazier et al., 2008 |
| Populus | Populus | 0.025 | Eisenhut, 1961 | Matthias et al., 2012 |
| Populus | Populus | 0.027 | - | Commerford et al., 2013 |
| Potentilla type | Rosaceae | 0.0066 | - | Xu et al., 2014 |
| potentilla type | Rosaceae | 0.011 | - | Bunting et al., 2013 |
| Potentilla type | Rosaceae | 0.018 | - | Broström et al., 2004 |
| Pterocarya | Pterocarya | 0.042 | Eisenhut, 1961 | Fililpova-Marinova et al. , 2010 |
| Quercus | Quercus | 0.016 | Zhang et al., 2017 | Li et al., 2018 |
| Quercus | Quercus | 0.018 | Han et al., 2017 | Li et al., 2018 |
| Quercus | Quercus | 0.018 | - | Li et al., 2015 |
| Quercus | Quercus | 0.018 | Wu et al., 2013 | Li et al., 2018 |
| Quercus | Quercus | 0.019 | Zhang et al., 2017 | Li et al., 2018 |
| Quercus | Quercus | 0.035 | Eisenhut, 1961 | Cao et al., 2019 |
| Quercus | Quercus | 0.035 | Mazier et al. 2012 | Chaput and Gajewski, 2018 |
| Quercus | Quercus | 0.025 | - | Li et al., 2017 |
| Ranunculaceae | Ranunculaceae | 0.014 | - | Broström et al., 2004 |
| Ranunculaceae | Ranunculaceae | 0.015 | - | Bunting et al., 2013 |
| Rhinantus-type | Orobanchaceae | 0.038 | - | Bunting et al., 2013 |
| Robinia/Sophora | Fabaceae | 0.021 | - | Li et al., 2017 |

| Pollen morphological type | Target taxon | FS (m/s) | Original publication | Found in: |
|---|---|---|---|---|
| Rubiaceae | Rubiaceae | 0.019 | - | Broström et al., 2004 |
| Rumex acetosa | Rumex | 0.013 | - | Bunting et al., 2013 |
| Rumex acetosa | Rumex | 0.018 | Jackson and Lyford, 1999 | Bunting et al., 2013 |
| Rumex acetosa type | Rumex | 0.018 | - | Broström et al., 2004 |
| Rumex acetosella | Rumex | 0.009 | Broström et al., 2004 | Bunting et al., 2013 |
| Rumex acetosella | Rumex | 0.016 | - | Bunting et al., 2013 |
| Salix | Salix | 0.009 | - | Bunting et al., 2013 |
| Salix | Salix | 0.022 | Gregory, 1973 | Sugita et al., 1999 |
| Salix | Salix | 0.022 | Jackson and Lyford, 1999 | Bunting et al., 2013 |
| Salix | Salix | 0.022 | Sugita et al., 1999 | Poska et al., 2011 |
| Salix | Salix | 0.034 | Gregory, 1973 | Cao et al., 2019 |
| Sambucus nigra-type | Sambucus nigra-type | 0.013 | - | Abraham and Kozáková, 2012 |
| Saussurea/Carduus/Cirsium-type | Asteraceae | 0.075 | - | Li et al., 2017 |
| Secale cereale (m) | PoaceaeCrop | 0.06 | Gregory, 1973 | Theuerkauf et al., 2015 |
| Selaginella | Selaginella | 0.041 | - | Li et al., 2017 |
| Sinapis type (m) | Brassicaceae | 0.035 | - | Theuerkauf et al., 2015 |
| Solanum nigrum-type | Solanaceae | 0.027 | - | Li et al., 2017 |
| Thalictrum | Thalictrum | 0.0066 | - | Xu et al., 2014 |
| Thalictrum | Thalictrum | 0.012 | - | Bunting et al., 2013 |
| Thymus | Thymelaeaceae | 0.031 | - | Bunting et al., 2013 |
| Tilia | Tilia | 0.03 | - | Li et al., 2015 |
| Tilia | Tilia | 0.032 | Gregory, 1973 | Cao et al., 2019 |
| Triticum | PoaceaeCrop | 0.078 | Soepboer et al., 2007 | Matthias et al., 2012 |
| Triticum-Typ (b) | PoaceaeCrop | 0.06 | Gregory, 1973 | Theuerkauf et al., 2015 |
| Trollius europaeus | Ranunculaceae | 0.013 | - | Mazier et al., 2008 |
| Tsuga | Tsuga | 0.056 | Gaillard et al., 2008 | Chaput and Gajewski, 2018 |
| Ulmus | Ulmus | 0.0095 | - | Xu et al., 2014 |
| Ulmus | Ulmus | 0.01 | Xu et al., 2014 | Li et al., 2018 |
| Ulmus | Ulmus | 0.022 | Han et al., 2017 | Li et al., 2018 |
| Ulmus | Ulmus | 0.022 | - | Li et al., 2015 |
| Ulmus | Ulmus | 0.032 | Gregory, 1973 | Cao et al., 2019 |
| Ulmus | Ulmus | 0.032 | - | Li et al., 2017 |
| Urtica | Urtica | 0.007 | - | Abraham and Kozáková, 2012 |
| Vaccinium | Ericales | 0.029 | - | Räsänen et al., 2007 |
| Vitex negundo | Lamiaceae | 0.016 | - | Li et al., 2017 |
| wild herbs | wild herbs | 0.0343 | - | Matthias et al., 2012 |
| Zea mays | - | 0.185 | - | Li et al., 2017 |

670

**References Appendix**

*Fall speeds*

Abraham, V. and Kozáková, R.: Relative pollen productivity estimates in the modern agricultural landscape of Central Bohemia (Czech Republic), Rev Palaeobot Palyno, 179, 1–12, doi:10.1016/j.revpalbo.2012.04.004, 2012.

Bodmer, H.: Über den Windpollen., 1922.

Broström, A.: Estimating source area of pollen and pollen productivity in the cultural landscapes of southern Sweden - developing a palynological tool for quantifying past plant cover, Quaternary Sci, Department of Geology, Lund University. [online] Available from: http://lup.lub.lu.se/record/464945 (Accessed 26 November 2018), 2002.

Broström, A., Sugita, S. and Gaillard, M.-J.: Pollen productivity estimates for the reconstruction of past vegetation cover in the cultural landscape of southern Sweden, The Holocene, 14(3), 368–381, doi:10.1191/0959683604hl713rp, 2004.

Bunting, M. J., Schofield, J. E. and Edwards, K. J.: Estimates of relative pollen productivity (RPP) for selected taxa from southern Greenland: A pragmatic solution, Rev Palaeobot Palyno, 190, 66–74, doi:10.1016/j.revpalbo.2012.11.003, 2013.

Cao, X., Tian, F., Li, F., Gaillard, M.-J., Rudaya, N., Xu, Q. and Herzschuh, U.: Pollen-based quantitative land-cover reconstruction for northern Asia covering the last 40 ka cal BP, Climate of the Past, 15(4), 1503–1536, doi:https://doi.org/10.5194/cp-15-1503-2019, 2019.

Chaput, M. A. and Gajewski, K.: Relative pollen productivity estimates and changes in Holocene vegetation cover in the deciduous forest of southeastern Quebec, Canada, Botany, 96(5), 299–317, doi:10.1139/cjb-2017-0193, 2018.

Commerford, J. L., McLauchlan, K. K. and Sugita, S.: Calibrating vegetation cover and grassland pollen assemblages in the Flint Hills of Kansas, USA, American J Plant Sci, 4(7A1), 1–10, doi:10.4236/ajps.2013.47A1001, 2013.

Dyakowska, J.: Researches on the rapidity of the falling down of pollen of some trees, nakl. Polskiej Akademji Umiejętności., 1936.

Eisenhut, G.: Untersuchungen über die Morphologie und Ökologie der Pollenkörner Heimischer und Fremdländischer Waldbäume, Paul Parey, Hamburg., 1961.

Filipova-Marinova, M. V., Kvavadze, E. V., Connor, S. E. and Sjögren, P.: Estimating absolute pollen productivity for some European Tertiary-relict taxa, Veget Hist Archaeobot, 19(4), 351–364, doi:10.1007/s00334-010-0257-z, 2010.

Gaillard, M.-J., Sugita, S., Bunting, J., Dearing, J. and Bittmann, F.: Human impact on terrestrial ecosystems, pollen calibration and quantitative reconstruction of past land-cover, Veget Hist Archaeobot, 17(5), 415–418, doi:10.1007/s00334-008-0170-x, 2008.

Ge, Y., Li, Y., Yang, X., Zhang, R. and Xu, Q.: Relevant source area of pollen and relative pollen productivity estimates in Bashang steppe, Quat Sci, 35(4), 934-945 (In Chinese with English abstract), 2015.

Gregory, P. H.: The microbiology of the atmosphere, London, L. Hill; New York, Interscience Publishers. [online] Available from: http://archive.org/details/microbiologyofat00greg (Accessed 3 May 2019), 1961.

Gregory, P. H.: Microbiology of the atmosphere, Leonhard Hill, Aylesbury (Bucks.), 1973.

Han, Y., Liu, H., Hao, Q., Liu, X., Guo, W. and Shangguan, H.: More reliable pollen productivity estimates and relative source area of pollen in a forest-steppe ecotone with improved vegetation survey, The Holocene, 27(10), 1567–1577, doi:10.1177/0959683617702234, 2017.

He, F., Li, Y., Wu, J. and Xu, Y.: A comparison of relative pollen productivity from forest steppe, typical steppe and desert steppe in Inner Mongolia, Chin Sci Bull, 61(31), 3388-3400 (In Chinese with English abstract), doi:10.1360/N972016-00482, 2016.

Hjelle, K. L. and Sugita, S.: Estimating pollen productivity and relevant source area of pollen using lake sediments in Norway: How does lake size variation affect the estimates?, The Holocene, 22(3), 313–324, doi:10.1177/0959683611423690, 2012.

Jackson, S. T. and Lyford, M. E.: Pollen dispersal models in Quaternary plant ecology: Assumptions, parameters, and prescriptions, Bot. Rev, 65(1), 39–75, doi:10.1007/BF02856557, 1999.

Knoll F.: Über die Fernverbreitung des Blütenstaubes durch den Wind. Forschungen und Fortschritte: Nachrichtenbl. Deutsch. Wiss. Tech. 8:301–302, 1932

Li, F., Gaillard, M. J., Sugita, S., Mazier, F., Xu, Q., Zhou, Z., Zhang, Y., Li, Y. and Laffly, D.: Relative pollen productivity estimates for major plant taxa of cultural landscapes in central eastern China, Veget Hist Archaeobot, 26(6), 587–605, doi:10.1007/s00334-017-0636-9, 2017.

Li, F., Gaillard, M. J., Xu, Q., Bunting, M. J., Li, Y., Li, J., Mu, H., Lu, J., Zhang, P., Zhang, S., Cui, Q., Zhang, Y. and Shen, W.: A Review of Relative Pollen Productivity Estimates From Temperate China for Pollen-Based Quantitative Reconstruction of Past Plant Cover, Front Plant Sci, 9, 1214, doi:10.3389/fpls.2018.01214, 2018.

Li, Y., Bunting, M. J., Xu, Q., Jiang, S., Ding, W. and Hun, L.: Pollen–vegetation–climate relationships in some desert and desert-steppe communities in northern China, The Holocene, 21(6), 997–1010, doi:10.1177/0959683611400202, 2011.

Li, Y., Nielsen, A. B., Zhao, X., Shan, L., Wang, S., Wu, J. and Zhou, L.: Pollen production estimates (PPEs) and fall speeds for major tree taxa and relevant source areas of pollen (RSAP) in Changbai Mountain, northeastern China, Rev Palaeobot Palyno, 216, 92–100, doi:10.1016/j.revpalbo.2015.02.003, 2015.

Matthias, I., Nielsen, A. B. and Giesecke, T.: Evaluating the effect of flowering age and forest structure on pollen productivity estimates, Veget Hist Archaeobot, 21(6), 471–484, doi:10.1007/s00334-012-0373-z, 2012.

Mazier, F., Broström, A., Gaillard, M.-J., Sugita, S., Vittoz, P. and Buttler, A.: Pollen productivity estimates and relevant source area of pollen for selected plant taxa in a pasture woodland landscape of the Jura Mountains (Switzerland), Veget Hist Archaeobot, 17(5), 479–495, doi:10.1007/s00334-008-0143-0, 2008.

Mazier, F., Gaillard, M. J., Kuneš, P., Sugita, S., Trondman, A. K. and Broström, A.: Testing the effect of site selection and parameter setting on REVEALS-model estimates of plant abundance using the Czech Quaternary Palynological Database, Rev Palaeobot Palyno, 187, 38–49, doi:10.1016/j.revpalbo.2012.07.017, 2012.

Poska, A., Meltsov, V., Sugita, S. and Vassiljev, J.: Relative pollen productivity estimates of major anemophilous taxa and relevant source area of pollen in a cultural landscape of the hemi-boreal forest zone (Estonia), Rev Palaeobot Palyno, 167(1), 30–39, doi:10.1016/j.revpalbo.2011.07.001, 2011.

Räsänen, S., Suutari, H. and Nielsen, A. B.: A step further towards quantitative reconstruction of past vegetation in Fennoscandian boreal forests: Pollen productivity estimates for six dominant taxa, Rev Palaeobot Palyno, 146(1), 208–220, doi:10.1016/j.revpalbo.2007.04.004, 2007.

Soepboer, W., Sugita, S., Lotter, A. F., van Leeuwen, J. F. N. and van der Knaap, W. O.: Pollen productivity estimates for quantitative reconstruction of vegetation cover on the Swiss Plateau, The Holocene, 17(1), 65–77, doi:10.1177/0959683607073279, 2007.

Sugita, S.: A Model of pollen source area for an entire lake surface, Quaternary Res, 39(2), 239–244, doi:10.1006/qres.1993.1027, 1993.

[revised manuscript text omitted]